# Interleukin-6 trans-signaling is a candidate mechanism to drive progression of human DCCs during clinical latency

Melanie Werner-Klein [1,12✉], Ana Grujovic[1,9,12], Christoph Irlbeck [1,2], Milan Obradović[1,10], Martin Hoffmann [2], Huiqin Koerkel-Qu[1], Xin Lu[2], Steffi Treitschke[2], Cäcilia Köstler[2], Catherine Botteron[2], Kathrin Weidele[2], Christian Werno[2], Bernhard Polzer [2], Stefan Kirsch[2], Miodrag Gužvić [1], Jens Warfsmann[2], Kamran Honarnejad[2], Zbigniew Czyz[1], Giancarlo Feliciello[2], Isabell Blochberger[1], Sandra Grunewald[1], Elisabeth Schneider[1], Gundula Haunschild[1], Nina Patwary[1], Severin Guetter [1], Sandra Huber[1], Brigitte Rack[3,11], Nadia Harbeck[3], Stefan Buchholz[4], Petra Rümmele [5,6], Norbert Heine[7], Stefan Rose-John [8] & Christoph A. Klein [1,2✉]

Although thousands of breast cancer cells disseminate and home to bone marrow until primary surgery, usually less than a handful will succeed in establishing manifest metastases months to years later. To identify signals that support survival or outgrowth in patients, we profile rare bone marrow-derived disseminated cancer cells (DCCs) long before manifestation of metastasis and identify IL6/PI3K-signaling as candidate pathway for DCC activation. Surprisingly, and similar to mammary epithelial cells, DCCs lack membranous IL6 receptor expression and mechanistic dissection reveals IL6 trans-signaling to regulate a stem-like state of mammary epithelial cells via gp130. Responsiveness to IL6 trans-signals is found to be niche-dependent as bone marrow stromal and endosteal cells down-regulate gp130 in premalignant mammary epithelial cells as opposed to vascular niche cells. *PIK3CA* activation renders cells independent from IL6 trans-signaling. Consistent with a bottleneck function of microenvironmental DCC control, we find *PIK3CA* mutations highly associated with late-stage metastatic cells while being extremely rare in early DCCs. Our data suggest that the initial steps of metastasis formation are often not cancer cell-autonomous, but also depend on microenvironmental signals.

[1] Experimental Medicine and Therapy Research, University of Regensburg, Regensburg, Germany. [2] Division of Personalized Tumour Therapy, Fraunhofer Institute for Toxicology and Experimental Medicine, 93053 Regensburg, Germany. [3] Department OB&GYN and CCCLMU, Breast Center, LMU University Hospital, 80337 Munich, Germany. [4] Clinic of Gynecology and Obstetrics, University Medical Center Regensburg, 93053 Regensburg, Germany. [5] Department of Pathology, University of Regensburg, 93053 Regensburg, Germany. [6] Institute of Pathology, University Hospital, Friedrich-Alexander-University Erlangen-Nürnberg, 91054 Erlangen, Germany. [7] University Center of Plastic-, Aesthetic, Hand- and Reconstructive Surgery, University of Regensburg, Regensburg, Germany. [8] Institute of Biochemistry, Christian-Albrechts-Universität Kiel, Kiel, Germany. [9] Present address: Telexos GmbH, Weilheim, Germany. [10] Present address: Wellmera AG, Basel, Switzerland. [11] Present address: Department of Gynecology, Ulm University Hospital, Ulm, Germany. [12] These authors contributed equally: Melanie Werner-Klein, Ana Grujovic. ✉email: melanie.werner-klein@ukr.de; Christoph.klein@ukr.de

In breast cancer, dissemination to distant sites precedes the clinical manifestation of metastasis by 6–8 years in the median, ranging from <1 year to >40 years[1–3]. These clinical data derived from breast cancer growth kinetics and imaging studies are strongly supported by recent experimental evidence. Whereas dissemination from the primary site occurs preferentially in early tumor stages[4–6], specific mechanisms reduce cancer cell dissemination in anatomically and molecularly advanced stages[5]. Furthermore, analysis of cancer growth kinetics suggests that not only cancer cells disseminate early but also that micrometastatic colony formation is initiated early. However, its manifestation may take considerable time[1,3,7]. Such data are consistent with the observation that early disseminated cancer cells (DCCs) often lack critical genetic and genomic alterations, which they need to acquire at the distant site in breast and other cancers. This process could explain the much longer clinical latency periods observed in humans as compared to mouse models[5,8,9] and be particularly relevant for cancers displaying late relapses, such as hormone receptor-positive (HR$^+$) breast cancer. However, early dissemination and prolonged clinical latency at distant sites raise questions about the identity and nature of signals conferring survival, genomic progression, and outgrowth of DCCs over extended periods of time.

DCCs are extremely rare. They are detected at very low frequencies (1–2 DCCs per $10^6$ BM cells[10,11]) in bone marrow (BM) of ~30% of breast cancer patients with no evidence of manifest metastasis. Besides genomic studies, the assessment of the DCC phenotype has been limited to testing for selected antigens[12] and to anecdotal transcriptomic studies[13,14]. Moreover, spontaneous or transgenic mouse models, such as the PyMT- or Her2-driven models, do not generate bone metastases. Hence, there is no in vivo model available to study the spontaneous progression and genomic evolution from early BM infiltration to manifestation of bone metastasis.

To unravel mechanisms operative during clinical latency periods, we interrogate transcript-derived pathway information from DCCs isolated from BM of breast cancer patients. Since early breast cancer DCCs often display close-to-normal genomes[5,15], we use mammary epithelial cells isolated from reduction mammoplasties and available immortalized premalignant breast cancer cell lines as cellular models for functional testing of candidate mechanisms in vitro. We identified interleukin-6 (IL-6) trans-signaling as a pathway that (i) activates normal and premalignant cells, (ii) induces a proliferative stem/progenitor-like phenotype in mammary epithelial cells, and (iii) whose activation in DCCs depends on regulatory niche cells in BM. These data shed light onto the so far dark stage of early metastasis formation in patients. Moreover, it may inform about future ways to delay or prevent metachronous metastasis in patients whose breast cancer is diagnosed to be locally confined by standard clinical means.

## Results

**Early DCCs do not engraft in immunodeficient mice.** For decades attempts to culture early DCCs, that is, DCCs from nonmetastasized M0-stage patients, have failed. Only anecdotal reports have been published that were not reproduced since then[16–18]. We recently observed in melanoma that early DCCs failing to generate xenografts differ genomically from DCCs that successfully engrafted, indicating a causative role of genomic maturation for metastasis and xenograft formation. Specific alterations were identified that are closely linked to colonization success in mice and patients[9]. We repeated these experiments for BM-derived breast and prostate cancer DCCs from early (M0) and advanced (M1) stages. Given the very low frequency of BM-DCCs (<$10^{-6}$), we either injected CD45-depleted or EpCAM-

enriched BM cells or generated and transplanted spheres as these have a higher engraftment likelihood[9]. In total, we tested 46 patient samples and different routes of application, including subcutaneous, orthotopic (site of origin), intrafemoral and intravenous injection. We then assessed tumor formation at the cutaneous injection sites and metastatic spread to lungs or BM. BM-derived DCCs from M1-stage patients engrafted in two out of four cases. In contrast, early DCCs from 42 M0-stage patients did not establish xenografts (Fig. 1a, b), neither at the injection sites nor in the lungs ($p = 0.006$; Fisher's exact test). We also explored the presence of minimal systemic cancer by testing for human cytokeratin (CK) or EpCAM$^+$ cells in murine BM. Interestingly, albeit DCCs of nonmetastatic patients did not expand in mice, they survived in murine BM in 4 out of 42 cases. We detected human EpCAM$^+$ or CK$^+$ DCCs at a frequency of 1–5 DCCs/million BM cells 4–14 weeks after injection of CD45-depleted human BM cells (Fig. 1c). For one of these rare events, we could not only prove human and epithelial, but also malignant origin by single-cell copy number alteration (CNA) analysis (Fig. 1d).

In summary and consistent with our findings in melanoma, early DCCs from patients without manifest metastasis failed to generate xenografts. Besides lower absolute cell numbers and fewer genetic alterations (see below), microenvironmental dependence of early DCCs could account for these results. We therefore decided to retrieve candidate interactions of early DCCs with the microenvironment via direct molecular analysis of early DCCs from breast cancer patients and implement these results into surrogate in vitro models.

**Pathway activation in mammary stem and progenitor cells**. We hypothesized that stemness traits are necessary for the ability to survive and progress in a hostile environment and to initiate metastasis. Therefore, we tested for pathways activated in cells with progenitor or stem-like traits using our highly sensitive whole transcriptome amplification (WTA) method[14,19]. To identify these cells, we labeled freshly isolated primary human mammary epithelial cells (HMECs) from reduction mammoplasties of healthy patients with the membrane dye PKH26. Labeled cells were then cultured under nonadherent mammosphere conditions, which support the expansion of stem/early progenitor cells and formation of multicellular spheroids of clonal origin with self-renewing capacity[20]. Cell divisions during mammosphere formation diluted the dye until only a few label-retaining cells (LRCs) were visible under the microscope (Fig. 2a). Isolating LRCs and non-LRCs (nLRCs) from disaggregated PKH26-labeled HMEC spheres and plating them as single cell per well confirmed that the sphere-forming ability was solely confined to LRCs (Fig. 2b, Fisher's exact test $p = 0.02$), which is consistent with previous findings[21]. For transcriptome analysis, we isolated: (i) LRCs, (ii) nLRCs from disaggregated spheres, and (iii) LRCs that had not divided or formed spheres over time, but remained as single cells and were therefore termed quiescent single cells (QSCs). From each group, we isolated single cells (LRC, $n = 8$; nLRC, $n = 5$; QSC, $n = 10$; Supplementary Table 1), and performed WTA and microarray analysis as previously described[19,22]. Bioinformatics analysis indicated most variable gene expression between LRCs and QSCs/nLRCs (Fig. 2c and Supplementary Tables 2 and 3), and we found 12 pathways significantly enriched in LRCs over QSCs/nLRCs (Fig. 2d and Supplementary Table 2).

**Identification of EpCAM$^+$ DCCs in BM**. In order to test whether any of these pathways were enriched in DCCs isolated from BM of breast cancer patients, we aimed to isolate DCCs with

**a**

| Stage | Cancer type | DCC isolation | Applicat. route (no. of patients) | Mouse strain | Median number of injected cells or spheres | No. of patient BM with DCCs | No. of xenograft formed/ no. of patient samples injected | Weeks until detect. | % Patients with DCCs in mouse BM |
|---|---|---|---|---|---|---|---|---|---|
| M1 | BrCa | CD45 depl. | s.c. (2) i.f. (2) i.v. (2) mfp (2) | NOD-*scid* IL-2Rγ$^{-/-}$ | $2 \times 10^4$ cells | 2/2 | 1/2 | 24 | — |
| M1 | PC | CD45 depl. | s.c (2) i.f. (2) i.v. (2) | NOD-*scid* IL-2Rγ$^{-/-}$ | $5 \times 10^3$ cells | 2/2 | 1/2 | 19 | — |
| M0 | BrCa | CD45 depl. | i.v. (4) | NOD-*scid* | $1.8 \times 10^6$ cells | 1/4* | 0/4 | — | 2/4* |
| M0 | BrCa | Spheres | s.c. (1) s.c. + s.r. (2) mfp (10) | NOD-*scid* IL-2Rγ$^{-/-}$ | 20 spheres | 6/13 | 0/13 | — | 0/13 |
| M0 | PC | EpCAM enrichment | i.v. (8) | NOD-*scid* | $2.2 \times 10^5$ cells | 1/8 | 0/8 | — | 0/8 |
| M0 | PC | CD45 depl. | i.v. (8) | NOD-*scid* | $3.8 \times 10^6$ cells | 3/8 | 0/8 | — | 2/8 |
| M0 | PC | Spheres | s.c. (9) | NOD-*scid* IL-2Rγ$^{-/-}$ | 17 spheres | 7/9 | 0/9 | — | 0/9 |

*Note: only one half of the sample's volume was injected into mice, since the other half was used for DCC enumeration (EpCAM for PC; CK for BrCa). Due to their low numbers, DCCs may have been unequally distributed, as indicated by one sample negative at DCC enumeration (see No. of patient BM with DCCs), but harboring positive DCCs in mouse bone marrow (see % patients with surviving DCCs in mouse BM).

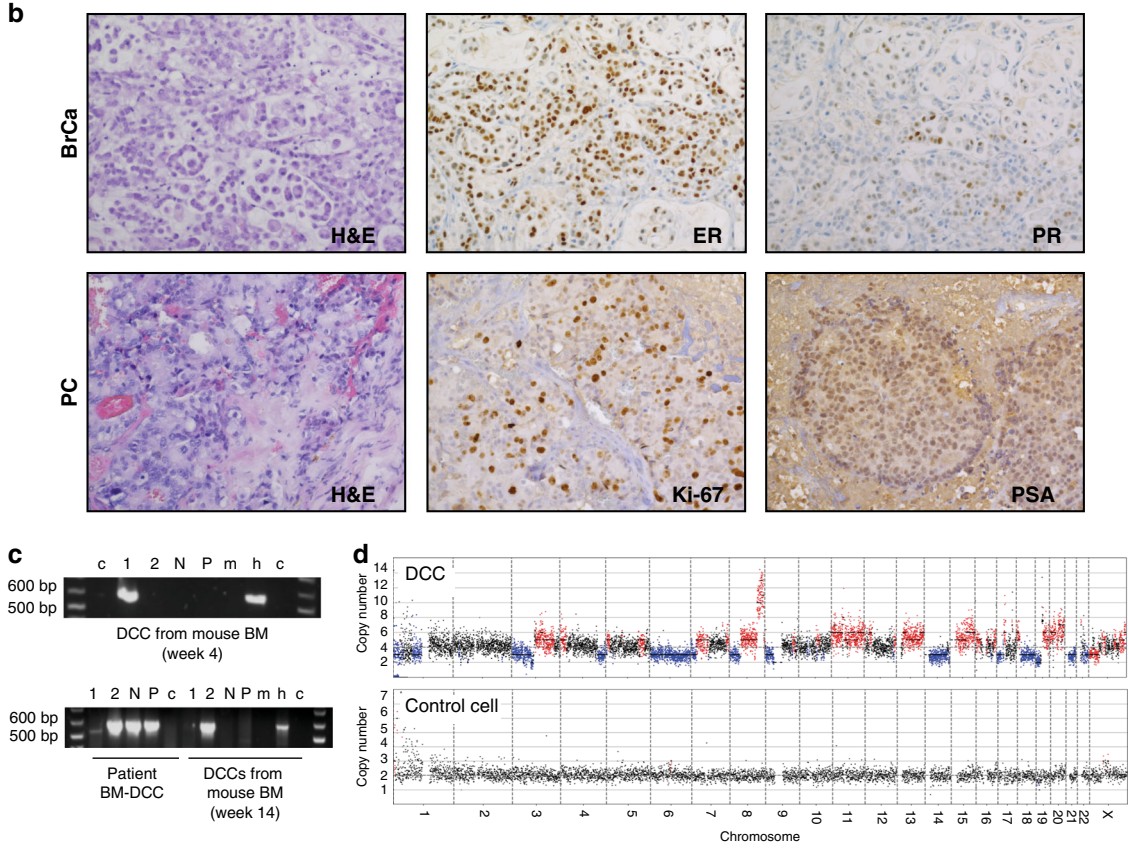

**b**

BrCa: H&E, ER, PR

PC: H&E, Ki-67, PSA

**c**

DCC from mouse BM (week 4)

600 bp / 500 bp

c 1 2 N P m h c

Patient BM-DCC / DCCs from mouse BM (week 14)

600 bp / 500 bp

1 2 N P c 1 2 N P m h c

**d**

DCC — Copy number — Chromosome

Control cell — Copy number — Chromosome

confirmed malignant origin[14]. We followed the reasoning that epithelial cell identity in BM of a cancer patient plus the presence of genetic alterations is sufficient to claim malignant epithelial origin or malignant potential of a cell. DCCs were detected by screening diagnostic BM aspirates from 246 M0-stage and 18 M1-stage patients for cells that stained positively for the epithelial marker EpCAM (Supplementary Fig. 1a). Forty percent of M0-stage and 72% of M1-stage patients harbored EpCAM$^+$ cells.

However, EpCAM is a surface marker that is not as specific for DCCs as the diagnostically used CKs[14] in BM because it is expressed by cells from the B cell lineage (own unpublished data and ref. [23]). Therefore, we sought to differentiate between EpCAM$^+$ cells of breast cancer patients and noncancer patients.

CNAs are found in <5% of nonmalignant cells with a median of 1.8%[24–26] and are diagnostically used to differentiate normal and malignant cells[27]. We therefore performed combined genome

**Fig. 1 Xenotransplantation of DCCs. a** Diagnostic bone marrow aspirates from breast (BrCa, $n = 19$) or prostate (PC, $n = 27$) cancer patients (M0- or M1-stage of disease) were either CD45-depleted, enriched for EpCAM, or cultured under sphere conditions. Resulting spheres, CD45-depleted, or EpCAM-enriched BM cells were injected intra-venously (i.v.), intra-femorally (i.f.), sub-cutaneously (s.c.), sub-renally (s.r.), or into the mammary fat pad (mfp) of NOD-scid or NOD-scidIL2Rγ-/- mice. Mice with sub-cutaneous or mammary fat pad injections were palpated weekly. All other mice were observed until signs of illness or were sacrificed after 9 months. Injection routes that led to xenograft formation are highlighted in red. **b** Immunohistochemistry for estrogen-receptor (ER), progesterone-receptor (PR), prostate-specific antigen (PSA), Ki-67, or H & E staining of M1-DCC-derived xenografts is shown. **c** Human EpCAM- or cytokeratin 8/18/19-expressing DCCs were detected in the BM of 4/42 mice transplanted with M0-stage patient samples. DCCs from two of the four mice were isolated and their human origin was verified by a PCR specific for human KRT19. Pure mouse or human DNA was used as control. 1, 2 = cytokeratin 8/18/19-positive DCCs; N = cytokeratin 8/18/19-negative BM-cell, P = pool of BM-cells of recipient mouse; m = mouse positive control; h = human positive control, c = non-template control. **d** Single cell CNA analysis of the EpCAM-expressing DCC isolated at 4 weeks after injection from NSG BM (**c**) and a human hematopoietic cell as control. Red or blue indicate gain or loss of chromosomal regions.

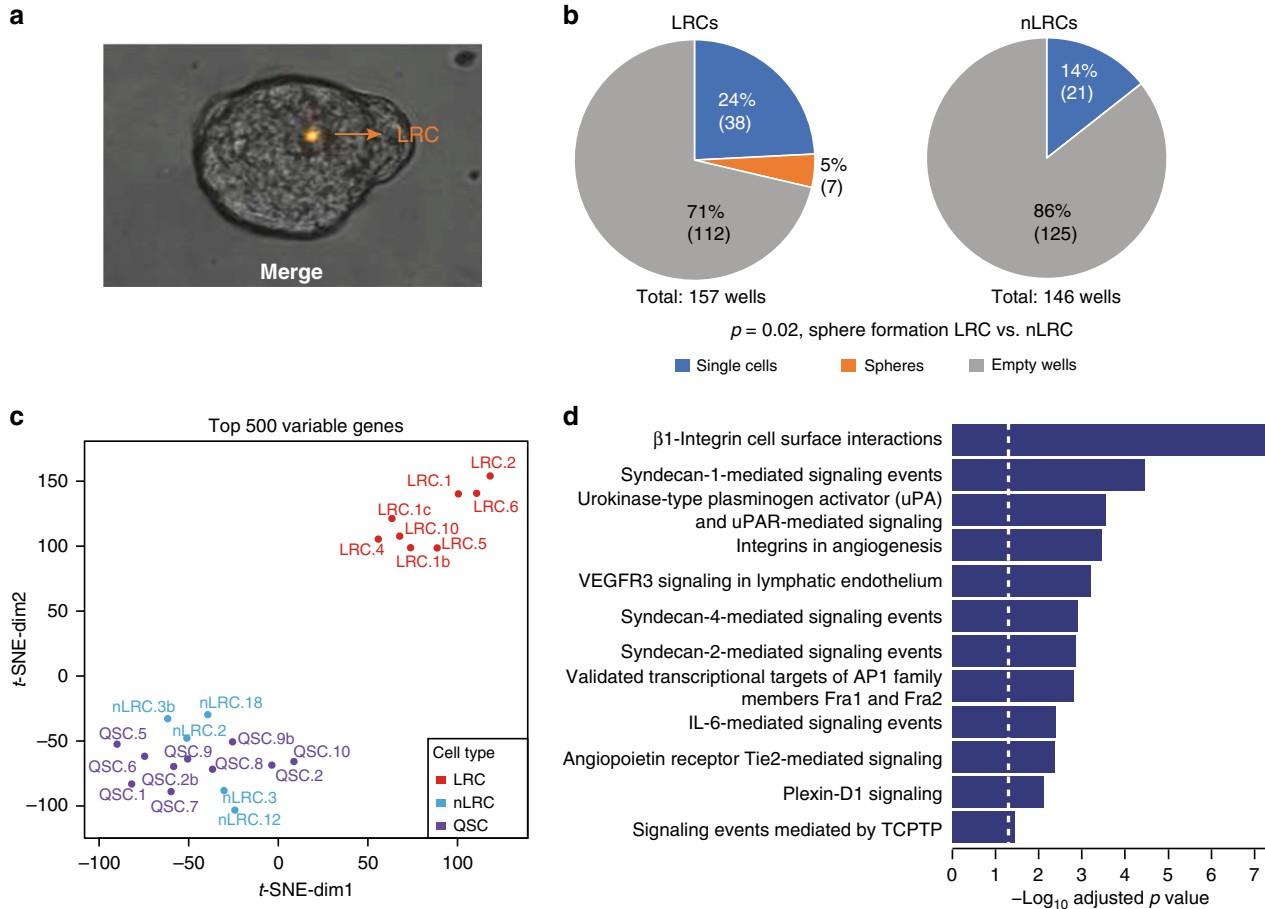

**Fig. 2 IL6 pathway is activated in mammary stem cells. a** Representative picture of a mammosphere (day 7) generated from PKH26-labeled HMECs. **b** PKH26-labeled HMEC-mammospheres from four patients were disaggregated, sorted by flowcytometry into PKH26+ LRCs and PKH26- nLRCs, plated as single cell per well, and tested for sphere-formation. Shown is the percentage and in parentheses the respective absolute number of empty wells, wells with single cells and spheres after two weeks of mammosphere-culture (number of spheres vs. no spheres for LRCs vs. nLRCs, $p = 0.02$, two-sided Fisher's exact test). **c**, **d** LRCs ($n = 8$), nLRCs ($n = 5$) and QSCs ($n = 10$) from three patients were subjected to single cell transcriptome microarray analysis. **c** t-SNE plot of the top 500 most variable genes. **d** Pathway analysis using the 216 genes differentially expressed between LRCs and the pooled nLRCs plus QSCs. See Supplementary Table 1 for patient/sample-ID allocation.

and transcriptome analysis and isolated genomic DNA and mRNA from the same single cell[5,14]. Although this approach fails in 10–50%[13,14,28] and a CNA profile cannot be obtained for every cell, we found that 50% and 88% of successfully analyzed EpCAM+ cells from M0- and M1-stage patients, respectively, harbored CNAs (Fig. 3a and Supplementary Fig. 1a, b). We selected these DCCs for single-cell RNA-sequencing (RNA-Seq) analysis (M0: $n = 30$ DCCs, 21 patients; M1: $n = 11$ DCCs, 5 patients). To provide additional evidence that the aberrant

EpCAM+ cells are derived from a nonhematopoietic cell lineage, we compared them with autochthonous EpCAM+ BM cells. The latter were isolated from patients without known malignant disease undergoing hip replacement surgery. Of note, cancer patient-derived and noncancer patient-derived EpCAM+ cells could be clearly separated using the overall gene expression as well as epithelial and B cell-annotated genes—with the exception of one M0-stage cell, which was therefore excluded from further analysis (Fig. 3b). Moreover, many cells from M0-stage breast

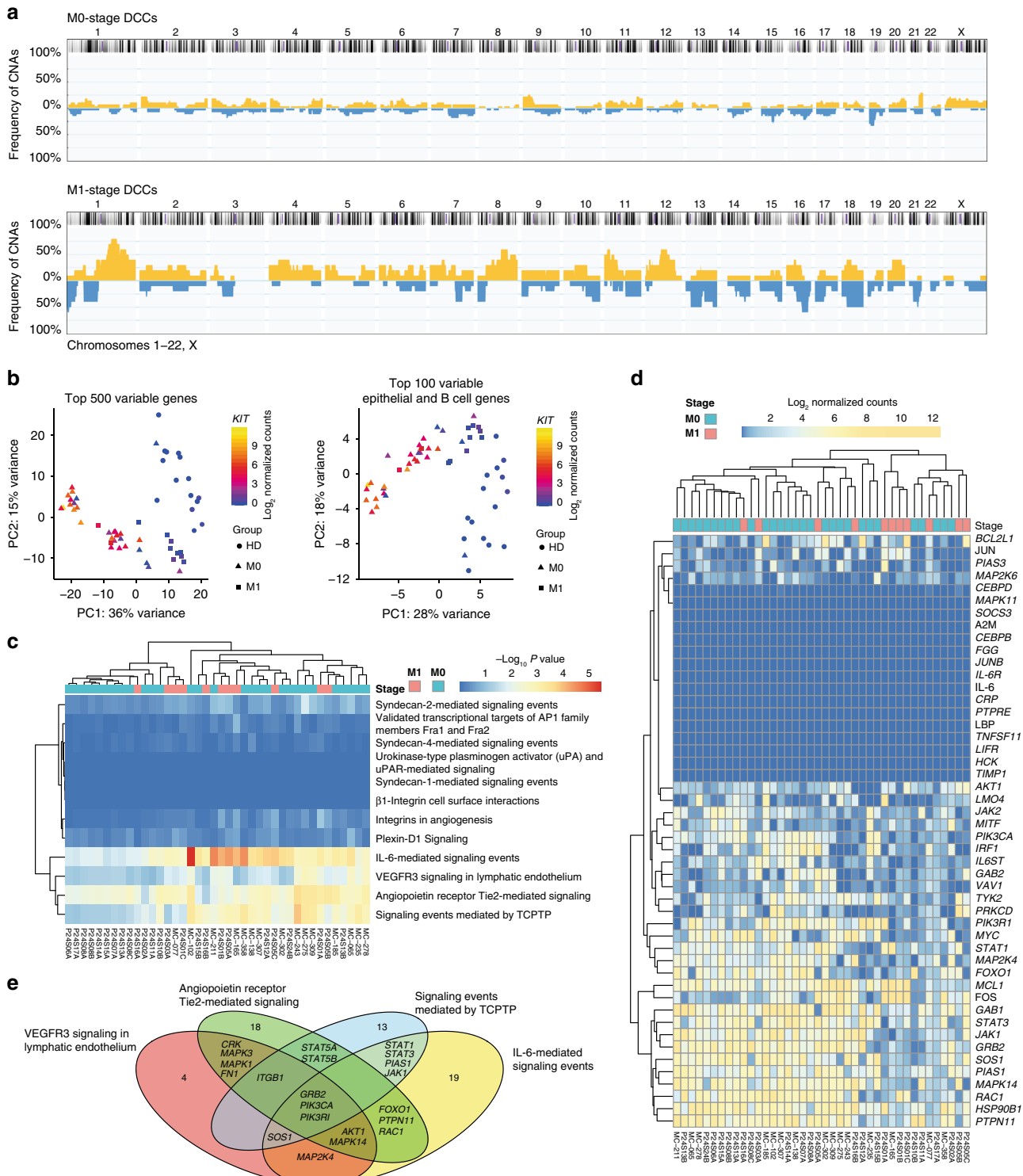

**Fig. 3 IL6 pathway is activated in BM-DCCs of breast cancer patients. a** DCCs from BM of 21 non-metastasized (M0-stage, $n = 30$ DCCs) and five metastasized (M1-stage; $n = 11$ DCCs) breast cancer patients were analyzed for CNAs. The cumulative frequency of genomic aberrations is given in yellow and blue indicating genomic gains and losses, respectively. M0-stage (**b**, **c** $n = 30$; **d**, **e** $n = 29$ and M1-stage ($n = 11$) DCCs and EpCAM+ BM cells of seven healthy donors, i.e. patients without malignant disease (HD; $n = 15$ cells) were analyzed by single cell RNA sequencing. **b** Principal component analysis of the top 500 most variable genes or top 100 most variable epithelial and B cell genes. **c** DCCs were tested for enrichment in pathways identified to be enriched in LRCs over QSCs/nLRCs (see Fig. 2d). **d** The heatmap displays log2 normalized read counts of mRNA expression of IL6 signaling pathway genes as listed in the NCI-Nature PID expanded by the LIFR gene. **e** Venn diagram for the gene-members of the four pathways (**d**) that are expressed in at least half of bone marrow DCCs (except for the BMX (5/40) and CEBPD (19/40) genes, Supplementary Fig. 1d). Pathway-private genes are annotated by their number, shared genes are named explicitly (see also Supplementary Table 5). See Supplementary Table 1 for patient/sample-ID allocation.

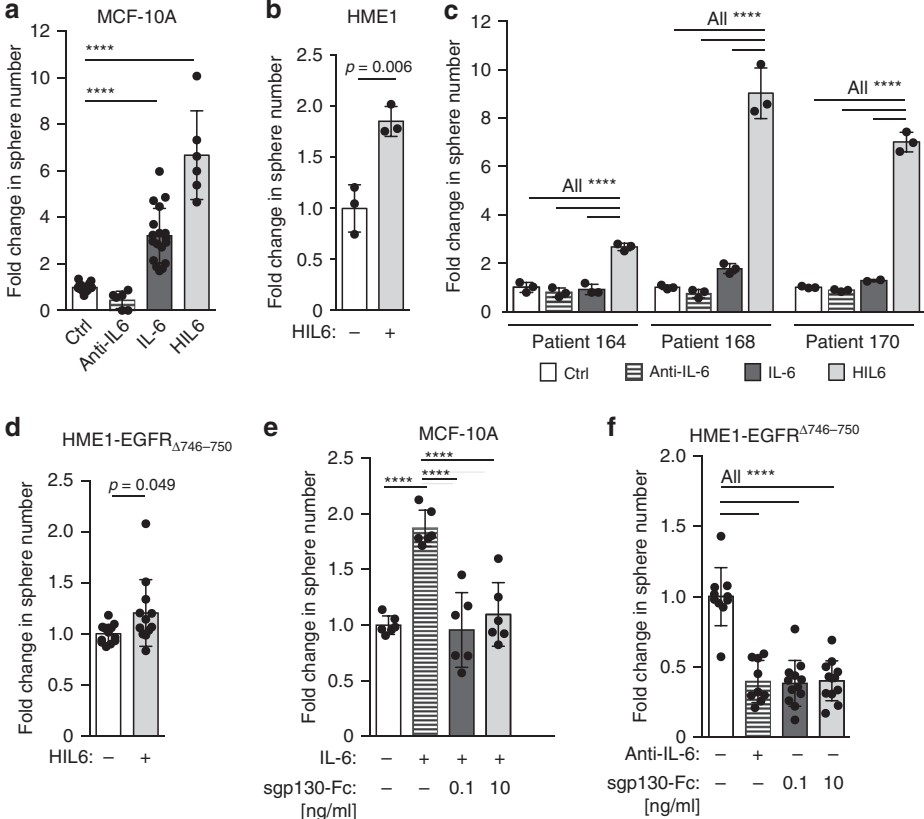

**Fig. 4 IL6 trans-signaling regulates sphere-forming ability. a** MCF 10A cells were cultured as spheres in the absence ($n = 18$) or presence of IL6 ($n = 18$), an IL6-blocking antibody ($n = 6$) or hyper-IL6 ($n = 6$). **b** hTERT-HME1 were cultured as spheres in the absence ($n = 3$) or presence ($n = 3$) of hyper-IL6. **c** HMECs were cultured as spheres without or with IL6, with an IL6 blocking antibody or hyper-IL6. $n = 3$ patients, each patient analyzed in triplicate. **d** hTERT-HME1-EGFRΔ746-750 cells were cultured as spheres in the absence or presence of HIL6 (each $n = 12$). **e** MCF 10A cells were cultured as spheres without ($n = 6$) or with IL6 ($n = 6$) and IL6 plus sgp130-Fc at indicated concentrations (each $n = 6$). **f** Sphere formation of hTERT-HME1-EGFRΔ746-750 in the absence ($n = 10$) or presence of an anti-IL6 antibody ($n = 9$) or with sgp130-Fc at indicated concentrations (each $n = 12$). Cumulative data of three experiments. $P$ values in **a**, **c**, **f**: one-way ANOVA with Dunnett's multiple comparisons test (post hoc); **b**, **d** two-sided Student's t-test; e one-way ANOVA with Tukey's multiple comparisons test (post hoc); asterisks indicate significance between groups (****$p < 0.0001$). All error bars correspond to standard deviation (Mean ± SD). See Supplementary Table 1 for patient/sample-ID allocation.

cancer patients strongly expressed the marker gene *KIT* (Fig. 3b and Supplementary Fig. 1c), characteristic for mammary luminal progenitor (LumProg) cells[29] (see below). *KIT* expression was significantly higher in M0-stage DCCs compared to M1-stage DCCs and absent in EpCAM+ cells from noncancer patients. *KIT* transcript expression in M0-stage DCCs was comparable to *EPCAM* transcript expression (Supplementary Fig. 1c). Together, CNAs and the epithelial, nonhematopoietic phenotype of cells isolated from patients with breast cancer provided compelling evidence that the selected cells were true DCCs.

**IL-6 pathway activation in DCCs**. We then tested whether any of the pathways enriched in mammary stem cells (LRCs; Fig. 2d) were also enriched in DCCs using the pathway membership enrichment analysis. We found 4 out of the 12 pathways to be significantly enriched in DCCs (Fig. 3c, d and Supplementary Table 4), including the pathway "IL-6-mediated signaling events," the "TCPTP" pathway, the "VEGF-VEGFR3," and "angiopoietin-Tie2 receptor" pathways. We decided to experimentally follow-up on the pathway "IL-6-mediated signaling events" for several reasons: (i) IL-6 signaling was previously found to be relevant for stemness maintenance, that is, mammosphere formation of ductal breast carcinoma and normal mammary gland;[30] (ii) the TCPTP pathway, a negative regulator of IL-6 signaling[31], was also enriched; and (iii) assessment of individual genes expressed in these

pathways (Supplementary Fig. 1d) revealed a substantial overlap of the four pathways (Fig. 3e and Supplementary Table 5), indicating that related signaling modules had been triggered. We also tested the expression of the extracellular signal receptors and found that neither the receptor VEGFR3 nor Tie2 was expressed by DCCs. In contrast, while the mRNA of *IL6RA* (the IL-6 binding receptor unit) was also absent in DCCs, the IL-6 signal-transducing unit gp130 was expressed (*IL6ST*; Fig. 3d), indicating the amenability of DCCs to solely IL-6 trans-signaling (see below) and thereby to microenvironmental control. Given the hints for a role of IL-6 signaling for stemness maintenance and the restricted expression of IL-6 signaling molecules, we decided to explore the activation of the IL-6 pathway in normal and premalignant mammary cells in detail.

**IL-6 trans-signaling activates sphere-forming ability**. The IL-6 pathway can be activated directly or in trans. Direct or classical IL-6 signaling involves IL-6 binding to the heterodimeric receptor consisting of the ubiquitously expressed signal-transducing receptor subunit gp130 (gp130) and the membrane-bound IL-6 receptor alpha chain CD126[32]. In contrast, trans-signaling does not involve the membranous IL-6R alpha chain (mIL-6RA), but binding of IL-6 to the soluble IL-6RA (sIL-6RA) prior to binding to gp130 on the cell surface. sIL-6RA can be generated by alternative splicing or limited proteolysis of the membrane-bound

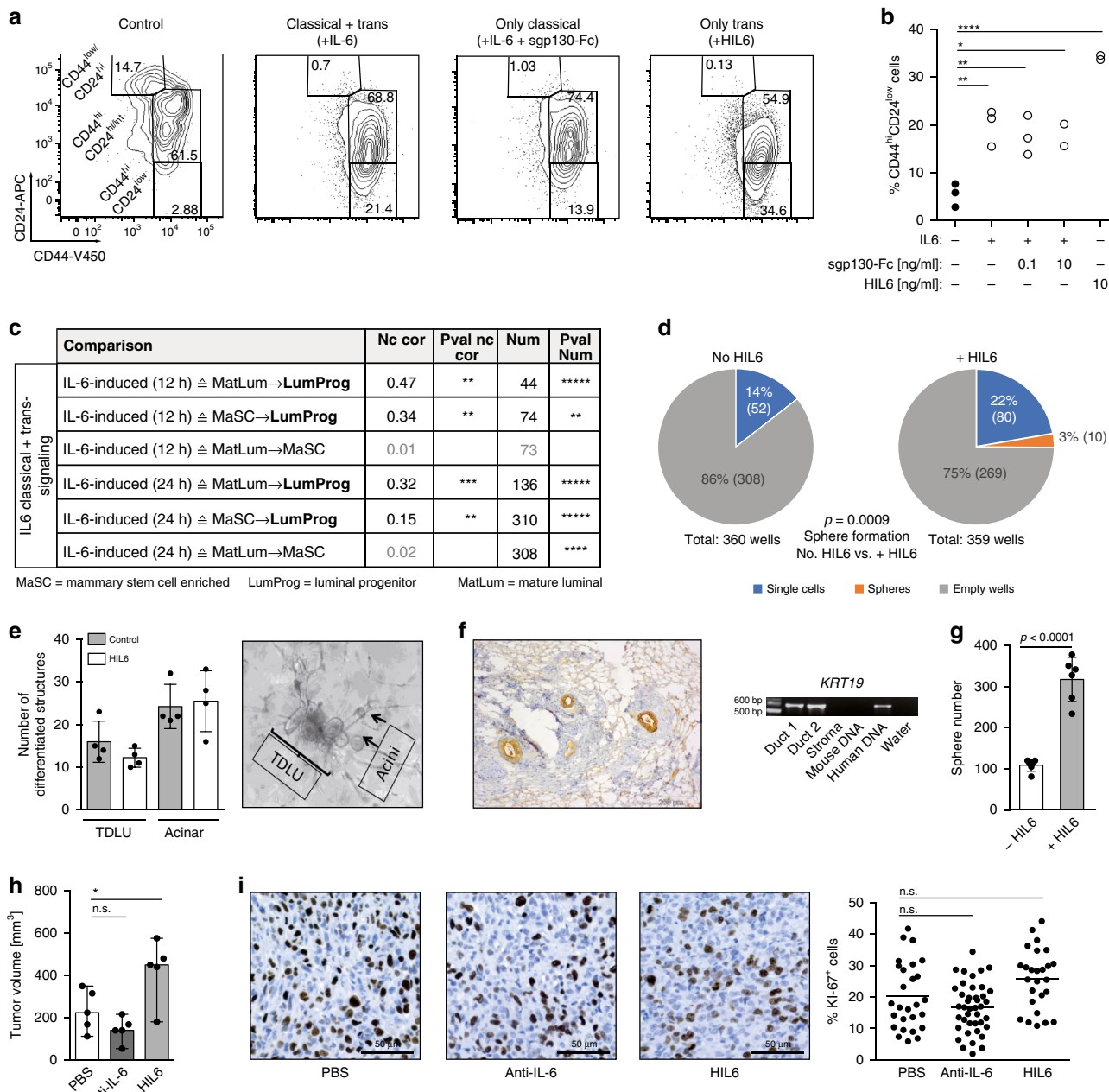

| | Comparison | Nc cor | Pval nc cor | Num | Pval Num |
|---|---|---|---|---|---|
| IL6 classical + trans-signaling | IL-6-induced (12 h) ≙ MatLum→**LumProg** | 0.47 | ** | 44 | ***** |
| | IL-6-induced (12 h) ≙ MaSC→**LumProg** | 0.34 | ** | 74 | ** |
| | IL-6-induced (12 h) ≙ MatLum→MaSC | 0.01 | | 73 | |
| | IL-6-induced (24 h) ≙ MatLum→**LumProg** | 0.32 | *** | 136 | ***** |
| | IL-6-induced (24 h) ≙ MaSC→**LumProg** | 0.15 | ** | 310 | ***** |
| | IL-6-induced (24 h) ≙ MatLum→MaSC | 0.02 | | 308 | **** |

MaSC = mammary stem cell enriched    LumProg = luminal progenitor    MatLum = mature luminal

receptor and provided via autocrine and paracrine secretion. To explore the impact of the different modes of IL-6 signaling on stemness or early progenitor traits, we used the premalignant human mammary epithelial cell lines MCF-10A and hTERT-HME1 and primary HMECs as models for early, genetically immature DCCs. Since metastasis founder cells are thought to display stem-like features[33,34], cells were cultured under mammosphere conditions. The analysis of expression and secretion of IL-6 signaling molecules by MCF-10A and primary HMECs using enzyme-linked immunosorbent assay (ELISA), flow cytometry, and single-cell PCR of LRCs and nLRCs (Supplementary Fig. 2a–d) indicated that (i) mIL-6RA is expressed only in a fraction of LRCs and nLRCs; (ii) expression of IL-6 signaling molecules (IL-6RA, IL-6, gp130) does not significantly differ between LRCs and nLRCs, (iii) coexpression of all signaling molecules in individual cells is extremely rare, and (iv) sIL-6RA is generated by shedding of mIL-6RA and not by splicing.

Therefore, and in line with our results for breast cancer DCCs, IL-6 trans-signaling via binding of IL-6 to sIL-6RA and subsequent binding to gp130 is much more likely involved in pathway activation than classical IL-6 signaling. We therefore asked if stemness or early progenitor traits in mammary epithelial cells or DCCs are activated by a paracrine mode via classical signaling or trans-signaling. As a model for endogenous trans-signaling activation, we identified normal mammary cell-derived hTERT-HME1 cells with a knock-in of constitutively active epidermal growth factor receptor (EGFR) (hTERT-HME1-EGFR$^{\Delta746-750}$)[35]. This genetic change resulted in significantly increased amounts of IL-6 trans-signaling components in the culture supernatant (Supplementary Fig. 2e).

We then treated mammosphere cultures of MCF-10A, hTERT-HME1, hTERT-HME1-EGFR$^{\Delta746-750}$, and primary HMECs with activators or inhibitors of both pathways: (i) an anti-IL-6 antibody to inhibit IL-6 classical and trans-signaling, (ii) IL-6 to activate

**Fig. 5 IL6 trans-signaling endows non-stem cells with stem-like abilities. a, b** MCF 10A spheres were analyzed by flow cytometry for the expression of CD44 and CD24. The percentage of CD44hiCD24low expressing cells was determined. Data represent cumulative results of three independently performed experiments, each performed in triplicate. **c** Fold change correlation analysis comparing IL6-induced gene expression in MCF 10A cells at 12 and 24 h, respectively, with gene expression signatures of luminal progenitor (LumProg), mature luminal (MatLum), and mammary stem cell enriched cells (MaSC) according to the study of Lim et al.[29]. Nc cor: non-centered correlation between fold-changes, Num: number of common differentially expressed genes; **d** LRCs and nLRCs were sorted by flow cytometry from PKH26-labeled HMEC-spheres, nLRCs were plated as single cell per well ($n = 3$ patients) single-cell deposit determined by manual microscopic evaluation and cultured under mammosphere-conditions with or without HIL6. Sphere-formation and survival of single cells was determined after 14 days ($p$ values are provided within **d**). Each patient-culture was set-up as duplicate in either freshly prepared or conditioned mammosphere-medium. As no significantly different outcome (Fisher's exact test, $p = 0.6$ and $p = 1$ fresh vs. conditioned medium for cultures w/o HIL6 and with HIL6, respectively) was detected, results are presented as pooled analyses. **e** In vitro generation of acinar and tubular (TDLU) structures of primary HMECs cultured with or without HIL6 (each $n = 4$). **f** Primary HMECs cultured with HIL6 and transplanted into NSG-mice. Staining for human cytokeratin 8/18/19 of the transplanted area eight weeks post-transplantation. PCR specific for human KRT19 of two microdissected cytokeratin 8/18/19-positive ducts and one cytokeratin 8/18/19-negative stromal area. Pure mouse or human DNA was used as control. **g** MDA-MB-231 cells were cultured as spheres in the absence ($n = 6$) or presence of HIL6 ($n = 6$). **h** Tumor volume of 20,000 MDA-MB-231 cells pre-treated for 3 h with PBS, an anti-IL6 antibody or HIL6 and transplanted into NSG-mice ($n = 5$). All mice were analyzed at the same day after tumor cell inoculation; n.s. = non significant. **i** TissueFAX cytometric quantification of tumors from panel (**h**) for the percentage of Ki-67-positive cells. $n = 27$, 41, or 26 regions (0.25 mm$^2$ each) for PBS, anti-IL6 or HIL6. $P$ values in panel **b, h, i**: one-way ANOVA with Dunnett's multiple comparisons test (post hoc); **c** $P$ values according to two-sided Student's t-distribution for (transformed) Nc cor and hypergeometric testing for Num; d two-sided Fisher's exact test; g two-sided Student's t test; asterisks indicate significance between groups in multiple comparisons (*$p < 0.05$, **$p < 0.01$, ***$p < 0.001$, ****$p < 0.0001$, *****$p < 0.00001$). All error bars correspond to standard deviation (Mean ± SD). See Supplementary Table 1 for patient/sample-ID allocation. Source data for (**c**) are provided in the Supplementary Table 6.

classical and trans-signaling, and (iii) hyper-IL-6 (HIL6) to selectively activate trans-signaling. HIL6 is a fusion protein consisting of sIL-6RA, a linker chain, and IL-6 and is used as a molecular model of the IL-6/sIL-6RA complex[36,37]. Adding IL-6 or HIL6 to MCF-10A, hTERT-HME1 cells or HMEC cultures significantly increased sphere formation (Fig. 4a–c, Student's $t$ test $p < 0.01$ or one-way analysis of variance (ANOVA)/Dunnett's test $p < 0.0001$). Interestingly, primary HMECs responded only to HIL6, but not IL-6 (Fig. 4c), indicating that (i) the increase in sphere number was due to IL-6 trans-signaling, (ii) spheres originated from cells without mIL-6RA expression, and (iii) endogenous sIL-6RA is a limiting factor (Supplementary Fig. 2c). Of note, hTERT-HME1-EGFR$^{\Delta746-750}$ could only marginally be stimulated by the addition of HIL6, suggesting that it added little to the already available IL-6/sIL-6RA complexes (Fig. 4d).

To dissect the impact of classical and trans-signaling on the observed increase in sphere formation and hence the number of cells with stem-like activity, we specifically inhibited IL-6 trans-signaling, but not classical signaling by adding the soluble form of gp130 (sgp130-Fc) to IL-6-stimulated MCF-10A sphere cultures[38,39]. At both concentrations of sgp130-Fc tested, IL-6-induced sphere formation was abolished (Fig. 4e, one-way ANOVA/Dunnett's test $p < 0.0001$), demonstrating that cells devoid of mIL-6RA accounted for the increase in sphere numbers by acquiring or activating stem-like functions in response to IL-6 trans-signaling. Consistently, blocking of endogenous classical and trans-signaling in hTERT-HME1-EGFR$^{\Delta746-750}$ by anti-IL-6 did not reduce sphere formation to a greater extent than blocking IL-6 trans-signaling only (Fig. 4f, one-way ANOVA/Dunnett's test $p < 0.05$ and $p < 0.01$).

**IL-6 trans-signaling converts progenitor into stem-like cells.** We noted that IL-6- and HIL6-stimulated mammosphere cultures showed an increase in the relative abundance of CD44$^{high}$/CD24$^{low}$ cells (Fig. 5a, b), a phenotype that has been ascribed to neoplastic and nontumorigenic mammary cells enriched in tumor-initiating and sphere-forming cells, respectively[33,40]. Here, HIL6-stimulated cultures displayed the highest increase (Fig. 5a, b, one-way ANOVA/Dunnett's test ctrl. vs. IL-6$^{+/-}$ sgp130-Fc, $p < 0.01$; ctrl. vs. HIL6, $p < 0.0001$). The increase in CD44$^{high}$/CD24$^{low}$ cells was not the result of increased proliferation of any

CD24/CD44 subpopulation, but seemed to be caused by the conversion of non-stem-like CD44$^{high}$/CD24$^{high/intermediate}$ into CD44$^{high}$/CD24$^{low}$ stem-like cells (Supplementary Fig. 3a–c). To corroborate these findings, we compared IL-6/HIL6-induced differential gene expression in MCF-10A cells to differences in gene expression between mammary stem cell-enriched (MaSC), LumProg, and mature luminal (MatLum) cells as published by Lim et al.[41]. The overlap between the respective differentially expressed genes was highly significant in almost all comparisons (Fig. 5c, Supplementary Fig. 3d, and Supplementary Table 6) and the observed expression fold changes were consistent with the notion that IL-6/HIL6 stimulation recruits progenitor populations from both more differentiated and more stem-like populations, with the dedifferentiation branch (MatLum → LumProg) being more consistent than differentiation (MaSC → LumProg). It should be noted that these in vitro generated data are fully consistent with the strong expression of the LumProg marker *KIT* in DCCs (Fig. 3b and Supplementary Fig. 1c) that we found to be activated via IL-6 signaling.

To confirm these findings, we tested if ex vivo derived primary HMECs converted to stem-like cells by HIL6 activation. We isolated nLRCs from non-IL-6-stimulated HMEC mammospheres (Supplementary Fig. 3e) and replated them as single cell per well with or without HIL6. Whereas in the absence of HIL6 nLRCs were unable to form spheres, the proportion of sphere-forming cells induced from nLRCs in the presence of HIL6 was similar to that of non-HIL6-stimulated LRCs (3% vs. 5% sphere formation; Figs. 2b and 5d). Moreover, replacing HIL6 by IL-6 in the first or second week of a 2-week mammosphere assay showed that continuous IL-6 trans-signaling is needed to induce and maintain the number of cells with stem-like activity (Supplementary Fig. 3f). Finally, we confirmed that primary HIL6-treated HMEC spheres retain their ability to form acinar and tubular structures in vitro (Fig. 5e) and mammary ducts in immunodeficient NSG mice (Fig. 5f). As MCF-10A cells do not form tumors in immunodeficient mice, we selected the LumProg-derived MDA-MB-231 cells to test whether the HIL6-induced increase in sphere formation translates into higher malignancy in vivo. Like MCF-10A cells, MDA-MB-231 cells show increased sphere formation in response to HIL6 (Fig. 5g). Upon xenotransplantation of an equal number of MDA-MB-231 cells pretreated with phosphate-buffered saline

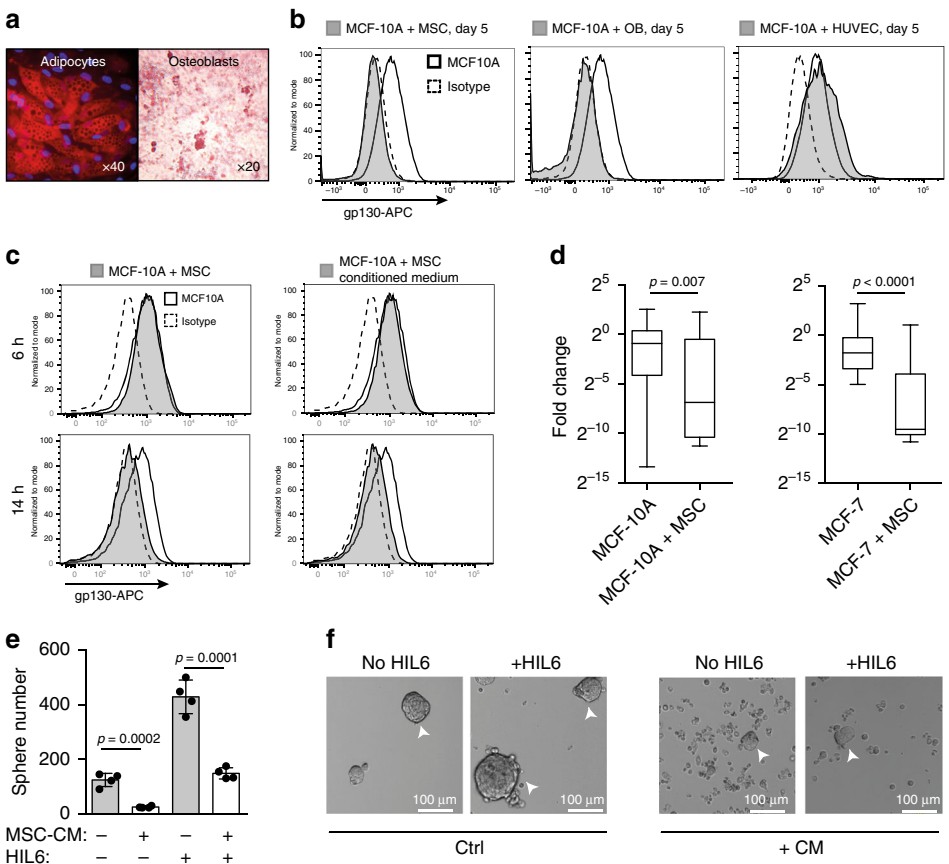

**Fig. 6 gp130 downregulation by soluble factors of bone marrow stromal cells. a** Mesenchymal stem cells were tested for their ability to differentiate in vitro into adipocytes (Nil oil red O staining) and osteoblasts (Alizarin red S staining). **b** Surface gp130 expression of MCF 10A after five days of co-culture with primary mesenchymal stem cells (MSCs) from a breast cancer patient, primary osteoblasts (OBs) derived thereof or primary human umbilical vein endothelial cells (HUVECs). **c** Surface gp130 expression of MCF 10A after 6 and 14 h of co-culture with MSCs or MSC-conditioned medium. **b**, **c** Grey filled histograms indicate MCF 10A co-cultured with MSCs, OBs, HUVEC or MSC-conditioned medium. Histograms with a thick black line indicate MCF 10A cells cultured alone and dashed histograms isotype control staining for gp130. **d** gp130 gene expression levels determined by single cell qPCR of MCF 10A cultured for 5 days with (n = 23) or without (n = 37) MSCs and MCF- 7 cultured for 5 days with (n = 20) or without (n = 20) MSCs. Box and whisker plots show the MCF 10A (MCF-7) fold change cultured with MSCs relative to MCF 10A (MCF-7) cultured without MSCs with boxes marking the median, lower-quartile, and upper-quartile, and lines connecting the extremes. **e**, **f** MCF 10A cells were left untreated or pre-treated for 14 h with MSC-conditioned medium, washed and then tested for their ability to form spheres in the presence of endogenously produced IL6/sIL6RA or exogenously added HIL6. Sphere-formation was assessed after seven days, n = 4 for each group. P values according to two-sided Mann–Whitney test (**d**) or two-sided Student's t-test (**e**). All error bars correspond to standard deviation (Mean ± SD).

(PBS), anti-IL-6 or HIL6 for 3 h, tumors in the HIL6-group were significantly larger than in the control group (Fig. 5h). This was not caused by increased proliferation or decreased apoptosis as the percentage of Ki-67+ tumor cells did not differ significantly between the groups (Fig. 5i), and caspase-3+ cells were not detected in any of the tumors. While this is consistent with an elevated stemness of MDA-MB-231 cells, limiting dilution experiments, ideally performed with additional cell lines, would be needed to fully establish the stemness-conferring activity of IL-6 trans-signaling to human cell line models in xenotransplantation.

**BM niche cells regulate responsiveness to IL-6 trans-signaling.** As gp130 expression is essential for IL-6 signaling, we tested whether BM stromal cells modulate the ability of mammary epithelial cells to receive IL-6 signals. We isolated primary human mesenchymal stem cells (MSCs) from diagnostic BM aspirates of nonmetastasized breast cancer patients or healthy volunteers and confirmed their ability to differentiate into adipocytes and osteoblasts (OBs) in vitro (Fig. 6a and Supplementary Fig. 4a). We then cocultured MCF-10A cells with (i) MSCs, (ii) in vitro

differentiated OBs, or (iii) human umbilical vein endothelial cells (E4ORF1 HUVECs[42]) under nonsphere conditions. Interestingly, flow cytometric analysis revealed cell surface downregulation of gp130 on MCF-10A cells cocultured with MSCs and OBs, but not with HUVECs (Fig. 6b). Separation of MCF-10A and MSCs by a transwell or using MSC-conditioned medium (CM) showed gp130 cell surface downregulation to be independent of cell–cell contact (Supplementary Fig. 4b). Moreover, downregulation was not immediate but observed between 6 and 14 h after initiation of the coculture with MSCs, OBs, or MSC-CM from healthy donors (HDs) or breast cancer patients (Fig. 6c and Supplementary Fig. 4c). This kinetic is consistent with the known independency of gp130 internalization from ligand binding[43] and points towards a transcriptional regulation of gp130 surface expression. To test this, we determined gp130 gene expression levels in single cells isolated from MCF-10A/MSC and MCF-7/MSC cocultures. Interestingly, both cell lines decreased their gp130 gene expression in response to MSCs (Fig. 6d), which is consistent with transcriptional regulation and demonstrates that early DCCs as well as more advanced cancer cells can respond to signals from neighboring cells.

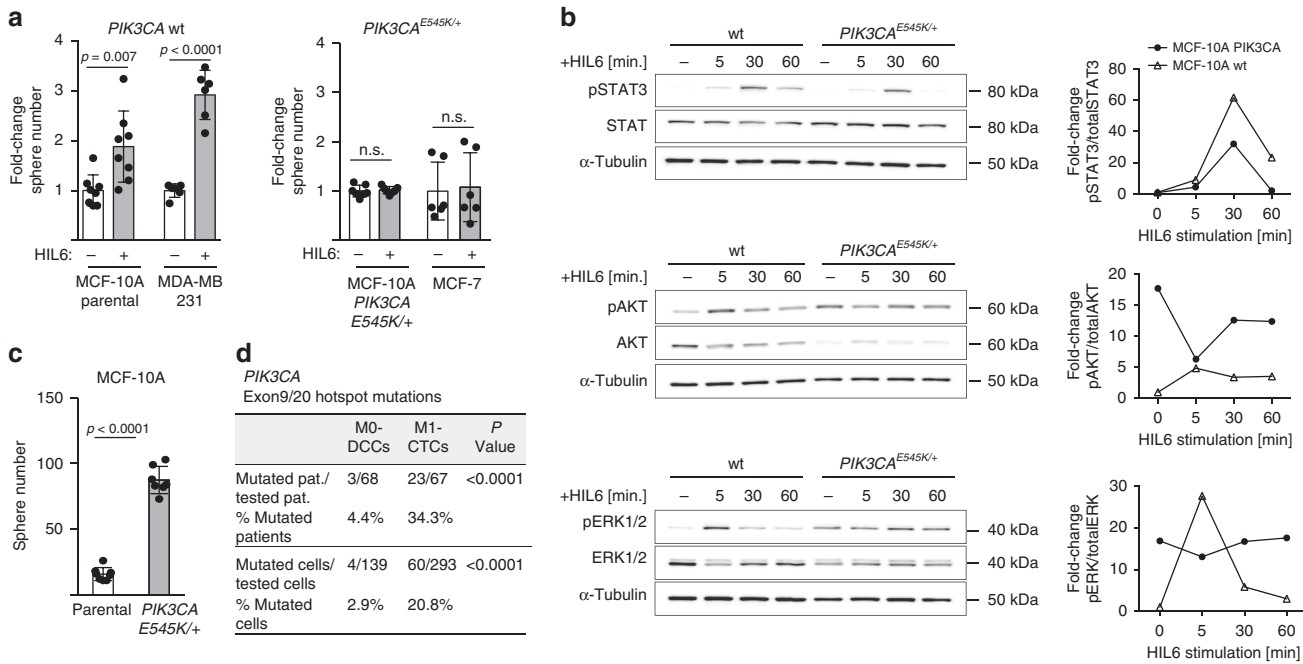

**Fig. 7 PIK3CA pathway activation confers independency from IL6 signaling. a** Fold change in sphere numbers of pre-malignant (MCF 10A) and tumorigenic cell lines (MCF-7, MDA-MB-231) without (MCF 10A parental, n = 8; MDA-MB-231, n = 6) or with mutational activation of PIK3CA (MCF 10A PIK3CAE545K/+, n = 7; MCF-7, n = 6) cultured in the presence or absence of HIL6. Note that MCF 10A PIK3CAE545K/+ cells are isogenic to MCF 10A parental; n.s. = non-significant. **b** Western blot analyses showing phosphorylation of STAT3Tyr705, AKTSer475 and ERK1/2Thr202/Tyr204 in MCF 10A or MCF 10A PIK3CAE545K/+ cells cultured without or with HIL6 for the indicated time. For quantification, the signal of the phosphorylated protein and total protein was normalized to α-tubulin, then the ratio of phosphorylated to total protein was calculated. The graphs show the fold change in signal ratio over time relative to the control (unstimulated MCF 10A wt = 1). **c** Sphere numbers of the isogenic cells MCF 10A parental (n = 8) and MCF 10A PIK3CAE545K/+ (n = 7) cultured in the absence of HIL6. **d** Cytokeratin 8/18/19+ DCCs from BM of non-metastasized (M0-stage) HR-positive breast cancer patients and CD45-/EpCAM+/cytokeratin 8/18/19+ CTCs isolated from peripheral blood of metastasized (M1-stage) HR-positive breast cancer patients were sequenced for hotspot-mutations in PIK3CA (Exon 9: E545K, E542K; Exon 20: H1047R, H1047L, M1043I). P values in **a**, **c** two-sided Student's test; **d** two-sided Fisher's exact test. All error bars correspond to standard deviation (Mean ± SD).

To explore the functional impact of gp130 downregulation induced by MSCs, we tested MCF-10A cells pretreated for 14 h with MSC-CM for their sphere-forming ability. Pretreated MCF-10A showed a significant decrease in sphere number and an increase in single, nonsphere-forming cells in the presence of both endogenously produced IL-6/sIL-6RA and exogenously added HIL6 (Fig. 6e, f, Student's t test, both p < 0.001). The data indicate that the microenvironment in which early DCCs reside determines (i) their responsiveness to IL-6 trans-signaling, with stromal and osteoblastic niches disabling IL-6 trans-signaling in DCCs, and as a consequence, (ii) the number of DCCs with stem-like phenotype and function, that is, metastasis-initiating ability.

**Fully malignant DCCs escape IL-6 trans-signaling dependence by oncogenic pathway activation**. Cancer progression is driven by genetic and epigenetic evolution overriding microenvironmental control mechanisms. Although consensus about the nature of the metastatic niche is still lacking, MSC/OB-rich endosteal or vascular niches are believed to regulate the fate of DCCs[42,44]. Our experiments indicate that endosteal niches, although being rich in IL-6 and sIL-6RA molecules[45–48], render DCCs unresponsive to IL-6 trans-signals. However, gp130+ DCCs in vascular niches would respond to IL-6/sIL-6RA complexes. Therefore, in both niches pathway activation by mutation would provide a selection advantage for DCCs that otherwise might depend on microenvironmental signals. We consequently sought for corroborating evidence that genetically variant DCCs may evade the need for IL-6 trans-signaling and become selected. We

considered the phosphatidylinositol-3-kinase (PI3K) pathway a strong candidate for such a selected oncogenically activated pathway[49] as (i) IL-6 signaling activates not only the Janus kinase/signal transducer and activator of transcription (JAK/STAT) pathway, but also the PI3K/AKT pathway[50], (ii) early DCCs expressed PIK3CA as core element of the four identified stemness-associated pathways (Fig. 3e), (iii) activating PIK3CA mutations in exons 9 and 20 are among the most frequent mutations occurring in human breast cancer, and (iv) constitutive activation of the PI3K pathway has been shown to evoke cell dedifferentiation of mammary gland cells into a multipotent stem-like state[51]. To test PI3K signaling, we analyzed sphere formation in response to HIL6 in premalignant MCF-10A and malignant breast cancer cell lines with or without a PIK3CA-activating mutation. Interestingly, HIL6 increased sphere formation only in cells with wild-type PIK3CA (Fig. 7a), but not in cells with an activating PIK3CAE545K/+ mutation. Using isogenic MCF-10A cells with (knock-in of PIK3CAE545K/+) and without PIK3CA mutation, we noted that HIL6 activated the STAT3 pathway in mutant and wild-type cells (Fig. 7b). In contrast, IL-6 trans-signaling increased the activation level of the ERK and AKT pathway only in wild-type cells as these pathways were already highly activated in PIK3CA-mutant cells (Fig. 7b). Consistently, untreated PIK3CA-mutant cells formed significantly more spheres than wild-type PIK3CA cells (Fig. 7c, Student's t test, p < 0.0001). MSC-induced gp130 downregulation was unaffected by the PIK3CA mutational status (Supplementary Fig. 4d). In summary, these data show that PIK3CA activation overrides

regulation of stemness traits by IL-6 trans-signaling and renders cancer cells more independent from microenvironment control.

Based on these findings, we tested for direct evidence of genetic selection in the PIK3CA pathway during cancer progression by analyzing the PIK3CA gene for mutations in exons 9 and 20 in BM-derived DCCs from nonmetastasized (M0-stage) and in circulating tumor cells (CTCs) from metastasized (M1-stage) breast cancer patients mostly displaying bone metastases. Both groups of CK[+] cancer cells had previously been shown to display CNAs[5,52]. As HR[+] breast cancer is the most frequent breast cancer type, we focused on this disease type to obtain meaningful sample numbers. Consistent with previous data, PIK3CA hotspot mutations were found in DCCs/CTCs of patients with HR[+] tumors (Fig. 7d). Strikingly, only 4.4% of M0-stage patients harbored PIK3CA mutations, whereas manifest HR[+] breast cancer patients with metastasis displayed them in 34.3% of cases (Fisher's exact test, $p < 0.0001$; Fig. 7d). These data are fully consistent with a scenario in which early DCCs depend on IL-6 trans-signaling and become increasingly independent thereof during cancer evolution.

## Discussion

In this study, we provide evidence for a role of the niche microenvironment to enable and drive the earliest stages of human metastasis formation. We identified IL-6 trans-signaling as an activator of stem-like and progenitor traits underlying epithelial colony formation, a mechanism that is characterized by dependence on both IL-6 and sIL-6RA, in contrast to IL-6 alone, when a cell is equipped with mIL-6RA. Our finding questions the concept of fully malignant and autonomous cancer cells as founders of metastasis. However, during subsequent malignant evolution, DCCs may evade microenvironmental control by acquiring IL-6 independence. Our data indicate that this could occur via mutational activation of the PI3K pathway.

Several observations characterize early metastatic BM colonization. First, breast cancer dissemination in humans often starts from lesions often measuring <1–4 mm in diameter[2,5]. Second, initially DCCs often do not display the typical karyotypic changes of breast cancer tumors or metastases, but are detected long before manifestation of clinical metastasis[5,8,15]. Third, metastasis formation in BM usually takes years to decades[53], indicating that evolutionary mechanisms and slow growth kinetics dictate the pace. This framework precludes the use of transgenic mouse models that are too short-lived to mimic the process in patients and rarely form bone metastases. The same applies to in vitro and in vivo studies involving commonly used and genomically highly aberrant breast cancer cell lines derived from manifest metastases or primary tumors. All of these do not represent the biology under investigation.

We therefore aimed at expanding early DCCs in immune-deficient mice, but failed. Reasons for this can be manifold: DCCs are extremely rare and we estimate that often <10 cells/mouse were injected. Also, species barriers may preclude engraftment of cancer cells that may critically depend on certain microenvironmental signals, among them the species-specific IL-6 signaling[54]. Our failure is also consistent with data from melanoma where we had identified that engraftment of DCCs requires activation of specific oncogenic pathways[55]—which in case of PIK3CA mutations early (M0-stage) breast cancer DCCs commonly lack. Consistently, M1-stage DCCs formed xenografts in two out of four cases. We therefore based our approach on two starting points: first, we analyzed transcriptomic data of early human breast cancer DCCs to identify candidate pathways, and second, we used genomically close-to-normal and normal mammary epithelial cells as models to functionally interrogate the transcriptome data.

We focused our analysis on EpCAM[+] cells from BM. We followed the rationale that cells with an epithelial phenotype and concomitant genetic alterations that are isolated from mesenchymal BM of a patient with known carcinoma are bona fide cancer cells. We are aware that this does not provide a formal proof of their origin from the individual cancer. However, given the enormous intratumoral heterogeneity not only of the cancer at diagnosis, but also of its complete evolutionary history, such a formal proof may be difficult to establish. Currently, EpCAM is the best marker to identify and isolate viable epithelial cells from BM. However, it is neither fully specific[14] as cells of the B cell lineage may express EpCAM[23] nor does it identify all breast cancer-initiating cells[56]. However, EpCAM[+] mammary progenitor cells give rise to chromosomally unstable cancers, such as triple-negative basal-like/postepithelial mesenchymal transition and HR[+] cancers[57]. Since we used chromosomal instability as an inclusion criterion to identify DCCs, our findings do not apply to claudin-low cancers, which are derived from EpCAM[−], chromosomally stable cells. Interestingly, early M0-stage DCCs isolated from BM highly expressed KIT, which is characteristic for LumProg cells. LumProg cells are the last common precursor of triple-negative, basal-like and HR[+] cancers[58]. Since HR[+] breast cancers can experimentally be normalized by functionally impairing their LumProg cells[59], cells with this phenotype are likely to comprise metastasis founder cells of EpCAM[+] triple-negative and HR[+] cancers.

This reasoning is fully consistent with our observed effects of IL-6 on normal mammary stem and progenitor cells. We identified an activated IL-6 pathway in DCCs and our in vitro models revealed that IL-6 trans-signaling, but not classical signaling, induces dedifferentiation of mammary epithelial cells and endows them with stemness and progenitor traits. IL-6 trans-signaling makes cells dependent on the microenvironment; however, BM represents an IL-6-rich environment[47]. There, IL-6/sIL-6RA is important for the regulation of hematopoiesis[60] and for the generation of hematopoiesis-supporting BM stromal cells[61] using autocrine/paracrine feedback loops[48]. In the context of breast cancer, it is noteworthy that serum levels of IL-6 and sIL-6R and their local production in BM by OBs depend on sex steroids, which change with the menstrual cycle, and are negatively regulated by estrogen and hormone replacement therapy and increase up to tenfold during menopause[48,62,63]. Therefore, systemic microenvironmental changes may provide a mechanism by which DCCs become activated in postmenopausal breast cancer patients. Consistently, bone-only metastasis is significantly associated with higher age at primary diagnosis in HR[+] breast cancer patients[64], known for very late relapses.

We have shown that IL-6 trans-signaling equips normal and transformed mammary cells with stemness and progenitor traits likely to be crucial for DCCs. Although ~6000–27,000 DCCs may lodge in the BM compartment[65], only a few of them will progress to manifest metastasis. Our data are in line with this observation as coculture with BM niche cells like OBs or stromal MSCs resulted in the loss of gp130 expression, rendering only DCCs in vascular niches possibly responsive to IL-6 trans-signaling. Moreover, only ~3–5% of mammary cells acquired mammo-sphere formation ability upon HIL6 stimulation. Genetic analysis of DCCs revealed that DCCs from HR[+] breast cancer progressing to manifest metastasis often acquired PIK3CA mutations, possibly because the activated PI3K/AKT pathway rendered mammary epithelial cells independent of IL-6 trans-signaling.

We are aware of data indicating that some metastases may be derived from late-disseminating cells[66,67]. One study showed that in three out of six cases tested CK[+] cells displayed alteration profiles very similar to their matched primary tumors[66]. Future studies will need to carefully address the role of tumor (sub)type,

disease stage and duration, growth kinetics, and a possible interdependence of early and late DCCs for metastasis formation. However, the findings presented here demonstrate that early DCCs are able to interpret signals from the BM environment that activate them and might drive their progression. Thereby, our study may lay the groundwork for novel adjuvant therapies. Early DCCs striving to form metastasis during yearlong evolutionary processes may be sensitive to different drugs than fully malignant cancer cells. Their Achilles' heel may consist of microenvironmental signals supporting their survival and genomic progression. Since there is hope that oncogene- or tumor-suppressor-gene-associated drug resistance mechanisms are not yet operative, depriving early DCCs of microenvironmental support pathways could render them vulnerable to nongenotoxic drugs.

## Methods

**Patient material.** Human noncancerous mammary tissue was obtained from female patients undergoing reduction mammoplasty surgeries at the University Center of Plastic-, Aesthetic, Hand-, and Reconstructive Surgery, University of Regensburg, Germany after informed, written consent of patients was obtained (ethics vote number 07/043, ethics committee of the University Regensburg). After verification of the noncancerous origin of the tissue by a pathologist, mammary glands were dissociated and primary HMECs isolated.

Human disseminated cancer cells were obtained from BM aspirates of breast or prostate cancer patients without and with distant metastases. EpCAM[+] cells were obtained from BM of patients without known malignant disease undergoing hip replacement surgery (HD, healthy donor). Human MSCs were obtained from BM aspirates of breast cancer patients or HDs. Written informed consent of cancer and control patients was obtained and the ethics committee of the University of Regensburg (ethics vote number 07/79) approved BM sampling and analysis of isolated cells.

Enrichment and detection of CTCs was performed within the SUCCESS (EUDRA-CT number 2005-000490-21) and DETECT (EUDRA-CT number 2010-024238-46)[68] studies using the CellSearch® system[69]. Written informed consent for CTC analysis and characterization was obtained for all patients included. All experiments conformed to the principles set out in the WMA Declaration of Helsinki and were approved by the ethical committees responsible for the corresponding studies (Universities of Munich, Dusseldorf, Tuebingen, and Ulm). Isolation and molecular analysis of CTCs was approved by the ethics committee of Regensburg (ethics vote number 07/079).

**Mice.** NOD.Cg-Prkdc[scid] IL2rg[tmWjl]/Sz (also termed NSG) or NOD.Cg-Prkdc[scid] mice were purchased from the Jackson Laboratory, USA and maintained under specific pathogen-free conditions, with acidified water and food ad libitum in the research animal facilities of the University of Regensburg, Germany. All approved experimental animal procedures were conducted according to German federal and state government of Upper Palatinate, 54-2531.1-10/07, 54-2532.1-34/11; 54-2532.1-17/11, 54-2532.1-1/12, 54-2532.4-7/12).

**Cell lines.** MCF-7 and MDA-MB-231 breast cancer cell lines were obtained from the German Collection of Microorganisms and Cell Cultures (DSMZ) and Cell Lines Service (CLS), respectively. MCF-10A (CRL-10317), a nontumorigenic mammary epithelial cell line, was obtained from the American Type Culture Collection[70]. The isogenic cell line MCF-10A PIK3CA[E545K/+] (HD 101-002) together with its parental cell line MCF-10A-parental (HD PAR-003) were purchased from Horizon Discovery, United Kingdom. MCF-10A-GFP cells were generated by transducing MCF-10A cells with pRRL.sin.cPPT.hCMV-GFP.WPRE (generously provided by Luigi Naldini, Italy). hTERT-HME1-derived cell lines, E4ORF1-transduced primary HUVECs, and murine embryonic fibroblasts C3H10T1/2 were generously provided by Alberto Bardelli, (University of Turin, Italy), Cyrus Ghajar (Fred Hutchinson Cancer Research Center, USA), and Max Wicha (University of Michigan, USA), respectively. The identity of all cell lines was confirmed by DNA fingerprinting analysis utilizing the GenePrint 10 System (Promega).

All MCF-10A-derived cell lines were cultured in Ham's Dulbecco's modified Eagle's/F12 (DMEM/F12) medium (Pan-Biotech, Germany) supplemented with 5% horse serum (Sigma-Aldrich, Germany), 2 mM L-glutamine (Pan-Biotech, Germany), 1% penicillin/streptomycin (Pan-Biotech, Germany), 20 ng/ml EGF (Sigma-Aldrich, Germany), 0.5 µg/ml hydrocortisone (Sigma-Aldrich, Germany), 10 µg/ml insulin (Sigma-Aldrich, Germany), and 0.1 µg/ml cholera toxin (Sigma-Aldrich, Germany). All hTERT-HME1-derived cells lines were maintained in DMEM/F12 medium (Pan-Biotech, Germany) supplemented with 10% fetal calf serum (FCS) (Sigma-Aldrich, Germany), 2 mM L-glutamine (Pan-Biotech, Germany), 1% penicillin/streptomycin (Pan-Biotech, Germany), 20 ng/ml EGF (Sigma-Aldrich, Germany), 0.5 µg/ml hydrocortisone (Sigma-Aldrich, Germany), and 10 µg/ml insulin (Sigma-Aldrich, Germany). MDA-MB-231 cells were cultured

in DMEM medium (Pan-Biotech, Germany) supplemented with 10% FCS (Sigma-Aldrich, Germany), 2 mM L-glutamine (Pan-Biotech, Germany), and 1% penicillin/streptomycin (Pan-Biotech, Germany). MCF-7 cells were propagated in RPMI-1640 medium (Pan-Biotech, Germany) supplemented with 10% FCS, 2 mM L-glutamine, and 1% penicillin/streptomycin. Murine embryonic fibroblasts C3H10T1/2 were grown in DMEM (Pan-Biotech, Germany) medium supplemented with 5% FCS (Pan-Biotech, Germany), 2 mM L-glutamine (Pan-Biotech, Germany), and 1% penicillin/streptomycin (Pan-Biotech, Germany). E4ORF1-transduced primary HUVECs were cultured using the EGM-2 Bullet Kit (Lonza, Germany). All cell lines were kept at 37 °C and 5% CO$_2$ in a fully humidified incubator and negatively tested for mycoplasma by PCR.

**Isolation of disseminated cancer cells from BM.** Mononuclear cells from BM of nonmetastasized breast cancer patients were plated on adhesive slides (Thermo Fisher) at a density of 0.5–1 × 10$^6$ cells/slide. Slides were stored at −20 °C. From each patient, 1–2 10$^6$ BM cells were stained with the monoclonal antibody A45-B/B3 (AS Diagnostik, Germany) against CK8/18/19 and developed with the anti-mouse AB-Polymer (Zytomed Systems, Germany). Unspecific binding was blocked using PBS/10% AB serum (Bio-Rad, Germany). Alkaline phosphatase was developed with 5-bromo-4-chloro-3-indolyl phosphate and nitroblue tetrazolium (Bio-Rad, Germany) as substrate. Slides were covered with PBS under a cover glass and assessed by bright-field microscopy. An identical number of cells served as a control for staining with mouse IgG1 kappa (MOPC-21) without known binding specificity. After removal of the cover glass, positive cells were isolated from the slide with a micromanipulator (Eppendorf PatchMan NP2) and subjected to whole-genome amplification (WGA) for subsequent PIK3CA mutation analysis.

For isolation of both RNA and DNA from the same disseminated cancer cells, mononuclear cells from BM of nonmetastasized breast cancer patients were subjected to immunofluorescent staining for EpCAM (Ber-EP4-FITC, Agilent or HEA-125-PE, Miltenyi Biotec, Germany). Positive cells were isolated with a micromanipulator (Eppendorf PatchMan NP2, Eppendorf, Germany) and cells were subjected to WTA to isolate RNA for subsequent PCR analyses, transcriptome microarrays or RNA-Seq, and WGA for isolation of genomic DNA for subsequent analysis of CNAs.

**Isolation of CTCs.** Up to three 7.5 ml blood samples per patient were collected into CellSave® tubes (Menarini Silicon Biosystems, Italy). The CellSearch® Epithelial Cell Test (Menarini Silicon Biosystems, Italy) was applied for CTC enrichment and enumeration according to the manufacturer's instruction. Samples from the SUCCESS study were prepared using a slightly modified protocol, pooling three separate CellSave® tubes (30 ml) as described elsewhere[52]. CTC[+] cartridges were sent from clinical centers to the Chair of Experimental Medicine and Therapy Research, Regensburg for cell isolation and molecular analysis. Cells were extracted from CellSearch® cartridges and isolated using the DEPArray™ system (Menarini Silicon Biosystems, Italy) and single-cell DNA was amplified by WGA for subsequent PIK3CA mutation analysis.

**Isolation of human primary mammary epithelial cells.** Primary human non-cancerous mammary tissue was mechanically dissociated using scalpels and then subjected to enzymatic digestion overnight at 37 °C in DMEM/F12 (Pan-Biotech, Germany) supplemented with 10 mM HEPES (Sigma-Aldrich, Germany), 2% bovine serum albumin (Sigma-Aldrich, Germany), 5 µg/ml insulin, 0.5 µg/ml hydrocortisone, 10 ng/ml cholera toxin (Sigma-Aldrich, Germany), 300 U/ml collagenase, and 100 U/ml hyaluronidase (all from Sigma-Aldrich, Germany). After removal of organoids and adipocytes by centrifugation at 210 × g for 2 min, the cell suspension was passed over a 100 and 40 µm cell strainer to obtain a single-cell suspension. Separation of fibroblasts from epithelial cells was accomplished by centrifugation at 350 × g for 4 min and epithelial cells from the cell pellet were cultured as mammospheres.

**Mammosphere culture.** Cell lines and primary HMECs were seeded at a density of 10,000 cells/ml and 50,000 cells/ml, respectively, in 3-, 6-, or 10-cm cell culture dishes or 96-well flat-bottom plates (Thermo Fisher Scientific, Germany; Sigma-Aldrich, Germany; TPP AG, Switzerland). For analyses using the Operetta high-content imaging system cells were plated with 10,000–50,000 cells/ml in 96-well µClear plates (Greiner Bio-One, Germany). To prevent attachment of cells all dishes/plates were coated with polyhydroxyethylmethacrylate (PolyHEMA) (12 mg/ml in 95% ethanol, Sigma-Aldrich, Germany) overnight. PolyHEMA-coated dishes/plates were ultraviolet sterilized for 30 min. Cells were cultured in mammosphere medium consisting of MEBM (mammary epithelial cell basal medium) (Lonza, Germany) supplemented with 1% penicillin/streptomycin (Sigma-Aldrich, Germany), 1×B27 (Life Technologies, Germany), 10 ng/ml EGF (Sigma-Aldrich, Germany), 10 ng/ml basic fibroblast growth factor (bFGF) (Sigma-Aldrich, Germany), 4 µg/ml heparin (Sigma-Aldrich, Germany), and 1% methylcellulose, if the Operetta CLS high-content imaging system was used. For some analyses, mammosphere media were supplemented additionally with 10 ng/ml IL-6 (Sigma-Aldrich, Germany), 1.5 µg/ml anti-IL-6 antibody (Sigma-Aldrich, Germany), 20 ng/ml HIL6, and 0.1 or 10 ng/ml recombinant human sgp130-Fc (R&D Systems,

Germany). Mammospheres were cultured in a humidified atmosphere with 5.5% $CO_2$ and 7% $O_2$ at 37 °C for 4 or 7 days.

For setting up of secondary mammosphere cultures, conducting flow cytometric or single-cell expression analyses, first-generation mammospheres were collected on day 7 by gentle centrifugation ($100 \times g$), dissociated into single-cell suspension with trypsin-EDTA (Pan-Biotech, Germany) for 3 min, followed by trypsin-neutralizing solution (Lonza, Germany). Single-cell suspensions of secondary mammospheres were obtained as described for day 7 first-generation mammospheres.

**Mammosphere counting.** The number of spheres with a diameter ≥50 µm was determined by manually counting of a complete plate/dish at day 7 using an inverted microscope (Olympus, ×10 air objective). Alternatively, spheres were stained and imaged using the Operetta CLS high-content imaging system (Perkin-Elmer, Hamburg, Germany) by adding CyTRAK Orange (BioStatus Ltd, UK) at day 4 to the wells at a final concentration of 10 µM. After 60 min incubation, fluorescence imaging of the plates was performed using a ×5 air objective and imaging of nine regions per well that were stitched to cover the entire well surface. Harmony high-content analysis software was used to analyze the images and to count the formation of spheres with a diameter ≥50 µm (Version 4.8; PerkinElmer, Hamburg, Germany).

**Isolation of LRCs, nLRCs, and QSCs from mammosphere cultures.** HMECs were labeled with the PKH26 Red Fluorescent Cell Linker Kit (Sigma-Aldrich, Germany) at 40 nM for 2 min at room temperature (RT). The reaction was stopped with 10% FBS containing medium, and cells were washed three times before plating into primary or secondary mammosphere cultures. LRCs and nLRCs were isolated from spheres of secondary mammosphere cultures that were dissociated with trypsin-EDTA (Pan-Biotech, Germany) for 3 min, neutralized with trypsin-neutralizing solution (Lonza, Germany), and stained with 4′,6-diamidino-2-phenylindole (DAPI) (Roche Diagnostics, Germany) for live/dead cell discrimination. Single LRCs and nLRCs were isolated as single PKH$^+$/DAPI$^-$ and PKH$^-$/DAPI$^-$ cells using a micromanipulator (Eppendorf PatchMan NP2, Eppendorf, Germany) or flow cytometric activated cell sorting. QSCs were isolated at day 7 from primary mammosphere cultures as single, DAPI$^-$/EpCAM$^+$/PKH$^+$ cells that did not form spheres and using the micromanipulator.

For flow cytometric assessment of the proliferation of MCF-10A-LRCs and MCF-10A-nLRCs, MCF-10A cells were labeled with carboxyfluorescein diacetate succinimidyl ester (ebioscience, Germany) at 2 µM for 10 min at 37 °C in PBS/1% FBS, washed twice after stopping of the reaction with 10% FBS containing medium, and cultured as mammospheres. At day 4, single-cell suspensions were obtained as described above and analyzed by flow cytometry.

**In vitro differentiation of HMECs.** Matrigel (growth factor reduced, without phenol red, BD Biosciences, Germany) was diluted 1:1 with a differentiation medium (Ham's DMEM/F12 medium (Pan-Biotech, Germany), 5% FCS (Pan-Biotech, Germany), 5 µg/ml insulin (Sigma-Aldrich, Germany), 1 µg/ml hydrocortisone (Sigma-Aldrich, Germany), 10 µg/ml cholera toxin (Sigma-Aldrich, Germany), 10 ng/ml EGF (Sigma-Aldrich, Germany), 1× penicillin/streptomycin/fungizone (Lonza, Germany)), smeared in 2-well slides and incubated for 15 min at 37 °C. On top, 50,000 cells from disaggregated secondary mammospheres were added. After incubation for 30 min at 37 °C, cells were covered with an additional Matrigel layer and incubated for additional 15 min at 37 °C. The differentiation medium was added at the end of the embedding procedure and exchanged every 2 days. Cultures were examined 3–4 weeks post embedding for the development of tubular and acinar structures.

**Culture of primary human MSCs and generation of OBs.** Mononuclear cells from BM aspirates were cultured at a density of $2 \times 10^6$ cells in a T75 flask (Sarstedt, Germany) in DMEM with 1 g/l glucose, 4 mM glutamine, and 1 mM sodium pyruvate (all from Life Technologies, Germany), supplemented with 10% MSC-qualified FBS (WKS Diagnostik, Germany), 1% penicillin/streptomycin (Life Technologies, Germany), and 1 ng/ml bFGF (Peprotech, Germany). Adherent cells were cultured for 3 weeks and cryo-conserved. Before cryo-conservation MSCs were tested for the expression of CD45, CD34, CD90, CD105, CD44, and Nestin by flow cytometry (Supplementary Fig. 4). In addition, the ability of MSCs to differentiate into adipocytes and OBs were tested[71]: OBs for coculture experiments were generated by culturing confluent MSC cultures in DMEM with high glucose (Life Technologies, Germany) supplemented with 10% MSC-qualified FBS (WKS Diagnostik, Frankfurt, Germany) 1% penicillin/streptomycin (Life Technologies, Germany), $10^{-7}$ M dexamethasone, 25 µg/ml L-ascorbic acid, and 3 mM sodium dihydrogen phosphate (all from Sigma-Aldrich, Germany) for 21 days with the medium being changed every day.

**Cocultures of MCF-10A with MSCs, OBs, and HUVECs.** MSCs and HUVECs were plated at a density of $4 \times 10^5$ cells/well of a 6-well plate (Corning, Germany) in their respective growth medium. The next day, the medium was exchanged to MCF-10A growth medium and $1 \times 10^5$ MCF-10A-GFP cells were added to each well. In case of cocultures with OBs, $4 \times 10^5$ MSCs per 9.6 cm$^2$ surface of a 6-well

plate (Corning, Germany) were plated and differentiated into OBs for 21 days. On day 22, the medium was exchanged to MCF-10A growth medium and $1 \times 10^5$ MCF-10A-GFP cells were added to each well. For cultures with transwells, MSCs were plated at $1.75 \times 10^5$ cells per 4.2cm$^2$ of a 6-well transwell insert (Falcon 353090, VWR, Germany).

**Xenotransplantations of DCCs, HMECs, and MDA-MB-231 cells.** For xenotransplantations of DCCs, mononuclear cells from BM aspirates of non-metastasized or metastasized breast or prostate cancer patients were enriched for human EpCAM or depleted of human CD45$^+$CD33$^+$CD11b$^+$ cells and erythrocytes using a mix of CD45, CD33, CD11b, and glycophorin A microbeads according to the manufacturer's instructions (Miltenyi Biotec, Germany). Each sample was then split into halves: one half was subjected to DCC enumeration by staining for CK8/18/19 or EpCAM. The other half of the cell suspension was transplanted without ex vivo expansion into NSG mice using one to two injection routes (nonmetastasized patients) or three to four different injection routes (metastasized patients). In some cases, mononuclear cells were cultured as mammospheres in 6-cm culture plates coated with PolyHEMA (12 mg/ml, Sigma), under hypoxic conditions (7% $O_2$) at 37 °C and in mammosphere medium containing 10 nM HEPES (Sigma-Aldrich, Germany), 10 µg/ml insulin (all from PAN-Biotech, Germany), 5 ng/ml growth-regulated oncogene-α (R&D Systems, Germany), 20 ng/ml HIL6 (kindly provided by S. Rose-John), and 0.2% methylcellulose (Sigma-Aldrich, Germany). Cultures were monitored weekly for sphere growth.

To transplant spheres or EpCAM-enriched or CD45/CD11b/erythrocyte-depleted BM, cells/spheres were collected in a microwell (volume 10–15 µl, Terasaki, Greiner Bio-One, Germany) precoated with PolyHEMA (12 mg/ml, Sigma-Aldrich, Germany). Cells or spheres were transplanted in a final volume of 30 µl and 25% high-concentration Matrigel (BD Biosciences, Germany) as published before[9]. Cells were injected with an insulin syringe (Microfine, 29G, U-50, BD Biosciences, Germany) subcutaneously, intravenously, intrafemorally, or subrenally in 4–8-week-old male or female NSG or NOD.Cg-Prkdc$^{scid}$ mice. Mammary fat pad injections were performed in the fourth precleared mammary fat pad of 3-week-old female mice in 50% Matrigel (BD Biosciences, Germany). Breast or prostate cancer origin of xenografts was verified by a pathologist.

To assess the differentiation ability of HMEC spheres in vivo, secondary mammospheres were dissociated and 200,000 cells were mixed with 225,000 preirradiated (15 Gy) C3H10T1/2 mouse fibroblasts. The cell suspension was then mixed 1:1 with Matrigel (growth factor reduced without phenol red, BD Bioscience, Germany) and injected in the fourth precleared mammary fat pad of 3-week-old female NSG mice. Mice were euthanized 8 weeks after transplantation and analyzed for the presence of human mammary gland tissue.

MDA-MB-231 cells grown under adherent conditions were pretreated with PBS, an anti-IL-6 antibody (1.5 µg/ml, Sigma-Aldrich, Germany) or HIL6 (20 ng/ml, a kind gift of Stefan Rose-John, Christian-Albrechts-University, Germany) for 3 h and 20,000 cells were transplanted into the mammary fat pad of NSG mice as 1:1 mixture with Matrigel (BD Biosciences, Germany) in the fourth precleared mammary fat pad of 3-week-old female NSG mice. All mice were analyzed when the first tumors reached a diameter of ~10 mm.

**Detection of human DCCs and mammary gland in NSG mice.** Lung and BM of mice transplanted with human DCCs were analyzed for the presence of DCCs or metastasis. Lungs were examined by a pathologist. For identification of disseminated cancer cells in the mouse BM, mononuclear cells were screened according to the method for human DCCs using immunofluorescent staining of the cell suspension with anti-human EpCAM (Ber-EP4-FITC, Agilent, or HEA-125-PE, Miltenyi Biotec Germany) or adhesive slides and staining anti-CK8/18/19 (A45-B/B3, AS Diagnostik, Germany). Unspecific binding was blocked by using PBS with 5% human AB serum (Bio-Rad, Germany) and 5% mouse serum (Agilent, Germany). Positive cells were isolated with the micromanipulator and subjected to WGA.

Mammary glands of mice transplanted with HMECs were dissected, fixed, and stained with an anti-human CK18 antibody (20 µg/ml, clone CK2, Millipore, Germany). Cells expressing human CK18 were laser microdissected (PALM Microbeam System, Bernried, Germany) and subjected to WGA.

The human origin of DCCs or CK18$^+$ cells isolated from mouse BM or laser microdissected from mammary glands of NSG mice was confirmed by a PCR discriminating between the human and mouse CK19 gene: forward primer, 5′-TTC ATG CTC AGC TGT GAC TG-3′ and reverse primer, 5′-GAA GAT CCG CGA CTG GTA C-3′, with an annealing temperature of 58 °C and an amplicon size of 621 bp for the human sequence.

**Quantification of HER2 and PGR staining in tissue sections by TissueFAX cytometry.** Tissue sections were stained with an automated staining machine (Ventana: BenchMark ULTRA). Tissue sections used for the analysis were stained within the same run. Images of stained tissue sections were scanned with the TissueFAXSi-plus imaging system (TissueGnostics, Vienna, Austria; acquisition software: TissueFAXS version 3.5.129) equipped with a digital Pixelink color camera (PCO AG, Kelheim, Germany). Images for the analysis of Ki-67 staining were analyzed with the HistoQuest software version 6.0.1.130 (TissueGnostics,

Vienna, Austria). Using that software, two markers were created: hematoxylin as "master marker" (nucleus) and Ki-67 as "nonmaster marker." To achieve optimal cell detection, the following parameters were adjusted: (i) nuclei size; (ii) discrimination by area; (iii) discrimination by gray; and (iv) background threshold. For the evaluation of the percentage of Ki-67-expressing cells, scatter plots were created, allowing the visualization of corresponding cells in the source region of interest using the real-time backgating feature. The cut-off discriminated between false events and specific signals according to cell size and intensity of Ki-67 staining. For the PBS group 112,098 cells (33 regions, 8.25 mm²), anti-IL-6 group 161,279 cells (41 regions, 10.25 mm²), and HIL6-group 98,812 cells (26 regions, 6.50 mm²) were analyzed.

**WGA and analysis of CNAs**. Single-cell genomic DNA was subjected to WGA using the previously described[72,73] or the commercially available version (Ampli1™ WGA, Menarini Silicon Biosystems). DCCs isolated from BM of patients or NSG mice were subjected to CNA analysis (mCGH)[72,73] or using the Ampli1™ LowPass Kit (Menarini Silicon Biosystems) according to the manufacturer's instructions.

**PIK3CA sequencing of single cells**. PIK3CA mutation in CTCs and DCCs was assessed using the Ampli1™ PIK3CA Seq Kit (Menarini Silicon Biosystems) or amplicon-based sequencing on single cells following Ampli1™ WGA. For amplicon-based sequencing, the following primers were used: exon 9 forward primer, 5′-AAG CAA TTT CTA CAC GAG A-3′ and reverse primer, 5′-CC TTA TTT ATT TCG TCT TAA ATG-3′, with an annealing temperature of 58 °C and an amplicon size of 189 bp; exon 20 forward primer, 5′-TCT AGC TAT TCG ACA GCA TGC-3′ and reverse primer, 5′-T ACC TAA CCT AGA AGG TGT GTT-3′, with an annealing temperature of 58 °C and an amplicon size of 221 bp. For each exon, 1 μl of WGA product of CTCs or DCCs was used for the PCR. Resulting products were loaded on a 1.5% agarose gel and negative PCR results were considered dropouts for PIK3CA analysis. Positive CTC samples were purified using the QIAquick Purification Kit (Qiagen, Germany) according to the manufacturer's protocol with the exception that elution at the end of the protocol was in 25 μl water. Purified CTC samples were sent to a sequencing provider (Sequiserve, Germany). PCR products from positive DCC samples were purified by the sequencing provider (GATC, Germany).

**WTA of single spheres and cells**. Single cells or spheres were placed in 5 μl lysis buffer (Active Motif, Belgium) supplemented with 1 μg protease (Active Motif, Belgium), 1 μl biotinylated oligo-dT peptide nucleic acids (PNAs) (Midi-Kit, Active Motif, dissolved in 400 μl water), and 10 ng tRNA (Roche, Germany). The proteolytic digest was performed at 45 °C for 10 min, followed by 1 min at 75 °C and 15 min at 22 °C for PNA annealing. mRNA was isolated with 4 μl streptavidin beads (Active Motif, Belgium) during 45 min rotation at room temperature. Ten microliters of cDNA wash buffer 1 (50 mM Tris-HCl, pH 8.3, 75 mM KCl, 3 mM MgCl₂, 10 mM dithiothreitol (DTT), and 0.25% Igepal) were added and the tubes placed into a magnetic rack. The supernatant containing the genomic DNA was collected and the beads were washed using 20 μl of cDNA wash buffer 2 (50 mM Tris-HCl, pH 8.3, 75 mM KCl, 3 mM MgCl₂, 10 mM DTT, 0.5% Tween-20). The supernatant was removed and the step repeated with 20 μl cDNA wash buffer 1. Reverse transcription was carried out under rotation for 45 min at 44 °C in a mix containing 0.5 mM dNTPs (VWR, Germany), 200 U Superscript II (Thermo Fisher Scientific, Germany), 30 μM CFL15CN8 primer (C₁₅GTCTAGAGN₈), 15 μM CFL15CT24 primer (C₁₅GTCTAGAT₂₄VN), 0.25% Igepal, 10 mM DTT (Invitrogen, Germany), and the buffer supplied by the manufacturer in a final volume of 20 μl. Primers were annealed at room temperature for 10–15 min before the addition of the enzyme. After completing the reverse transcription, beads were washed in 20 μl tailing wash buffer (50 mM KH₂PO₄, pH 7, 1 mM DTT, 0.25% Igepal) and resuspended in 10 μl tailing buffer (10 mM KH₂PO₄, pH 7, 4 mM MgCl₂, 0.1 mM DTT, 200 μM dGTP). After denaturation of the cDNA–mRNA hybrids at 94 °C for 4 min, 10 U terminal deoxynucleotide transferase (Affymetrix, Germany) were added and the samples were incubated at 37 °C for 60 min for the G-tailing reaction. After inactivation of the tailing enzyme (70 °C, 5 min), PCR mix I [4 μl buffer 1 (Expand Long Template, Roche, Germany), 3% deionized formamide] was added to each sample. A hotstart PCR was performed adding 5.5 μl PCR mix II [350 mM dNTPs, 1.2 μM CP₂ primer (TCAGAATTCATGC₁₅) and 5 U Pol Mix (Expand Long Template)]. Forty cycles were run: 20 cycles of 15 s at 94 °C, 30 s at 65 °C, 2 min at 68 °C, and 20 cycles with an elongation of the extension time of 10 s and a final elongation step of 7 min at 68 °C.

The quality of WTA products was assessed by expression analysis of three housekeeping genes: EE1A1 forward primer, 5′-CTG TGT CGG GGT TGT AGC CA-3′ and reverse primer, 5′-TGC CCC AGG ACA CAG AGA CT-3′, with an annealing temperature of 58 °C and an amplicon size of 290 bp; ACTB forward primer, 5′-GCG TGA CAT TAA GGA GAA GCT G-3′ and reverse primer, 5′-CGC TCA GGA GGA GCA ATG AT-3′, with an annealing temperature of 58 °C and an amplicon size of 378 bp; GAPDH forward primer, 5′-CCA TCT TCC AGG AGC GAG AT-3′ and reverse primer, 5′-CAG TGG GGA CAC GGA AGG-3′, with an annealing temperature of 58 °C and an amplicon size of 489 bp. Only samples positive for all three markers were used for downstream analyses.

**mRNA microarray experiments**. MCF-10A cells were cultured as mammospheres in the presence or absence of 10 ng/ml IL-6 (Sigma-Aldrich, Germany), 10 ng/ml IL-6 + 0.1 ng/ml recombinant human sgp130-Fc (R&D Systems, Germany) or 20 ng/ml HIL6 (a kind gift of S. Rose-John, Christian-Albrechts-University, Germany) for 12 and 24 h. Cells were seed in triplicates for all conditions and time points. After 12 and 24 h, cells were collected by centrifugation (5 min at 500 × g) and RNA was isolated using RNeasy Mini Kit (Qiagen, Germany) according to the manufacturer's protocol. Microarray analysis was performed using the Whole Human Genome Microarray Kit, 4x44K (G4112F, Agilent Technologies, Germany).

For transcriptome analysis of LRCs, nLRCs, and QSCs, HMECs were cultured and cells isolated as described above and cDNA was obtained from manually isolated single cells using WTA.

Labeling of cDNA was performed by PCR with Cy5-labeled primers. Reaction mix contained 5 μl of buffer I (Expand Long Template, Roche, Germany), 3% (v/v) deionized formamide, 0.35 mM each dNTP, 2.5 μM 5′-U*CAGAAU*TCAU GCCC*CCCC*CCCC*C-3′ primer (*denotes nucleotides conjugated with Cy5 fluorophore; Metabion), 3.75 U of Pol Mix (Expand Long Template, Roche, Germany) and 1 μl of WTA product or 100 ng cDNA from bulk RNA preparations of MCF-10A cells in a final volume of 49 μl. PCR parameters were: one cycle with 1 min at 95 °C, 11 cycles with 15 s at 94 °C, 1 min at 60 °C, and 3 min 30 s at 65 °C, three cycles where the elongation time was increased 10 s per cycle, and finally one cycle with an elongation time of 7 min. Labeled products were purified using a PCR Purification Kit (Qiagen, Germany) according to the instructions of the vendor. Purified Cy5-labeled DNA was denatured by incubation for 5 min at 95 °C, followed by incubation on ice. Hybridization solution was prepared by mixing 42 μl of denatured Cy5-labeled DNA, 55 μl of 2× HiRPM hybridization buffer (Agilent, Germany), 11 μl of 10× GE Blocking Agent (Agilent, Germany), 4 μl of 25% (v/v) Tween-20, and 4 μl of 25% (v/v) Igepal. Four 100 μl samples of hybridization mix were overlaid on four hybridization fields of Agilent Whole Human Genome Microarray Kit (4x44K) with SurePrint microarray slides and incubated for 17 h at 65 °C under constant rotation. After hybridization, slides were washed in Agilent Wash Buffer 1 for 1 min on a shaker in the dark and incubation continued in Agilent Wash buffer 2 prewarmed to 37 °C. Slides were dried by washing for 30 s in acetonitrile and scanned on a GenePix 4400A scanner (Molecular Devices, Germany). Numerical readouts of fluorescence intensities (GPR files) were generated using GenePixPro 7 (Molecular Devices, Germany).

**NGS mRNA library preparation and sequencing**. The majority of the (TTT)₇ and (CCC)₅ nucleotides forming the ends of cDNA products were removed by a limited-cycle PCR with primers introducing BpuEI and BglI restriction sites (Protocol 1; Supplementary Table 1) or a BglI restriction site (Protocol 2; Supplementary Table 1), followed by a restriction enzyme digestion. For Protocol 1, 1 μl each of two separate 1/5 dilutions of the original WTA sample was used in two sets of five separate PCR reactions (total of ten reactions) with a total volume of 20 μl per reaction with 1.44 μM final concentration of primer CP2-BpuEI (5′-TCA GAA TTC ATG (CCC)₅ GTC TTG AGT TTT TT-3′) and 1.44 μM final concentration of primer Cp2-BglI-13C (5′-TCA GAA TTC ATG (CCC)₂ CGG (CCC)₂-3′) for amplification. After an initial denaturation at 95 °C for 1 min, five cycles of 94 °C for 15 s, 55 °C for 1 min, and 65 °C for 180 s and three cycles of 95 °C for 15 s, 55 °C for 1 min, and 65 °C for 210 s (+10 s/cycle) were carried out, followed by a final extension step of 7 min. For Protocol 2, 1 μl of a 1/5 dilution of the original WTA sample was used for five separate PCR reactions with a total volume of 20 μl per reaction with 2.4 μM final concentration of primer Cp2-BglI-13C (sequence above) for amplification. After an initial denaturation at 94 °C for 2 min, eight cycles of 94 °C for 15 s, 68 °C for 1 min, and 68 °C for 240 were carried out, followed by a final extension step of 7 min. Following the PCR, the single PCR reactions were pooled and the remaining steps were identical for both protocols. Resulting cDNA products were purified with 1.8 volume of Ampure XP beads (Beckman Coulter, USA) according to the manufacturer's instructions and eluted in 40 μl of distilled water. Next, 5 μl of EcoRI buffer supplemented with 80 μM S-adenosyl methionine (New England Biolabs, Germany), 2.5 μl distilled water, and 2.5 μl BpuEI (5 U/μl) were added for a total volume of 50 μl and incubated at 37 °C for 1 h, followed by heat inactivation of the enzyme for 20 min at 65 °C. Subsequently, 1 μl of EcoRI buffer supplemented with 80 μM S-adenosyl methionine, 6.5 μl distilled water, and 2.5 μl BglI (10 U/μl) were added for a final volume of 60 μl and incubated for 3 h at 37 °C, followed by heat inactivation of the enzyme for 20 min at 65 °C. The complete restriction digest was purified with 1.8 volume of Ampure XP beads according to the manufacturer's instructions and eluted in 16 μl of 10 mM Tris-Cl, pH 8.5 (Elution buffer, Qiagen, Germany). The length distribution of purified cDNA populations was determined on the Bioanalyzer 2100 (Agilent Technologies, USA). Optimal Covaris settings for fragmentation of each purified cDNA sample to 350 bp insert size were determined on the basis of the average length distribution. Subsequently, sequencing libraries were prepared according to the TruSeq DNA PCR-Free Library Prep Kit (Illumina, USA). Resulting libraries were quantified with KAPA Library Quantification Kit for Illumina Platforms (Kapa Biosystems, RSA), pooled in equal molar ratios, and sequenced on Illumina NovaSeq 6000 platforms.

**IL-6, IL-6RA, and gp130 mRNA expression analysis in single cells**. IL-6, membrane IL-6 receptor, spliced IL-6 receptor, and gp130 expression was assessed by PCR using the MJ Research Peltier Thermal Cycler Tetrad (Bio-Rad, Germany) with the following primers: IL-6 (forward primer, 5′-GAG AAA GGA GAC ATG TAA CAA GAG T-3′ and reverse primer, 5′-GCG CAG AAT GAG ATG AGT TGT-3′, with an annealing temperature of 62 °C and an amplicon size of 388 bp), membrane vs. spliced IL-6RA (forward primer, 5′-CTG CAA ATG CGA CAA GCC TC-3′ and reverse primer, 5′-GTG CCA CCC AGC CAG CTA TC-3′, with an annealing temperature of 62 °C). The spliced and membrane-bound IL-6 receptor can be distinguished according to their PCR product size: mIL-6RA 380 bp and spliced IL-6RA 286 bp. Gp130 forward primer was 5′-GGA CCA AAG ATG CCT CAA CT-3′ and reverse primer was 5′-GGC AAT GTC TTC CAC ACG A-3′, with an annealing temperature of 58 °C and an amplicon size of 280 bp.

gp130 expression was assessed by quantitative PCR (qPCR) on reamplified and purified (Qiagen PCR Purification Kit, Qiagen, Germany) WTA products of single cells[74]. To normalize for the template input, quantification of yields in the individual samples was spectrophotometrically conducted using the NanoDrop 2000 instrument. The DNA input for each qPCR of a single cell was normalized to 2.5 ng and the qPCR run[74] with the following primers: gp130 forward primer, 5′-ATA TTG CCC AGT GGT CAC CT-3′ and reverse primer, 5′-AGG CTT TTT GTC ATT TGC TTC T-3′, with an annealing temperature of 58 °C and an amplicon size of 125 bp. Fold changes in gp130 expression were calculated from the delta Cp values between MCF-10A or MCF-7 cultured with and without MSCs.

**Flow cytometry**. Spheres or adherent cells were trypsinized with trypsin/EDTA (Pan-Biotech, Germany) for 3 min, if not stated otherwise. MSC monocultures and cocultures of MCF-10A-GFP cells with MSCs, OBs, and HUVECs were harvested by trypsin/EDTA (Pan-Biotech, Germany) for 5 min and using cell-scrapers. To reduce nonspecific binding single-cell suspensions were incubated for 5 min at 4 °C with PBS/10% AB serum (Bio-Rad, Germany), subsequently stained with fluorescence-labeled or biotinylated antibodies for 15 min at 4 °C and washed once with PBS/2% FCS/0.01% NaN₃. In case of biotinylated primary antibodies, phycoerythrin (PE)-labeled streptavidin (Dianova, Germany) was used as a secondary staining reagent. Cells were stained using the following antibodies: anti-human CD24-APC (ML5), anti-human CD34-PE (581), anti-human CD44-V450 (G44-26), anti-human CD45-FITC, APC, or PerCP-Cy5.5 (HI30), anti-human CD90 Alexa Flour 700 (5E10), anti-human CD105-FITC (43A3), anti-human CD130-APC (2E1B02), anti-human Nestin-PE (10C2), biotinylated anti-human IL6R (UV4), isotype control mouse IgG2a-APC (MOPC-21), isotype control mouse IgG2b-V450 (MOPC-21), isotype control IgG1-biotin (MOPC-21) (all purchased from BioLegend, Germany), and anti-human EpCAM (HEA-125, Miltenyi Biotech, Germany). Viability dye eFlour 780 (ebioscience, Germany) was used for live/dead cell discrimination. Cells were analyzed on an LSR II machine equipped with the FACS DIVA 5.03 software (BD Bioscience, Germany) and data were analyzed with FlowJo 8.8.6, 10.1, or 10.5.3 (Treestar, USA). Sorting of PKH26-labeled LRC and nLRCs was performed with a FACSAria cell sorter (BD Bioscience, Germany).

**IL-6 and sIL-6RA detection by ELISA**. IL-6 and sIL-6RA concentrations were assessed in 100 µl cultured media obtained from HMECs or MCF-10A cells propagated under anchorage-dependent or anchorage-independent conditions with the Human IL-6 DuoSet or Human sIL-6R alpha DuoSet ELISA Kit (R&D Systems, Germany) following the manufacturer's recommendations.

**Inhibition of ADAM proteases**. MCF-10A cells were treated for 48 h with 20 µM TAPI-2 (TNF protease inhibitor 2) acetate salt (Sigma-Aldrich, Germany). The culture supernatant was tested for the presence of IL-6 and sIL-6RA by ELISA.

**Immuno-(western) blotting**. Cell lysates were prepared using ice-cold RIPA buffer (radioimmunoprecipitation assay buffer) supplemented with cOmplete, EDTA-free Protease Inhibitor Cocktail and PhosSTOP™ (all from Sigma-Aldrich, USA). The protein concentration of lysates was determined with Pierce™ BCA Protein Assay Kit (Thermo Fisher Scientific, USA). Cell lysates were mixed with 4× Laemmli Sample Buffer (Bio-Rad, USA) containing 10% 2-mercaptoethanol (Sigma-Aldrich, USA) and denatured for 5 min at 95 °C. Ten micrograms of protein/lane were loaded on 12% Mini-PROTEAN® TGX™ Gels (Bio-Rad, USA) and protein separation was performed with sodium dodecyl sulfate (SDS)-polyacrylamide gel electrophoresis running buffer (25 mM Tris, 192 mM glycine, 0.1% SDS). Proteins were blotted onto Immobilon-P PVDF Membranes (Millipore, USA). For washing of membranes TBS-T (137 mM NaCl, 20 mM Tris, 0.05% (w/v) Tween-20, pH 7.6) was used. To detect signaling protein, the following primary antibodies (all from Cell Signaling Technology, USA) were used at dilutions according to the manufacturer's instructions: anti-phospho-STAT3[Tyr705] (clone D3A7, 1:2000), anti-phospho-AKT[Ser473] (clone D9E, 1:2000), anti-phospho-ERK1/2[Thr202/Tyr204] (clone E10, 1:2000), anti-STAT3 (clone 124H6, 1:1000), anti-AKT (clone 40D4, 1:2000), and anti-ERK1/2 (clone 137F5, 1:1000). As a loading control, an anti-α-tubulin antibody (Sigma-Aldrich, USA, clone DM1A; 1:5000) was used. This was followed by incubation with horseradish peroxidase-conjugated goat anti-rabbit IgG (anti-phospho-STAT3Tyr705, anti-phospho-AKTSer473, anti-ERK1/2) or goat anti-mouse IgG (anti-STAT3, anti-AKT, anti-phospho-ERK1/2Thr202/

Tyr204, anti-α-Tubulin). Secondary antibodies were used at 1:10000 (both from Sigma-Aldrich, USA). Protein bands were visualized using SuperSignal™ West Pico PLUS Chemiluminescent Substrate (Thermo Fisher Scientific, USA). Chemiluminescence was recorded by a ChemiDoc™ MP Imaging System (Bio-Rad, USA) and analyzed with Image Lab™ (version 6.1, Bio-Rad, USA). Membranes were stripped for reprobing using with anti-AKT, anti-ERK1/2 and anti-STAT or anti-α-Tubulin using Restore™ Plus Western Blot Stripping Buffer (Thermo Fisher Scientific, USA) for 15 min at RT. Removal of the enzyme conjugate was tested by incubating the membrane with new chemiluminescent substrate working solution. We noted that remnant anti-phospho-AKT and anti-phospho-ERK1/2 antibodies reduced anti-AKT and anti-ERK1/2 antibody binding.

**MCF-10A HIL6/IL-6 stimulation mRNA microarray data**. Gene expression data were obtained using the Agilent Whole Human Genome Microarray Kit (4x44K) and quality assessed by inspection of chip raw images and gene expression frequency distributions. All 24 expression profiles (three biological replicates, four treatment groups, two time points) were of sufficiently high quality for further bioinformatic analysis. Raw gene expression data were background corrected (limma Bioconductor package[75], version 3.36.5, normexp method), log₂-transformed, and normalized by quantile normalization. Replicated probes (identical Agilent IDs) were replaced by their median per sample. Gene ranking was performed using empirical array quality weights[76] and linear models from the limma Bioconductor package (version 3.36.5) using standard treatment vs. control contrasts. Gene annotation (Supplementary Table 3) was obtained by aligning Agilent oligo sequences to NCBI RefSeq genes (August 8, 2019) using BLAST[77] (version 2.9.0) requiring 100% identical matches, a maximum length difference between oligo and target sequence of one, and <100 hits per oligo. In addition, ensembl annotation[78] (version 97) was retrieved and used as a secondary information source (e.g., for oligos that were unannotated by NCBI RefSeq). GENCODE metadata (version 32)[79] were used as complementary annotation. For gene lists, graphical display, and functional annotation, probes targeting the same gene were disambiguated by retaining only the probe with the lowest p value. Differential gene expression was defined by a maximum false discovery rate (FDR)-adjusted p value of 0.05 and a minimum absolute log₂fold change of log₂(1.5) = 0.58. Computations were performed using R version 3.5.1[80].

**Mammary cell subpopulation mRNA microarray data**. Human mRNA expression data from Lim et al.[41] based on Illumina HumanWG-6 v3.0 BeadChip microarrays were downloaded from the Gene Expression Omnibus (series GSE16997). Data preprocessing, analysis, and annotation was performed analogous to the procedure detailed above for MCF-10A cells, except that the linear model included all pairwise contrasts between the three cell types.

Fold-change analysis of the MCF-10A and mammary cell subpopulation data were performed by first selecting a pairwise comparison from the MCF-10A data (e.g., classical IL-6 stimulation vs. control) and another from the mammary subpopulation data (e.g., LumProg vs. MatLum), each performed according to moderated t-testing (limma Bioconductor package, version 3.36.5). The differential gene lists of both comparisons were intersected and the randomness of their overlap quantified using hypergeometric testing (Supplementary Table 6). Second, the log fold changes of both comparisons were correlated without centering (i.e., without subtracting the respective group means) because reference to zero log fold was intended. Correlation p values were calculated according to centered Pearson's correlation.

**LRC/QSC/nLRC mRNA microarray data**. Gene expression data were obtained using the Agilent Whole Human Genome Microarray Kit (4x44K). All chips passed quality assessment and were preprocessed and annotated as described for MCF-10A cells above, except that no fold-change limit (originally used in the data of Lim et al.[41] and thus also employed for MCF-10A cells) was applied. The data showed patient effects that were accounted for by including patient IDs as second covariate in the linear model after safeguarding independence between patients and sample groups: Cramer's V with bias correction = 0; R-package rcompanion version 2.3.7[81]. For graphical display, these effects were compensated for by using the function removeBatchEffect from the Bioconductor package limma (version 3.36.5). Dimension reduction to two-dimensional according to t-SNE (Rtsne R-package, version 0.15[82]; Fig. 2c) and pairwise differential expression analysis (number of differentially expressed genes was: 35 for nLRC vs. QSC, 127 for LRC vs. nLRC, and 163 for LRC vs. QSC; FDR-adjusted p value <0.05; Supplementary Table 2) revealed that nLRC and QSC were much more similar to each other as compared to LRC. To concentrate on the main effects, nLRC and QSC were pooled resulting in 216 differentially expressed genes for LRC vs. (nLRC + QSC). Enrichment analysis was aimed at the NCI-Nature Pathway Interaction Database[83] for its focus on cancer research and treatment and conducted using the R-package enrichR[84] (version 2.1).

**DCC and HD mRNA sequencing data**. The sequencing quality was evaluated per sample with FastQC (version 0.11.8)[85] and in a multisample comparison with MultiQC (version 1.8)[86] before and after adapter trimming and contamination screening. Briefly, raw sequencing data of single cells (30 M0-stage DCCs, 11

M1-stage DCCs, and 15 EpCAM[+] cells from HDs from 21, 5, and 7 patients, respectively) were trimmed, and remaining adapter sequences as well as low sequencing quality bases at the end of each read were removed using BBDuk[87]. In order to increase the mapping quality (lowering false-positive alignments), read decontamination was performed using BioBloom Tools[88] with filters for the genomes of *Homo sapiens* (hg38), *Mus musculus* (mm10), *Escherichia coli* (BL21), *Mycoplasma pneumoniae* (M129), *Sphingobium* sp. (SYK-6), *Bradyrhizobium japonicum* (USDA 110), *Pichia pastoris* (GS115), *Malessia globosa* (CBS 7966), *Aspergillus fumigatus* (Af293), and a set of viral genomes (RefSeq, 5k+ genomes). All reads that did not map exclusively to hg38 (GENCODE version 27, GRCh38.p10) or did not map at all were defined as likely contaminations and discarded from downstream processing. Subsequently, the cleaned sample reads were aligned to the reference genome hg38 with STAR (version 2.5.1b)[89]. Uniquely mapped reads were counted per gene per sample using featureCounts from Subread[90]. We performed quality control and checked for outlier samples with the Bioconductor package scater (version 1.12.2)[91] using the functions calculateQCMetrics and plotPCA for QC metrics with outlier detection enabled. The results showed that none of the samples was an outlier. Thus, we kept all samples for further analysis. Samples were sequenced in two batches with only very little association between batches and phenotype (M0/M1/HD): Cramer's V with bias correction = 0.11; R-package rcompanion[81] (version 2.3.7). We applied the multiBatchNorm (Bioconductor package: batchelor 1.0.1)[92] to all cells and further rescaleBatches (Bioconductor package: batchelor 1.0.1) to DCCs to remove batch effects and get the log$_2$ normalized counts. After batch correction, we obtained 8626 and 7359 expressed genes for HD cells and DCCs on average, respectively.

The top 500 most variable genes were analyzed using principal component analysis (PCA). From the genes annotated by GO terms containing "B cell," "epithelial," or "epithelium," the top 100 most variable were subjected to PCA. PCAs were calculated using prcomp (R stats package).

For pathway enrichment analysis, we filtered for protein coding genes and compiled 2 × 2 contingency tables for each sample and each pathway according to whether genes were expressed (log$_2$ (normalized counts) > 0) and present in the pathway. Contingency tables were subsequently evaluated according to one-tailed Fisher's exact test (R stats package). Calculations were performed using R version 3.6.0.

**Analysis of CNAs**. To enable the combined analysis of mCGH- and LowPass-Seq-derived CNA profiles, the genomic coordinates obtained with the LowPass bioinformatics analysis pipeline (Menarini Silicon Biosystems, Italy) were converted to cytoband information using a custom script for R[80] and the UCSC Goldenpath reference (version hg38[93]). Afterwards, the aberrations were manually screened and compared to the respective CNA profile images before being annotated according to the specifications of the International System for Human Cytogenetic Nomenclature (ISCN)[94]. Small aberrations <1 Mb as well as recurring technical artifacts in chromosome 1p and centromeric and telomeric regions were excluded. Finally, the combined ISCN-annotated aberration data (mCGH and LowPass-Seq) were stratified into M0 and M1 groups and submitted to the Progenetix user data tool[95] to generate individual frequency plots for M0 and M1 cells.

**Statistical analysis**. Statistical analysis was performed using the GraphPad Prism 6.0 software (GraphPad Software, Inc., USA). Differences in mean values between groups were analyzed by Student's *t* test, Mann–Whitney test, or one-way ANOVA, followed by post hoc statistical testing, where appropriate. Time dependencies were analyzed by regression analysis (*F* test). Independence in contingency tables was assessed by Fisher's exact test. All tests were realized two-sided. A *p* value < 0.05 was considered statistically significant.

**Reporting summary**. Further information on research design is available in the Nature Research Reporting Summary linked to this article.

## Data availability
The microarray (Figs. 2c and 5c) and RNA-Seq (Fig. 3b and Supplementary Fig. 1c) data have been deposited in the European Genome-phenome Archive under the accession code EGAS00001004597. All the other data supporting the findings of this study are available within the article and its Supplementary information files and from the corresponding author upon reasonable request. A reporting summary for this article is available as a Supplementary information file. Source data are provided with this paper.

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

## Acknowledgements

We thank Irene Nebeja, Irina Hartmann, Michaela Becker, and Silvia Materna-Reichelt for excellent technical assistance. We are grateful to Alberto Bardelli for hTERT-HME1 cell lines and Cyrus Ghajar for E4ORF1 HUVECs. This work was supported by grants to C.A.K. from the Deutsche Krebshilfe (TransLUMINAL-B 111536, to C.A.K. and BP DETECT CTC Project 6, 70112504), the Dr. Josef Steiner foundation, the ERC (grant 322602), the Bavarian ministry of economic affairs, energy and technology (AZ 20-3410.1-1-1), the Deutsche Forschungsgemeinschaft (DFG; KL 1233/10-2), and the Bavarian Research Foundation (Bayerische Forschungsstiftung, DOK-165-13). The work of M.W.-K. has been funded by the DFG (WE 4632/1-1, WE 4632/4/1. and WE 4632/5-1 within the FOR2127). TissueFAX was supported by the DFG (INST 89/341-1 FUGG). The work of S.R.-J. has been funded by grants from DFG in the SFB841 (project C1), the SFB 877 (projects A1 and A14), by the Deutsche Krebshilfe, and by the German-Israeli Foundation for Scientific Research and Development.

## Author contributions

Conception and design: C.A.K. and M.W.-K. Development of methodology: M.W.-K., A.G., M.O., M.H., S.K., S.T., J.W., K.H., and S.R.-J. Acquisition of data: M.W.-K., A.G., C.I., M.O., S.T., C.K., C.B., K.W., C.W., S.K., M.G., K.H., G.F., I.B., S.G., E.S., G.H., N.P., S.G., S.H., B.R., N.H., S.B., P.R., and N.H. Analysis and interpretation of data: M.W.-K., A.G., M.H., X.L., H.K.-Q., B.P., J.W., K.H., Z.C., C.A.K. Writing of the manuscript: M.W.-K., M.H., and C.A.K. Review and/or revision of the manuscript: all authors.

## Funding

## Competing interests

The authors declare no competing interests.
