## [Peer Review File · Nature Communications]

Reviewers' comments:

Reviewer #1 (Remarks to the Author):

The manuscript is well written and addresses a key outstanding question in breast cancer metastasis, the regulation of tumour cell dormancy and the nature of DTCs. Almost impossible to study in patients, the authors understandably carry out their work mainly in cell models to demonstrate mechanistic link between IL6 signalling and DTC progression.

The study does provide novel information in this important area, however there are a number of points that require clarification.

1) In line 84 in the introduction, the authors state that their data “shed light onto the so far completely dark stage of early metastasis formation....” This dismisses a large volume of work in this area and should be toned down.

2) The information provided in the highly important table in figure 1 is difficult to interpret. It is unclear which of the different injection routes of tumour cells resulted in engraftment. For example, DTCs from only 2 M1 BrCa patients were injected How? As cells or spheres? And in what site were they implanted to generate the single engraftment? It is also confusing to use ‘injected patients’ term here. It is not clear if the different application methods were tried for individual patient samples, or whether samples were expanded ex vivo and then split and injected in different formats/sites. This information should be included. Were the successful engraftments from ER+ve disease (re breast cancer)?

3) The authors state (line 115) that genetically mature DCCs from M1 stage patients generated xenografts in about 50% of cases. The table in figure 1 appears to show that this happened in exactly 50% of cases, as cells from only 2 out of 4 patients were successfully engrafted. It is therefore an overstatement (line 105) that ‘BM-derived DCCs from M1 patients readily engrafted’. If my interpretation of the data presented in figure 1 is correct, then it is not possible to generalise in this way based on such a small sample size. The authors should clarify this limited samples size and rephrase the statements accordingly.

4) DTCs harvested from patients were in some cases grown as spheres (expanded ex vivo in 3D), this clearly results in genetic/phenotypic changes compared to their original state as single cells in the bone marrow. The authors should include some discussion of this approach.

5) It is difficult to follow in which cases samples from different patients are analysed, or when multiple DCCs were isolated from a single patients. For example (line 155) – were the 26 DCCs from a single patient?

6) Throughout the manuscript, the authors are very assertive when it comes to the description of the implications of their data, despite some of the effects shown being obtained from different sample sizes. For example, in figure 6, the groups analysed are HR+(n=71), Her2 amplified (n=7) and TN (n=12). It would be helpful to include this information in the text where the results are described

(line 336). This imbalance may come from the patient material available to study, however it would strengthen the case if the patient samples analysed in each category were more balanced. When one group comprises 10 times the number of patients in another the likelihood of obtaining a positive result is affected. This should be discussed.

7) There is no direct link between the human cell work (DTCs from bone marrow samples) and bone metastasis/dormancy in bone. The authors focus on stemness in MCF10A but fail to demonstrate that their findings are of relevance using in vivo models of disseminated tumour cells in bone to support their findings. How does manipulation of the IL6 pathway affect the ability of tumour cells to home to, survive and form colonies in mice?

Reviewer #2 (Remarks to the Author):

In their manuscript titled “Interleukin-6 trans-signaling is a candidate mechanism to drive progression of human DCCs during periods of clinical latency”, Werner-Klein and colleagues profile model systems as well as disseminated cancer cells (DCCs) from the bone marrow and identify IL6/PTEN/PI3K signalling as a candidate DCC activation pathway.

This reviewer was asked to specifically provide an opinion on the NGS and bioinformatics related aspects of the work, and the comments below therefore relate mainly to those results.

First of all, it is important to note there is ongoing controversy in the field regarding the origins of at least some of the (aberrant) cells picked from bone marrow of mainly non-metastatic patients through EpCAM+ and/or CK+ staining (e.g. PMID 27974799, 27931250). Comparison with the primary tumour as well as sequencing efforts interrogating matched primary-metastasis pairs suggest that (micro)-metastases often derive only relatively late in mutational time, sharing a considerable number of mutations with the primary tumour (PMID 26410082, 28810143, 29656895). Vice versa, the mere presence of copy number aberrations in CK+ bone-marrow derived cells was shown not to be a guarantee for DCC status.

However, no mention of these contrasting hypotheses is made in the current manuscript, leading to frequent overinterpretation of the data. Moreover, while authors describe confirming the presence of copy number changes in the isolated DCCs (“M0- and M1-stage DCCs with confirmed malignant

origin5”), they fail to show these data or contrast them with the copy number profiles of the primary tumours.

In addition, many statistical approaches employed are clearly not appropriate:

- Based on the findings in Fig 2C-D (and a previous description of the role of IL6 in mammosphere formation), the authors solely scrutinise the IL6 pathway in DCCs in all downstream analyses, despite having access to the whole transcriptome. This is really cherry-picking from the data.

- K-means clustering on the 6000 most variable genes in expression array data from 23 WTA single cells is unlikely to yield reproducible results (Fig 2D). This is reflected in the sense that 3 populations are expected (LRC, nLRC, QSC) and the clustering analysis fails to accurately reproduce this (LRC and QSC split up and LRC2 is closer to all other classes)

- In the LRC/nLRC/QSC analysis, the authors should also indicate which cells derive from which reduction mammoplasty patient, as one may expect this to affect clustering as well. Importantly, a table should be provided relating cells to patients.

- Patient annotation is also missing for the M0- and M1-stage DCCs in Fig 2D, despite this being a likely confounder during clustering. Further issues with the clustering analysis likely arise from improper handling of dropouts (a single pseudocount is currently added to each gene/sample instead).

- In Fig 6B (and S2E, S3f) multiple testing correction is applied to just 4 distinct t-tests, while overly conservative, this is still a bias.

- The same applies to Fisher’s exact tests, which are best replaced by exact unconditional tests (Lydersen et al., doi 10.1002/sim.3531).

- Vice versa, for the gene set/pathway analysis (Fig 2C) in which a large number of hypotheses is tested, such corrections are omitted and results with $p < 0.05$ are simply scored as 1 and summed. As a likely result, many likely irrelevant gene sets (e.g. “pathogenic E. coli infection” and “mitochondrial tRNA aminoacylation”) are reported.

In addition, there are discrepancies between Fig 2D and 2F: IL6R is shown in panel D and appears to be expressed in a number of cells, while panel F fails to confirm this in the same cells through qPCR (present in 0% of cells). Gp130 and IL6 should also be included in panel D if not all 0.

Therefore, in conclusion, this study presents a very unbalanced view of the field, and seems to use cherry picking of results by statistically unsound criteria, and it is unclear if their results and conclusions are correct or not.

Reviewer #3 (Remarks to the Author):

Review of NCOMMS-19-01747A-Z

Interleukin-6 trans-signaling is a candidate mechanism to drive progression of human DCCs during periods of clinical latency

Summary

This work is aimed at indentifying the molecular mechanisms that drive the outgrowth of disseminated cancer cells into life-threatening metastases in the context of breast cancer dissemination to bone. The authors found that gp130-mediated IL6 trans-signaling played critical roles in allowing the stem-like outgrowth of disseminated cells, provided they end up in the vicinity of a favorable niche. Cells harboring oncogenic lesions exhibit less dependency on such signaling.

The work is of interest and relies on a wealth of bioinformatic genomic and transcriptomic data. However, there are serious concerns in regards to the models used, the extrapolations the authors make regarding the effects of the sgp130Fc, as well as the correlative nature of many of the findings, especially those pertaining to PI3K role in the last figure.

Some comments:

- 1- The fact that primary-tumor-derived cancer cells from M0 patients did not establish xenografts, but those from M1 patients did, suggests that genetic alterations within the primary tumor are necessary for engraftment, albeit not sufficient. This point needs to be made clearer as the text gives the impression that non-cancer-cell-autonomous pathways are both necessary BUT ALSO, sufficient.
- 2- With the caveat that clinical tissues are not easy to come by, it would be advisable to increase the numbers of M1 specimens in Table 1. As is, the 50% take rate could be stochastic. Considering the conclusion based on the comparisons between M0 and M1, this deserves a bigger n number.
- 3- It is not clear how faithful the hTERT-HME1-EGFRdelta746-750 is. It exhibits IL6 trans-signaling, but undoubtedly so many other pathways (compensatory or not) as well.
- 4- Does the increase in sphere formation abilities of the different groups in Fig. 3 result in fluctuations in label-retaining cells? This is important to the premise of the studies.
- 5- The relevance of MCF10A co-cultures with MSCs and OB is not convincing. Yes, in these settings, IL6 trans-signaling might be inhibited, but that does not necessarily mean that DCCs per se will exhibit the same reaction. The authors observed that DCCs exhibit lower levels of gp130 than normal cells from reduction mammoplasty, but the comparison should be to non DCC from primary tumors instead.
- 6- Do the HME1-shPTEN cells interact differently with MSCs/OB than MCF10A in Figs. 4 and 5?
- 7- How about cells in which PIK3CA is actually mutated, such as at position 1047? Do those exhibit similar traits to the PTEN sh cells?
- 8- Gp130 also serves as a LIFR, which was found by Dr. Giaccia and colleagues to play critical roles in regulating dormancy of breast cancer cells in the bone. How is this different than the present study invoking IL6 trans-signaling? The authors do not even mention this prior study at all and at the very least, the prior work needs to be discussed.

Reviewer #4 (Remarks to the Author):

Werner-Klein and colleagues investigate the role of IL6 in DCC activation. This is a clinically-relevant and very important question with an impact on breast cancer management. However, at this stage, the manuscript suffers from the following major weaknesses:

- (1) What is the evidence that DCCs used for xenotransplantation (Fig 1) are cancer cells? CNV analysis of a few representative ones would help.

(2) The number of LRCs (n=8), nLRC (n=5) and QSC (n=10) is extremely low for a single cell sequencing (or microarray) experiment, leaving too much room for technical biases to interfere with the real signals (e.g. dropouts, cell cycle phase etc.). This is confirmed by the fact that the pathway analysis reveals a number of pathways that appear to be enriched but that are completely ignored by the authors (Fig 2c). Also, data quality is not described (e.g. how many transcripts/genes are detected per cell on average?). Selection of IL6 seems more like a cherry-picking exercise rather than an unbiased/result-driven conclusion. A much higher number of cells for each category would help to gain confidence in the results.

(3) Linked to the above. Even when confirmed with a higher cell number, the finding of IL6 being involved in mammosphere formation is not novel. The authors themselves point out "IL6 signaling had been previously found relevant for mammosphere-formation from ductal breast carcinoma and normal mammary gland²³".

(4) IL6 pathway activation in M1-DCCs as opposed to M0 is unconvincing. Most of the IL6 signature genes are lowly expressed in M1 samples, with only some very limited subset of these genes displaying a higher expression. This is not enough to conclude that IL6 pathway is active. Also, data quality is not described: can the authors exclude that M1 cells have a higher number of detected transcripts/better quality compared to M0?

(5) The functional testing of IL6 role for DCC is entirely done in non-neoplastic cells (e.g. HMECs and MCF10A). No direct evidence is provided to support a role of IL6 in DCC progression.

(6) Experiment in Fig. 6c is unclear. How many cells were analyzed and for how many of these a CNV profiling was obtained (i.e. confirmation of cancer origin). The table refers to patients and not cells. This is important to make sure that comparisons are done in homogeneous groups. I disagree with the statement "These data are fully consistent with a scenario in which early DCCs depend on IL6 trans-signaling and become increasingly independent during cancer evolution." Rather, current data on DCCs in figure 6 simply shows that some of them may harbor a PIK3CA mutation. Metastasis-derived CTCs are not representative of early DCC events.

Minor:

(1) Fig1a refers to "engrafted patients". Aren't these mice instead?

Reviewer #1: (Expertise: Breast cancer bone metastasis, Remarks to the Author)

We would like to thank the reviewer for his/her careful reading of our manuscript. We addressed all points raised and performed the requested changes. We feel that the manuscript has been improved on the basis of his/her criticism.

The manuscript is well written and addresses a key outstanding question in breast cancer metastasis, the regulation of tumour cell dormancy and the nature of DTCs. Almost impossible to study in patients, the authors understandably carry out their work mainly in cell models to demonstrate mechanistic link between IL6 signalling and DTC progression. The study does provide novel information in this important area, however there are a number of points that require clarification.

1) In line 84 in the introduction, the authors state that their data "shed light onto the so far completely dark stage of early metastasis formation...." This dismisses a large volume of work in this area and should be toned down.

Reply: What indeed is fairly "dark" is early metastasis formation in patients. We agree that some work has been done in mouse models, however, doing intensive research over the last 25 years in this specific area we would argue that in patients our knowledge of early metastasis formation is rather circumstantial.

Changes: This sentence in our introduction was toned down and changed to: "The data shed light onto the so far dark stage of early metastasis formation in patients..."

2) The information provided in the highly important table in figure 1 is difficult to interpret. It is unclear which of the different injection routes of tumour cells resulted in engraftment. For example, DTCs from only 2 M1 BrCa patients were injected. How? As cells or spheres? And in what site were they implanted to generate the single engraftment? It is also confusing to use 'injected patients' term here. It is not clear if the different application methods were tried for individual patient samples, or whether samples were expanded ex vivo and then split and injected in different formats/sites. This information should be included.

Reply: We modified table, figure legend and the methods section accordingly.

Changes: In brief, each sample was divided into two halves. In one half, DCCs were enumerated by staining for CK8/18/19 or EpCAM. The other half of the cell suspension from an M0-stage patient was transplanted into mice using one injection route or in some cases two (s.r. + s.c) as indicated in the table. In case of BM-aspirates from M1-stage patients, the BM-sample was split in several parts and transplanted via different injection routes as indicated in the table. Samples were not expanded ex vivo prior to injection unless spheres were injected. In this case, the DCC suspension was cultured under mammosphere conditions, spheres were isolated and transplanted via the indicated route in the table.

Were the successful engraftments from ER+ve disease (re breast cancer)?

Reply: The successful engraftment came from an ER-positive breast cancer (see ER-staining of the xenograft in Fig. 1b, upper panel).

3) The authors state (line 115) that genetically mature DCCs from M1 stage patients generated xenografts in about 50% of cases. The table in figure 1 appears to show that this happened in exactly 50% of cases, as cells from only 2 out of 4 patients were successfully engrafted. It is therefore an overstatement (line 105) that 'BM-derived DCCs from M1 patients readily engrafted'. If my interpretation of the data presented in figure 1 is correct, then it is not possible to generalise in this way based on such a small sample size. The authors should clarify this limited samples size and rephrase the statements accordingly.

Reply: We rephrased the statement. BM aspiration is rarely performed in M1-stage patients. Yet, in two out of four M1-samples, we observed engraftment whereas 42 samples of M0-patients never engrafted.

Changes: We replaced the "50%" by providing the numbers transplanted (i.e. "two out of four") and leave it up to the reader how to judge it.

4) DTCs harvested from patients were in some cases grown as spheres (expanded ex vivo in 3D), this clearly results in genetic/phenotypic changes compared to their original state as single cells in the bone marrow. The authors should include some discussion of this approach.

Reply: This is an interesting commentary. We shared this thinking before we started this type of experiments and therefore controlled for evidence of in vitro selection on a genome level. For melanoma, we were able to grow spheres that successfully implanted and formed xenografts. Interestingly, we did not note more than spurious differences in copy number alterations between ex vivo isolated DCCs and DCC-derived spheres or xenografts. This does not exclude that other molecular changes may occur that we haven't analyzed so far. This is ongoing work and will be summarized elsewhere.

For the data presented in the manuscript, however, such analysis was not possible, as we have not successfully expanded M0-stage DCCs that could be compared.

Changes: Since we felt this problem not to be relevant for the data presented we refrained from discussing it.

5) It is difficult to follow in which cases samples from different patients are analysed, or when multiple DCCs were isolated from a single patients. For example (line 155) – were the 26 DCCs from a single patient?

Reply: For the study presented, we investigated 246 breast cancer patients. We doubled the number of M0-stage DCCs (from n=15 to n=30). Figure S1 now transparently explains how many samples were initially tested and explains the multiple selection steps and reasons for individual samples/cells/patients dropping-out from the analysis.

Changes: See Figure S1.

6) Throughout the manuscript, the authors are very assertive when it comes to the description of the implications of their data, despite some of the effects shown being obtained from different sample sizes. For example, in figure 6, the groups analysed are HR+(n=71), Her2 amplified (n=7) and TN (n=12). It would be helpful to include this information in the text where the results are described (line 336). This imbalance may come from the patient material available to study, however it would strengthen the case if the patient samples analysed in each category were more balanced. When one group comprises 10 times the number of patients in another the likelihood of obtaining a positive result is affected. This should be discussed.

Reply: We agree with reviewer #1 that we only superficially treated the problem of breast cancer subtypes. Most of the cells analyzed are from patients with HR-positive breast cancers. The data are now provided in Figure S1a for the gene expression profiling.

Since most patients, for whom we present the pathway analysis, are HR-positive breast cancer patients, we now included in Figure 6c only HR-positive patients for the analysis of *PIK3CA* mutations.

Changes: See Figure S1a and Figure 6c. The discussion adds the limitation regarding other breast cancer subtypes.

7) There is no direct link between the human cell work (DTCs from bone marrow samples) and bone metastasis/dormancy in bone. The authors focus on stemness in MCF10A but fail to demonstrate that their findings are of relevance using in vivo models of disseminated tumour cells in bone to support their findings. How does manipulation of the IL6 pathway affect the ability of tumour cells to home to, survive and form colonies in mice?

Reply: In vivo experiments are hampered by two facts:

1) There is no model with spontaneous bone metastasis. 2) Regarding xenotransplantation the major difficulty to study the effect of IL6 signaling in mice is that binding of mouse IL6 to IL6RA is species-specific. Therefore, our ability to study the impact of the IL6 pathway on homing, survival and colony formation of human tumor cells is very limited. This may be one additional reason (i.e. additional to

the lack of M1-cell-associated alterations) why our xenotransplantation attempts presented in Fig. 1 failed – as the whole DCC/DTC community has failed in expanding early bone marrow DCCs to date.

Changes: To address the reviewer’s point (more spheres result in more/larger tumors) we took a tumor-forming cell line (MDA-MB-231), which has no PIK3CA mutation and forms more spheres in response to HIL6. We pretreated the MDA-MB-231 cells for 3 hrs with HIL6, PBS or an anti-IL6 antibody before injecting the same number of cells into NSG-mice. Tumors from HIL6 pretreated cells had a significantly increased tumor size as compared to mice injected with PBS or anti-IL6 treated cells (Fig. 4h). These experiments show that IL6 trans-signaling increases the tumorigenicity of cancer cells.

Reviewer #2: (Expertise: NGS, bioinformaticss, Remarks to the Author)

We thank reviewer #2 for his/her constructive criticism. We spent considerable effort to address the points raised, particularly, we doubled the number of M0-stage DCCs analyzed, reconsidered all bioinformatics analyses and provide now a more stringent rationale of the approach. We are grateful for the chance to address these concerns and hope that the results are satisfying.

In their manuscript titled “Interleukin-6 trans-signaling is a candidate mechanism to drive progression of human DCCs during periods of clinical latency”, Werner-Klein and colleagues profile model systems as well as disseminated cancer cells (DCCs) from the bone marrow and identify IL6/PTEN/PI3K signalling as a candidate DCC activation pathway. This reviewer was asked to specifically provide an opinion on the NGS and bioinformatics related aspects of the work, and the comments below therefore relate mainly to those results.

First of all, it is important to note there is ongoing controversy in the field regarding the origins of at least some of the (aberrant) cells picked from bone marrow of mainly non-metastatic patients through EpCAM+ and/or CK+ staining (e.g. PMID 27974799, 27931250). Comparison with the primary tumour as well as sequencing efforts interrogating matched primary-metastasis pairs suggest that (micro)-metastases often derive only relatively late in mutational time, sharing a considerable number of mutations with the primary tumour (PMID 26410082, 28810143, 29656895). Vice versa, the mere presence of copy number aberrations in CK+ bone-marrow derived cells was shown not to be a guarantee for DCC status.

Reply: PMID 27931250 is a study that in our opinion is very difficult to discuss because the approach violated the consensus of the community. In a 2006 consensus paper (coauthored by one of the authors of PMID 27931250!) on how DTCs should be evaluated (Fehm, Cancer 2006), it was stated that isotype controls should be performed to identify patients with non-specifically stained individual cells using mouse monoclonal antibodies. Since then, it is the rule to exclude false positive samples. We have done so for the last 20 years and consider only patients positive that react only with cytokeratin antibodies or EpCAM antibodies and not with isotype control antibodies. We do not know why in PMID 27931250 the authors decided to exclusively (!) investigate false-positive samples. While we do not question the positive finding of that study (i.e. that 3 out of 6 patients had DCCs similar to the primary tumor), we do not share the interpretation of the authors that their data exclude early dissemination of low-aberrant cells. Rather, we still hold that true epithelial cells (i.e. cytokeratin positive cells in BM of isotype control negative patients) with aberrations do not belong into the BM and are indicative of epithelial cancer present in the patient - a reasoning strongly supported by the poor outcome of DCC positive patients in large clinical studies. (corroborating breast cancer data of 11000 patients that have recently been presented during the SABCS meeting in San Antonio).

PMID 26410082 clearly demonstrates that linear progression is rare. In fact, the study supported the parallel progression model by finding not a single example of linear progression (see supplementary information.)

Regarding PMID 29656895 and 28810143, it may suffice to state that none of these studies has analyzed DTCs or the origin of metastasis. We agree that the Yates paper (PMID 28810143) claims late dissemination for breast cancers, however, that paper does not directly address the origin of metastasis. From comparisons between primary tumors and metastases, it is concluded - based on numerous assumptions which may or may not be correct (e.g. is the infinite sites assumption really correct for cancer?) - that the founder cells disseminated late. It should further be noted that the Yates paper focuses on triple negative patients whereas most DCC data are for HR-positive patients. Regarding the renal cell cancer paper (PMID 29656895), we have no data on RCC and cannot comment on it. However, for colorectal cancer two recent publications (PMID 30318143, PMID 31209394) indicate that dissemination is very early - one early in relation to tumor size and one with relation to the observed genetic alterations. Thus, paired analysis of primary tumors and metastases does not provide such a uniform picture. The direct analysis of DTCs/DCCs, however, provides strong *direct* evidence that DCCs often disseminate early and acquire mutations after dissemination. Furthermore, in all spontaneous mouse models tested early dissemination was the rule.

For the present paper, we still think that the expression of typical epithelial markers plus the presence of copy number alterations provides sufficient evidence of a malignant epithelial origin. Most importantly, we do not share the assumption that copy number alterations are abundantly present in normal cells. Using our whole genome amplification method, we have performed numerous control experiments on normal cells, demonstrating that randomly picked, cytokeratin-negative bone marrow cells or cells positive for specific lineage markers (e.g. for B cells, T cells macrophages) display normal genomes. These data have been published extensively (Klein, PNAS 1999; Klein, Nat Biotechnology, 2002; Schmidt-Kittler, 2003; Guzvic, Cancer Research 2014; Werner-Klein, Nat Communications, 2018, among others, and independently from us by the Stoecklein group, for example, that is using the same technology).

However, no mention of these contrasting hypotheses is made in the current manuscript, leading to frequent overinterpretation of the data.

Reply: We thank for the suggestion to include this discussion.

Changes: We added a short paragraph on the issue of early vs. late dissemination.

Moreover, while authors describe confirming the presence of copy number changes in the isolated DCCs ("M0- and M1-stage DCCs with confirmed malignant origins"), they fail to show these data or contrast them with the copy number profiles of the primary tumours.

Reply: We now display the data for the cells analyzed. It is interesting to see that the EpCAM-positive aberrant cells from M0-stage patients (whose primary tumor is resected at BM sampling) present fewer aberrations than EpCAM-positive aberrant cells from M1-stage patients who developed metastasis after primary tumor resection. These data suggest that M0-DCCs progress to M1-DCCs. The same finding was made upon analysis of cytokeratin-positive cells published previously (Hosseini, Nature 2016), corroborating the results by using two different epithelial markers.

In addition, many statistical approaches employed are clearly not appropriate:

- Based on the findings in Fig 2C-D (and a previous description of the role of IL6 in mammosphere formation), the authors solely scrutinise the IL6 pathway in DCCs in all downstream analyses, despite having access to the whole transcriptome. This is really cherry-picking from the data.

Reply: Although we disagree that we cherry-picked from the data, we re-addressed this issue in a completely unbiased way. Doing so, the manuscript has improved in our opinion. We therefore thank the reviewer for his/her criticism.

Changes: Please, see revised Figure 2. In a first step, we re-assessed the group differences of label-retaining cells (LRCs), non-label-retaining cells (nLRCs) and quiescent single cells (QSCs) generated

from normal mammary glands. We focused on the gene expression of LRCs (i.e. the putative stem cells) as compared to the other two mammary cell types. We identified ALL pathways significantly enriched for genes differentially expressed between LRCs and the pooled nLRCs and QSCs and subsequently tested ALL of these pathways for enrichment in DCCs. Strikingly, four pathways, responsible for transmitting extracellular signals from outside to the cell nucleus, were highly enriched, among them the IL6 pathway. As these four pathways largely overlap for the expressed genes, we asked which pathway was triggered and found the most likely candidate to be IL6 trans-signaling. Specifically, we concluded that two of the pathways (connected to VEGFR2 and Tie2) are less relevant for microenvironmental interactions of DCCs as the ligand-binding receptors (VEGFR2 and Tie2) were not expressed in DCCs and expressed pathway genes mostly comprised intracellular proteins. The third identified pathway was the TCPTP pathway, which is a negative regulator of the IL6 pathway, on which we focus. This data-driven, unbiased approach was corroborated by our subsequent functional analyses.

- K-means clustering on the 6000 most variable genes in expression array data from 23 WTA single cells is unlikely to yield reproducible results (Fig 2D). This is reflected in the sense that 3 populations are expected (LRC, nLRC, QSC) and the clustering analysis fails to accurately reproduce this (LRC and QSC split up and LRC2 is closer to all other classes)

Reply: We now corrected for patient effects and performed unsupervised dimension reduction as well as differential expression analysis of the three groups. These analyses indicated that gene expression is most variable between LRCs and nLRCs/QSCs. We therefore concentrated on differentially expressed genes between LRCs and nLRCs/QSCs and used these to identify corresponding enriched pathways.

Changes: Please, see revised figure 2c.

- In the LRC/nLRC/QSC analysis, the authors should also indicate which cells derive from which reduction mammoplasty patient, as one may expect this to affect clustering as well. Importantly, a table should be provided relating cells to patients.

Reply and changes: We added labels for the patient-origin of each cell (Supplementary Table 1) and accounted for patients effects during differential gene expression analysis .

- Patient annotation is also missing for the M0- and M1-stage DCCs in Fig 2D (now added), despite this being a likely confounder during clustering. Further issues with the clustering analysis likely arise from improper handling of dropouts (a single pseudocount is currently added to each gene/sample instead).

Reply and changes: Please, find Supplementary table 1 containing the missing patient information. Adding a pseudocount is mathematically necessary to avoid minus infinity values when transforming count data to log-scale (as $\log(0) = -\infty$). This general practice is the *de facto* standard employed by all methods for mRNA-seq data analysis known to the authors. Adding a unit pseudocount is also the default method in the function `rescaleBatches` of the `batchelor` R-package (<http://bioconductor.org/packages/devel/bioc/manuals/batchelor/man/batchelor.pdf>) used in our present work. To emphasize that we do indeed conform to common analysis standards we refrain from our previous explicit mentioning of the pseudocount that may have raised the reviewer's concerns and refer to normalized read counts as obtained from the `batchelor` R-package instead. A distinction between technical dropouts and unexpressed genes in single cells is hard to make. Methods for differential expression (DE) accounting for dropouts exist (e.g. `DEsingle`) but we do not consider DE in our sequencing data. Also, these as well as dropout imputation methods heavily rely on model assumptions regarding stochastic gene expression and technical processes (e.g. by using the ZINB-model) as well as transferability of information between different cells that are not sufficiently verified to date. For this reason, we did not attempt to correct for dropouts.

- In Fig 6B (and S2E, S3f) multiple testing correction is applied to just 4 distinct t-tests, while overly conservative, this is still a bias.

Reply and changes: To avoid needless multiple-testing issues we switched to linear regression analysis wherever applicable (e.g. Fig. S2e). Regarding figures for which we used ANOVA in combination with post-hoc analysis (e.g. Fig. S3c/f; 3a/c/e/f; 4b/h; 5d) we still think that this is the appropriate way of analysis as it also conforms to the workflow guidance and help system implemented in the GraphPad Prism (version 6) statistical software we use: first, we test whether there are any significant differences between group means using ANOVA (F-test) and subsequently determine the groups that differ most from each other in a post-hoc analysis. To the best of our knowledge, post-hoc testing should be done using Dunnett's test when comparing the mean of each treatment group to the mean of a control group, as is appropriate in our cases (or Tukey's test if comparing group means to each other). The reviewer's concern that this approach is overly conservative may result from common notational imprecision regarding Dunnett's test. First, this test derives modified (i.e. stabilized against small variances) pairwise t-statistics based on a single variance estimate calculated from across all samples. Second, the largest of those t-statistics is used as the formal test statistics and allows for the control of the familywise error rate (FWER). Accordingly, Dunnett's test can account for multiple testing on its own right. However, in common applications the above individual pairwise t-statistics are used in post-hoc testing and their alleged 'multiple comparison' property exclusively refers to the stabilized variance estimate and not to 'hard' multiple testing corrections like Bonferroni, FWER or FDR. Nevertheless, as requested by the reviewer, we performed ANOVA together with a post-hoc test that does not correct for multiple testing (Fisher's LSD test) and included this information in Supplementary Table 7.

- The same applies to Fisher's exact tests, which are best replaced by exact unconditional tests (Lydersen et al., doi 10.1002/sim.3531).

Reply and changes: As suggested by the reviewer we applied the unconditional exact test to our data (Fig. 2b; 4c/d; 6c; S2a;). However, this test did not change the overall message of the respective figures. We therefore would prefer to use the Fisher's exact test as this is a well-known test with which the readers are familiar, the more so as computing p-values for e.g. Boschloo's unconditional exact test under the multinomial model typically took one hour to complete on a standard workstation. For the reviewer's interest we have included a table comparing results from Fisher's exact test with those from unconditional exact testing according to Boschloo (using binomial as well as multinomial sampling) for the respective figures (Fig. 2b; 4c/d; 6c; S2a;) and have included this information in Supplementary Table 7. We also assessed the differences between Fisher's exact test and Boschloo's unconditional exact test (binomial model) by evaluating 20,000 random 2x2 contingency tables (with integer entries between 0 and 50) and found high correlation between both methods and Fisher's test providing more conservative p-values (see figure below).

- Vice versa, for the gene set/pathway analysis (Fig 2C) in which a large number of hypotheses is tested, such corrections are omitted and results with $p < 0.05$ are simply scored as 1 and summed. As a likely result, many likely irrelevant gene sets (e.g. “pathogenic E. coli infection” and “mitochondrial tRNA aminoacylation”) are reported.

Reply and changes: In response to this criticism we have changed our approach. Please, see above for the description how the pathway analysis was handled.

In addition, there are discrepancies between Fig 2D and 2F: IL6R is shown in panel D and appears to be expressed in a number of cells, while panel F fails to confirm this in the same cells through qPCR (present in 0% of cells). Gp130 and IL6 should also be included in panel D if not all 0.

Reply: As suggested by several reviewers, we doubled the number of M0-stage patients to obtain more robust data. After batch correction, expression of IL6 and IL6R is not visible. Whether spurious reads that might be detected upon deeper sequencing would be biologically relevant is highly questionable. We think the answer is given by our functional analysis of model cells: although we found very low expression of the IL6R in MCF10A cells, all effect was mediated by trans-signaling. For this reason, we deem it highly unlikely that such rare reads detectable only by deeper sequencing, would be functionally relevant.

Changes: The results are now fully displayed for all IL6 pathway genes, including IL6, IL6R and gp130 (IL6ST). Please, see revised Figure 2f.

Therefore, in conclusion, this study presents a very unbalanced view of the field, and seems to use cherry picking of results by statistically unsound criteria, and it is unclear if their results and conclusions are correct or not.

Reply: We hope that reviewer #2 is satisfied with the changes made and our more balanced presentation of the results.

Reviewer#3: (Expertise: PI3K, CSC, Remarks to the Author) Review of NCOMMS-19-01747A-Z

We thank reviewer #3 for spending time on our manuscript and for his/her careful evaluation. We found the comments very useful to improve the manuscript. We carefully addressed all points raised and hope that we were able to overcome the concerns of the reviewer.

Interleukin-6 trans-signaling is a candidate mechanism to drive progression of human DCCs during periods of clinical latency

Summary

This work is aimed at indentifying the molecular mechanisms that drive the outgrowth of disseminated cancer cells into life-

threatening metastases in the context of breast cancer dissemination to bone. The authors found that gp130-mediated IL6 trans-signaling played critical roles in allowing the stem-like outgrowth of disseminated cells, provided they end up in the vicinity of a favorable niche. Cells harboring oncogenic lesions exhibit less dependency on such signaling.

The work is of interest and relies on a wealth of bioinformatic genomic and transcriptomic data. However, there are serious concerns in regards to the models used, the extrapolations the authors make regarding the effects of the sgp130Fc, as well as the correlative nature of many of the findings, especially those pertaining to PI3K role in the last figure.

Some comments:

1- The fact that primary-tumor-derived cancer cells from M0 patients did not establish xenografts, but those from M1 patients did, suggests that genetic alterations within the primary tumor are necessary for engraftment, albeit not sufficient. This point needs to be made clearer as the text gives the impression that non-cancer-cell-autonomous pathways are both necessary BUT ALSO, sufficient.

Reply: We are not sure whether this comment is based on a misunderstanding: we did not use "primary tumor derived cancer cells" but bone marrow derived disseminated cancer cells. Maybe such a discussion is academic (as we think that also BM-derived DCCs are derived from the primary tumor), however, many data indicate that important mutations are acquired after dissemination (see Werner-Klein Nat Comm 2018). The notion that our text gives the impression "that non-cancer-cell-autonomous pathways are both necessary BUT ALSO, sufficient" was certainly not our intention. On the contrary, we think that the acquisition of constitutive PI3K pathway activation (most likely acquired outside the primary tumor (!)) provides strong evidence that the dependency on non-autonomous signals needs to be overcome.

Changes: We modified the text to clarify this point.

2- With the caveat that clinical tissues are not easy to come by, it would be advisable to increase the numbers of M1 specimens in Table 1. As is, the 50% take rate could be stochastic. Considering the conclusion based on the comparisons between M0 and M1, this deserves a bigger n number.

Reply: We agree with the reviewer that a higher number of M1 patients would be highly desirable. However, diagnostic BM aspirates of M1 patients are almost impossible to get as these patients are very sick and are therefore reluctant to undergo additional procedures for sole research purposes. However, our goal was not to study engraftment of M1 specimens but M0 specimens and M1 specimens just served as a positive, technical control. Therefore, we transplanted BM samples of 42 M0 patients, which we deem to be a sufficiently high number. Although the numbers for M1 samples are small the difference in engraftment is statistically significant ($p = 0.001$; Fisher's exact test).

Changes: To address the concern of reviewer #3, we refrained from stating that 50% of M1 samples engrafted. Instead, we write that 2 out of 4 M1 samples engrafted. We focus on our inability to generate xenografts from M0-stage DCCs and only reference against the M1 data.

3- It is not clear how faithful the hTERT-HME1-EGFRdelta746-750 is. It exhibits IL6 trans-signaling, but undoubtedly so many other pathways (compensatory or not) as well.

Reply: It certainly does. However, if we block IL6 trans-signaling by sgp130Fc, sphere-formation is reduced by more than 50%, which shows that IL6 trans-signaling is relevant for sphere-formation despite the activation of other pathways. Please, see Figure 3f.

4- Does the increase in sphere formation abilities of the different groups in Fig. 3 result in fluctuations in label-retaining cells? This is important to the premise of the studies.

Reply: This is an interesting question, which however, leaves room for two interpretations. First, the reviewer may hold that an increase of sphere formation *ability* should correlate with an increase of LRCs when cells that acquired the ability are used in a label-retention assay. We would think so,

however, this is *not* a premise of our study. If an increase of sphere formation ability is not correlated with an increase of LRCs, none of our conclusions is refuted. We deem important that

- 1) we find in LRCs activation of the IL6 pathway.
- 2) activation of the IL6 pathway increases sphere formation
- 3) this activation results from IL6 trans-signaling.

Although we further show that (i) only LRCs can form spheres and (ii) IL6 trans-signaling stimulates nLRCs to form spheres, our conclusions do not depend on whether or not nLRCs would become LRCs by IL6 trans-signaling. For us the relevant functional finding is that spheres are formed, i.e. that cancer cells may acquire (stem-like) properties needed for metastasis initiation.

A second interpretation of the reviewer's question could be if LRCs are activated by IL6 trans-signaling to form more spheres or if nLRCs acquire sphere forming ability. To address this question of the reviewer, we added to Fig. S3c an additional data set of LRCs. This data set shows that the number of CFSE^{high} cells (LRCs) is not significantly changed after 4 days in response to IL6 stimulation indicating that the increase in spheres is caused by other cells, i.e. nonLRCs, which is confirmed by our experiments shown in Fig. 4d.

Changes: Please, find data in Fig. 4a-d, and Fig. S3c+d, particularly the panel in S3c on LRCs, demonstrating that IL6 trans-signaling confers stem-like ability to non-label retaining cells/non-stem like cells.

5- The relevance of MCF10A co-cultures with MSCs and OB is not convincing. Yes, in these settings, IL6 trans-signaling might be inhibited, but that does not necessarily mean that DCCs per se will exhibit the same reaction. The authors observed that DCCs exhibit lower levels of gp130 than normal cells from reduction mammaplasty, but the comparison should be to non DCC from primary tumors instead.

Reply: Our interpretation of this question is that the reviewer asks whether primary tumor-derived cells (i.e. cells that may be "more" malignant than MCF10A) respond with gp130 down-regulation to co-culture with OBs. We thank the reviewer for this suggestion and tested it.

Changes: We performed additional experiments and demonstrate that also cancer cells can respond to OB signals with gp130 down-regulation. We assessed gp130 expression levels by single cell qPCR in MCF 10A cells and MCF-7 cells as models for DCCs from different stages after culturing them in the presence or absence of MSCs. Fig. 5d shows that co-culture of either MCF 10A or MCF-7 cells with MSCs reduces their gp130 expression to a similar degree. The experiment further shows that the regulation occurs via transcriptional regulation.

6- Do the HME1-shPTEN cells interact differently with MSCs/OB than MCF10A in Figs. 4 and 5?

Reply: We thank the reviewer for pointing out that our analysis of the PI3K-pathway was somehow lacking stringent comparators / controls. To address this point we obtained an isogenic pair of parental and PIK3CA-mutated cells and performed experiments with PIK3CA mutant and wild type cancer cells.

Changes: We obtained an isogenic MCF 10A cell line with a knock-in of the E545^{K/+} mutation in *PIK3CA* and tested whether MSC-induced gp130 down-regulation is influenced by the mutational status of *PIK3CA*. Although we noted a generally lower expression of gp130 in the syngeneic MCF10A pair (likely due to a founder effect during cloning), Fig S4e clearly demonstrates an MSC-induced down-regulation of gp130 cell surface expression for MCF 10A *PIK3CA* E545^{K/+} cells.

Furthermore, to pick up criticism 5 of reviewer #3 (regarding whether MCF10A is representative for cancer cells) we took cancer cell lines with or without an activating mutation of *PIK3CA* and tested

whether sphere-formation in response to IL6 was influenced by the mutational activation of the *PIK3CA*. Intriguingly and in line with the observed increased frequency of *PIK3CA* mutated CTCs in metastatic patients as compared to DCCs from M0-patients, mutational activation of *PIK3CA* renders sphere formation independent from microenvironmental IL6-trans-signaling: whereas IL6 trans-signaling increases sphere-formation only in cells with wild type *PIK3CA*, it does not do so in cells with *PIK3CA* mutation. Moreover, cells with *PIK3CA* mutation do not only not react to IL6 trans-signaling anymore, but show increased sphere-formation in the absence of IL6 trans-signaling (Fig. 6a/b).

7- How about cells in which *PIK3CA* is actually mutated, such as at position 1047? Do those exhibit similar traits to the PTEN sh cells?

Reply: We kindly refer reviewer #3 to our reply to point 6.

8- Gp130 also serves as a LIFR, which was found by Dr. Giaccia and colleagues to play critical roles in regulating dormancy of breast cancer cells in the bone. How is this different than the present study invoking IL6 trans-signaling? The authors do not even mention this prior study at all and at the very least, the prior work needs to be discussed.

Reply: Giaccia et al. shows that in some cell lines (like MCF-7 cells) knock down of LIFR leads to loss of stemness. This is in accordance with our finding that gp130 down-regulation reduces sphere-formation. However, we found that cells with an activating *PIK3CA* mutation (such as MCF-7) are independent from IL6-transsignaling, whereas cells without (such as MDA-MB 231) are still responsive. Of note, Giaccia et al. show in their publication that in contrast to MCF-7 cells (*PIK3CA* mut), bone colonization by MDA-MB-231 cells (*PIK3CA* wt) is independent of the effects of LIFR.

Therefore, our study is not in discordance with theirs, since both studies show that the microenvironment influences the fate of disseminated cancer cells. However, our results also differ in that early DCC mostly lack *PIK3CA* mutations and do not express the LIFR.

Changes: Please, see revised heatmap in Figure 2f, displaying the expression of LIFR.

Reviewer #4 (Remarks to the Author):

We thank the reviewer for his/her critical reading. We are particularly grateful for pointing out gaps in our analysis and several ambiguously phrased claims or conclusions. We hope that we have improved the manuscript by additional experiments and a revised presentation of the data.

Werner-Klein and colleagues investigate the role of IL6 in DCC activation. This is a clinically-relevant and very important question with an impact on breast cancer management. However, at this stage, the manuscript suffers from the following major weaknesses:

(1) What is the evidence that DCCs used for xenotransplantation (Fig 1) are cancer cells? CNV analysis of a few representative ones would help.

Reply: Unfortunately, transplanted DCCs can only be analyzed for copy number alterations if they successfully engraft. For the experiments in Figure 1 we divided the BM aspirate in two halves: one for transplantation, the other for DCC-diagnosis. Due to the rareness of the cells (1-2 DCCs/million BM cells), it is possible that we have DCCs in the injected half, but not in the diagnostic half and *vice versa*. Having DCCs in the injected half, but not in the diagnostic half was the case for a sample in Fig.1a indicated by a star (*).

Changes: Please, see copy number alteration analysis added to Figure 1d. To prove that M0-stage DCCs isolated from BM of transplanted mice in Fig. 1c did not only stain positive for cytokeratin and were of human origin, but are indeed cancer cells with copy number alterations, we successfully conducted copy number alteration analysis for one of the DCCs detected after transplantation and confirmed the cancer origin of the cell.

(2) The number of LRCs (n=8), nLRC (n=5) and QSC (n=10) is extremely low for a single cell sequencing (or microarray) experiment, leaving too much room for technical biases to interfere with the real signals (e.g. dropouts, cell cycle phase etc.). This is confirmed by the fact that the pathway analysis reveals a number of pathways that appear to be enriched but that are completely ignored by the authors (Fig 2c). Also, data quality is not described (e.g. how many transcripts/genes are detected per cell on average?). Selection of IL6 seems more like a cherry-picking exercise rather than an unbiased/result-driven conclusion. A much higher number of cells for each category would help to gain confidence in the results.

Reply and changes: For changes in pathway analyses please see replies to Reviewer #2. Data quality is now described in terms of average number of read counts and genes expressed.

Furthermore, to address the reviewer's wish for higher numbers, we doubled the number of M0-stage DCCs, which were retrieved from the investigation of 246 bone marrow samples. We kindly refer the reviewer to the revised Figure S1a.

(3) Linked to the above. Even when confirmed with a higher cell number, the finding of IL6 being involved in mammosphere formation is not novel. The authors themselves point out "IL6 signaling had been previously found relevant for mammosphere-formation from ductal breast carcinoma and normal mammary gland²³".

Reply: We fully agree and therefore quoted the paper. The literature (as always) served as important starting point of our study. However, the mechanism of IL6 signaling in the context of mammosphere formation had not been studied so far. We differentiated between classical and trans-signaling because the type of signaling is fundamental for the understanding of cancer progression. In contrast to classical signaling, which relies on the expression of IL-6R on target cells, trans-signaling may act on all cells due to the ubiquitous expression of gp130. Importantly, in trans-signaling cancer cells are not autonomous, but dependent on their microenvironment, whereas previous literature suggested that the pathway might be activated by a classical cell-autonomous autocrine loop. We now show that IL6 trans-signaling controls stemness of mammary epithelial cells. In addition, it was not known before that the responsiveness to trans-signaling components (IL6 and sIL6R) from the microenvironment is regulated by the microenvironment itself. We show that neighboring "metastatic niche" cells (MSCs and OBs, but not endothelial cells) are able to regulate the signal transducing unit gp130 of the DCCs. Last but not least, we show that IL6-downstream signaling events are activated in ex vivo isolated DCCs and that acquisition of PIK3CA mutations is one way into cellular autonomy from the microenvironment.

(4) IL6 pathway activation in M1-DCCs as opposed to M0 is unconvincing. Most of the IL6 signature genes are lowly expressed in M1 samples, with only some very limited subset of these genes displaying a higher expression. This is not enough to conclude that IL6 pathway is active. Also, data quality is not described: can the authors exclude that M1 cells have a higher number of detected transcripts/better quality compared to M0?

Reply: Obviously, this is a misunderstanding. At no point of the manuscript we made any claim on differences regarding the IL6 pathway activation between M0 and M1 stage cells. On the contrary, we noted that PIK3CA mutations could also do the job and render cancer cells independent from IL6 trans-signaling.

Changes: To address the concerns of the reviewer, we kindly refer reviewer #4 to our revised pathway analysis as depicted in Figure 2. We tested all pathways that were significantly enriched regarding differential gene expression comparing LRCs and nLRCs/QSCs for their enrichment in M0- and M1-DCCs using single sample pathway enrichment analysis on the RNAseq data (see Methods). We confirm that in contrast to the other pathways tested, the IL6 pathway is significantly enriched in

M0 and M1-DCCs. Furthermore, we refer the reviewer to the novel experiments in Figures 5 and 6 where the link between PI3K pathway activation as found in Figure 2g is functionally explored.

(5) The functional testing of IL6 role for DCC is entirely done in non-neoplastic cells (e.g. HMECs and MCF10A). No direct evidence is provided to support a role of IL6 in DCC progression.

Reply: We thank the reviewer for the suggestion to add evidence for a role of IL6 in breast cancer cells. We now included in vitro and in vivo data of cancer cells.

Changes: First, we pre-treated tumorigenic MDA-MB-231 cells with and without HIL6 for a short time, injected the respectively same number of cells into mice and monitored tumor growth. We found an increased sphere number in response to HIL6 and revealed that HIL6-pre-treated cells form significantly larger tumors than cells pre-treated with PBS or an anti-IL6 antibody. These data are displayed in Figure 4g and h. Furthermore, we tested cancer cells for responsiveness to IL6 trans-signaling and gp130 regulation. These data are shown in Figures 5 and 6.

(6) Experiment in Fig. 6c is unclear. How many cells were analyzed and for how many of these a CNV profiling was obtained (i.e. confirmation of cancer origin).

Reply: For CTC analysis we used the FDA-approved CellSearch assay. The cells were also tested for copy number alterations (see Polzer Embo Mol Med 2014). For the BM-DCCs we performed copy number alteration or LOH analysis (data are shown in Schmidt-Kittler, PNAS 2003, Schardt, Cancer Cell 2005 and Hosseini, Nature 2016). Based on the concerns of reviewer 1 (comment 6)), we focus in the revised version of the paper on HR+ breast cancer.

Changes: We added the notion that the cells were tested for copy number alterations in previous publications.

The table refers to patients and not cells. This is important to make sure that comparisons are done in homogeneous groups.

Reply and changes: We added the information about mutated/non-mutated cells.

I disagree with the statement “These data are fully consistent with a scenario in which early DCCs depend on IL6 trans-signaling and become increasingly independent during cancer evolution.” Rather, current data on DCCs in figure 6 simply shows that some of them may harbor a PIK3CA mutation. Metastasis-derived CTCs are not representative of early DCC events.

Reply: This is a misunderstanding that is very important to clarify. We use CTCs as a **contrast to** the early DCCs. All CTCs are derived from metastases as the patients had resection of their primary tumors years before. All patients included are HR-positive breast cancer patients. Therefore, we interpret the significantly increased PIK3CA mutations in metastatic CTCs as opposed to non-metastatic M0-stage DCCs as evidence for selection of PIK3CA mutations. We think that this conclusion is justified as with the resection of primary tumors a potential source of PIK3CA mutated cells is removed and M0-stage DCCs become representative for early systemic cancer and CTCs representative for advanced systemic cancer.

Consistent with this evolutionary model are our functional experiments showing that responsiveness (i.e. gain of stem-like ability) to IL6 trans-signaling is dependent on the *PIK3CA* mutational status. Whereas pre-malignant and malignant cells with non-mutated PIK3CA respond to IL6 trans-signaling with increased sphere-formation, PIK3CA mutated cells do not, but display an intrinsically increased sphere-forming ability.

Changes: We clarified this reasoning in the discussion.

Minor:

(1) Fig1a refers to “engrafted patients”. Aren’t these mice instead?

Reply and changes: The reviewer is right. We apologize and corrected the wording.

Reviewers' comments:

Reviewer #1 (Remarks to the Author):

The authors have addressed my comments by either revising the manuscript or providing a reasonable explanation as to why studies have been limited in some cases. Overall I am satisfied with the changes they have made in relation to the points I raised and I have no further comments to this interesting manuscript.

Reviewer #2 (Remarks to the Author):

This reviewer appreciates the authors have made considerable efforts to improve the manuscript. Statistical analyses are amended or replaced and appear more robust in the current version. However, the main critique of reviewers 2, 3 and 4 remains unresolved. That is, the evidence linking the DCCs used for transplantation to the patient's primary tumor, remains very weak. Over the last decade, several studies have provided evidence for the presence of somatic copy number changes, and even catastrophic events such as chromothripsis, in physiologically normal tissues of both healthy and diseased individuals (e.g. doi: 10.1038/nature11629, 10.1038/nature09807, 10.1038/ng.3641, 10.1016/j.ajhg.2016.02.003, 10.1038/s41587-019-0297-6, 10.1038/s41586-018-0321-x, 10.1126/science.aaa6806, 10.1126/science.aau3879, 10.1038/s41586-018-0811-x, 10.1038/s41586-020-1961-1). As a result, the inference of a direct link between primary tumor and CK/EPCAM+ cells in the bone marrow, purely based on the presence of random copy number aberration(s) (the authors provide a single example profile of each type and a summary of the other cells in Supplementary Fig. 1b), has become untenable. As the selection and correct classification of these cells underpins all of the downstream analyses and interpretation, this reviewer remains unconvinced of the findings presented in this work.

Reviewer #3 (Remarks to the Author):

The authors have largely addressed the comments of the 4 reviewers with new experiments, new analyses, and new discussion. As a result, the manuscript has indeed improved in quality.

Three points remain in my mind that would be advisable/desirable to address. They are not an absolute must, and are hence stated more to the attention of the editor than the authors.

#1: The authors discuss IL6 signaling and PI3K signaling as regulators of DCC dormancy/growth, but they do not show actual downstream signaling protein activation status, e.g., JAK/STAT, SOCS, MAPK, AKT, etc. It is assumed that such signaling nodes are affected downstream of gp130 and PIK3CA deregulation, but this has not been demonstrated.

#2: The authors focus on HR+ breast cancer, but they employ suboptimal models: MCF7 cells which are not known to metastasize to (mouse) bone and MDA MB-231 cells which are TNBC cells.

#3: There is no evidence provided that the DCCs transit from microenvironmental dependence to microenvironmental independence VIA PIK3CA activation in vivo. What the authors offer are correlative models and observations. The essentiality of PIK3CA to advanced DCC/CTC metastatic growth has also not been established.

Reviewer #4 (Remarks to the Author):

This Reviewer appreciates the significant efforts the authors made to answer critical questions raised by all Reviewers. The manuscript has certainly improved. Minor points should be considered before acceptance:

(1) New figure 1d: This data is very reassuring, yet single cell CNV is notoriously prone to interpretation errors, mostly given by uneven coverage of the sequenced cell, etc.. While I do not doubt the result obtained with a representative DCC, it would be good to see a CNV analysis of one (or more) non-neoplastic single cells as control.

(2) Experiment with MDA-MB-231 treated with IL6. The authors are certainly aware that bigger tumor size does not necessarily mean "conversion of progenitor into stem-like cells" like stated in their paragraph title. Authors should exclude differences in proliferation and apoptosis rate during tumor growth, at minimum.

(3) Linked to the point above: it would be important to confirm the results (cancer cells treated with IL6, then injected in mice) with an independent tumor model, possibly a HR-positive model for consistency with the rest of the manuscript.

Reviewer #2 (Remarks to the Author):

This reviewer appreciates the authors have made considerable efforts to improve the manuscript. Statistical analyses are amended or replaced and appear more robust in the current version. However, the main critique of reviewers 2, 3 and 4 remains unresolved. That is, the evidence linking the DCCs used for transplantation to the patient's primary tumor, remains very weak. Over the last decade, several studies have provided evidence for the presence of somatic copy number changes, and even catastrophic events such as chromothripsis, in physiologically normal tissues of both healthy and diseased individuals (e.g. doi: 10.1038/nature11629, 10.1038/nature09807, 10.1038/ng.3641, 10.1016/j.ajhg.2016.02.003, 10.1038/s41587-019-0297-6, 10.1038/s41586-018-0321-x, 10.1126/science.aaa6806, 10.1126/science.aau3879, 10.1038/s41586-018-0811-x, 10.1038/s41586-020-1961-1). As a result, the inference of a direct link between primary tumor and CK/EPCAM+ cells in the bone marrow, purely based on the presence of random copy number aberration(s) (the authors provide a single example profile of each type and a summary of the other cells in Supplementary Fig. 1b), has become untenable. As the selection and correct classification of these cells underpins all of the downstream analyses and interpretation, this reviewer remains unconvinced of the findings presented in this work.

Reply: We thank the reviewer for her/his reflections on an important aspect of our study. We fully agree with the reviewer that:

- 1) identification of DCCs is fundamental for our study and the conclusions.
- 2) distinction of our candidate DCCs from confounding normal cells is essential.

Ad point 1: Identification of DCCs in bone marrow

We follow the reasoning that **epithelial cell identity in bone marrow plus the presence of genetic alterations is sufficient to claim malignant epithelial origin and malignant potential of a cell**. Dissemination from a solid organ is **the** defining criterion of malignancy for an aberrant, proliferative cell and decades of research demonstrated that there are no epithelial cells in bone marrow. We do **not** hold that the putative cancer cell must be genetically identical to the primary tumor. This would be against all pathological experience and practice and violates principles of evolution. For example, consider cancer of unknown primary where no primary tumour can be found, proving that DCCs disseminated early and developed genetically independently from the "primary". Intratumoral heterogeneity is so widespread that the request of a clearly defined genetic similarity appears in our opinion unjustifiable. It raises several questions that depend on numerous assumptions that currently cannot be decided. For example, which number or percentage of genetic alterations present in the primary tumor and the DCCs would suffice to fulfill the request of similarity? Would genetic alterations shared with a subclone or even a super-subclone suffice or do only events count that are shared with the predominant clone? Is an early separation and therefore a smaller degree of similarity less biologically important than a late? Do we truly understand the role of early (i.e. less similar) vs. late (potentially more similar) DCCs to exclude the relevance of either one? To our knowledge all mouse models published so far demonstrate that early dissemination and generation of metastases from early DCCs is frequent, if not predominant¹⁻⁵. Sparse snapshot data of human primary tumour – metastasis pairs fall short to be a definite proof that cancer cells disseminate late from the primary tumor. In fact, at least 20-60% of primary tumour – metastasis pairs are very heterogeneous and in conflict with a simple linear progression model⁶⁻¹⁶. A definition that a DCC is only a DCC if it is similar to the primary tumor is not justified by any data available so far, but requires more scientific studies on patient samples. Studies, which require correct identification of DCCs.

Therefore, we agree with reviewer #2 that identification of a DCC is essential, but not that genetic similarity with the primary tumour is a *sine-qua-non*.

The reviewer quoted a single study that questions the suitability of epithelial markers for the identification of DCCs in bone marrow (Demeulemeester, Genome Biology 2017). However, a closer look into that paper shows that the authors did not analyze epithelial-marker positive cells, but MOPC21-positive (i.e. isotype control-positive) cells. There is a long-standing consensus that patient samples that react with the isotype control (the antibody is directed against mineral oil (!)) must be excluded, however for unknown reasons this was not done in the Demeulemeester study. Therefore,

this study cannot determine the phenotype of any cell, i.e. whether the epithelial marker was expressed or not, not even for cells that appeared to react positively with the cytokeratin-antibody. Surprisingly, Demeulemeester et al. took mineral-oil-antigen-positive cells as proof against cytokeratin-positive cells being early disseminated cancer cells from the primary tumour. We do not follow this reasoning. **For us, epithelial identity in bone marrow remains a hallmark for a DCC.** We therefore addressed the point of epithelial identity linking putative DCCs with the primary tumor in one additional experiment:

Since we study EpCAM-positive cells from bone marrow of cancer patients, we isolated EpCAM-positive cells from the bone marrow of healthy donors, i.e. patients without a known cancerous disease. In bone marrow of non-cancer patients rare EpCAM+ cells can be found and our own data and the work of Cackowski, 2019 suggest that these cells belong to the B cell lineage.

As for EpCAM+ cells isolated from the BM of cancer patients, we performed RNA-seq analysis. In some cases, we also succeeded to isolate the DNA of EpCAM+ cells from BM of healthy donors, which revealed a balanced CNA profile (Supplementary Fig. 1b). More importantly, we then compared the expression profiles of EpCAM+ cells from non-cancer patients with those obtained from EpCAM-positive/CNA-displaying cells from breast cancer patients. We performed PCA analysis using all variable genes or variable genes in the GO categories (gene ontology terms) "B cell", "Epithelial" or "Epithelium". We show a clear separation based on these sets of genes. **We conclude that EpCAM-positive cells from breast cancer patients that harbor CNAs are epithelium-derived non-autochthonous cells, whereas EpCAM-positive cells from non-cancer patients are autochthonous bone marrow cells of the B-cell lineage.** Of particular importance for the discussion below (reviewer #3 and #4) is that the marker for mammary luminal progenitor cells, *KIT*, is highly expressed in early stage MO-DCCs. **Therefore, we provide a direct link of the analyzed cells to the tissue of origin (mammary gland) and the target cell of the IL6 signaling that is the focus of the remaining manuscript.**

Ad point 2: Distinction of our candidate DCCs from confounding normal cells.

EpCAM/CNA-double positive cells clearly differ from normal bone marrow inhabitants by gene expression. However, how relevant is the presence of CNAs to identify cancer cells? In our study, the second hallmark for the identification of DCCs is chromosomal instability (CIN). Here, the reviewer correctly points out that occasionally chromosomal changes have been identified in normal cells. However, looking into the papers provided by the reviewer, we would re-iterate our conclusion that CIN is indeed a hallmark of DCCs.

Evaluation of the Reviewer's cited literature

First author + year	DOI
Abyzov 2012	10.1038/nature11629
Yoshida 2020	10.1038/s41586-020-1961-1
Navin 2015	10.1038/nature09807
Jakubek 2020	10.1038/s41587-019-0297-6
Martincorena 2015	10.1126/science.aaa6806
Vattathil 2016	10.1016/j.ajhg.2016.02.003
Loh 2018	10.1038/s41586-018-0321-x
Martincorena 2018	10.1126/science.aau3879
Yokoyama 2019	10.1038/s41586-018-0811-x

- a) Checking the papers quoted by the reviewers, we first noted that several studies are not informative for the question addressed. We conclude this, because it is not clear whether the tested cells are at all suited control cells.

- We analyzed CNAs in DCCs that were not cultured, therefore studies analyzing cells that had been cultured prior to CNA analysis are unsuited controls, as culture-conditions are known to change genotypes. This affects the papers: *Abyzov, 2012* and *Yoshida, 2020*.
- Moreover, data on CNA analysis of tumor-adjacent tissue (*Jakubek et al. 2020*) are also not informative in the context of our paper, as it is unclear to which extent “field-effects” contribute to the genotype of cells, i.e. a field of “injury” in tissues surrounding tumors at cancer sites. Tumor-adjacent tissues might be morphological normally appearing, however, phenomena such as field cancerization, morphological reversion and morphological mimicry are widespread and well known. For example, the study found that tumor-adjacent tissue harbors more CNAs than blood cells and CNAs of tumor-adjacent tissue are significantly more concordant with CNAs of the PT than CNAs of blood and PT-samples (*Jakubek, 2020*). Also, the study of *Martincorena, 2015* on sun-exposed skin has to be interpreted with care, as these samples were exposed to a carcinogen (UV-light) and displayed a UV-signature. EpCAM-positive cells in bone marrow were not.
- The study of Navin 2015 employs single cell nucleus sorting, however does not differentiate between nuclei from non-tumor vs. tumor cells.

b) However, some studies are indeed informative for the question posed by the reviewer.

Studies analyzing CNAs in healthy tissues (*Jakubek, Vattathil, Loh, Martincorena*, see below) found them to occur in less than 5% of samples (Median = 1.85 %, min. to max.= 0% to 8.5%). This is in striking contrast to the frequency of CNAs detected in DCCs of our study (M0 =50%, M1=90%).

- *Jakubek, 2020*: CNAs in blood cells= median 1.8% of cases (n=7756 samples, 28 cancer-types)
- *Vattathil, 2016*: CNA in blood + buccal swab cells =2.9% of cases (n=31100 samples)
- *Loh, 2018*: CNV in blood cells =5.5% of cases, cell fraction affected <5% (151202 patients)
- *Martincorena, 2018*: the authors states explicitly “*The rarity of copy number changes in large clones... suggests that the background rate of copy number changes is low in normal cells of the esophagus or that such changes are negatively selected. Either way, this represents a major difference between normal esophageal cells and ESCCs, suggesting that structural changes may occur late in the evolution of esophageal cancers.*”
- *Yokoyama, 2019*: CNAs in carcinogen-exposed esophageal tissue: 7-9% with gains/losses; n=188 samples).
- In addition, in a diagnostic setting (to differentiate lymph node nevus cells from true melanomas), CIN was found to be the most reliable indicator of malignancy (10.1111/j.1529-8019.2005.00055.x).

In summary, CIN in normal (non-carcinogen exposed) cells is rare and displayed in less than 5%. In our experience this is the upper limit. We published a series of 50 normal hematopoietic/lymphatic cells analyzed with our single cell technique and found all cells to display a normal karyotype (10.1038/s41467-017-02674-y). Therefore, the significantly higher rate of CIN in EpCAM-positive cells (50-90%) from bone marrow of breast cancer cells strongly supports the notion that these cells are cancer-derived.

In conclusion, we provide a compelling evidence that EpCAM/CNA-double positive cells isolated from bone marrow are indeed DCCs. The last piece of doubt whether they may be of other epithelial origin was removed by checking all patient data for a history of a secondary carcinoma. **None was reported. Therefore, the presence of a diagnosed breast cancer justifies categorization of the included cells as breast cancer DCCs.**

Reviewer #3 (Remarks to the Author):

The authors have largely addressed the comments of the 4 reviewers with new experiments, new analyses, and new discussion. As a result, the manuscript has indeed improved in quality.

Three points remain in my mind that would be advisable/desirable to address. They are not an absolute must, and are hence stated more to the attention of the editor than the authors.

#1: The authors discuss IL6 signaling and PI3K signaling as regulators of DCC dormancy/growth, but they do not show actual downstream signaling protein activation status, e.g., JAK/STAT, SOCS, MAPK, AKT, etc. It is assumed that such signaling nodes are affected downstream of gp130 and PIK3CA deregulation, but this has not been demonstrated.

Reply: We have addressed the points raised and performed western blot analyses of MCF 10A (wt and *PIK3CA* mutated) stimulated with HIL6 (Fig. 7b). These analyses show that adding HIL6 activates the JAK/STAT, MAP and AKT pathway in MCF 10A wt cells. We also show that MCF 10A cells with an activating *PI3KCA* mutation display an activated MAPK and AKT pathway activated in absence of IL6.

#2: The authors focus on HR+ breast cancer, but they employ suboptimal models: MCF7 cells which are not known to metastasize to (mouse) bone and MDA MB-231 cells which are TNBC cells.

Reply: This is a misunderstanding which also applies to reviewer #4. **We are not focusing on HR+ vs TN-breast cancer. Our focus is on luminal progenitor-derived cancers**^{17,18}. Both basal like (e.g. MDA-MB-231) breast cancers as well as HR+-BC (MCF-7) are derived from luminal progenitor cells. We model these by using the normal mammary cells from reduction mammoplasties and MCF 10A cells. We show that HIL6 is conferring mammosphere formation and luminal cell differentiation ability to the nLRCs progenitor cells. These cells do not have myoepithelial ability and therefore, we do not claim that HIL6 would also work on mammary stem cell derived cancers (MaSC), such as claudin-low TN-breast cancers. However, our patient cohort did not comprise these rare cancers. **To strengthen this point, we now show the expression of *KIT* as marker for luminal progenitor cells (Fig. 3b; reference¹⁹ below).**

The reviewer is correct with the statement that MCF-7 does not metastasize in mice. However, the cell line was derived from a pleural metastasis and therefore is a metastatic cancer regardless of whether or not the artificial transplantation into mice “validates” its metastatic ability. In fact, this is rather an argument against the relevance of mouse models than against our selection of the cell line.

#3: There is no evidence provided that the DCCs transit from microenvironmental dependence to microenvironmental independence VIA PIK3CA activation in vivo. What the authors offer are correlative models and observations. The essentiality of PIK3CA to advanced DCC/CTC metastatic growth has also not been established.

Reply: We apologize for the misunderstanding. It was not our intention at any point to claim that *PIK3CA* mutations is essential. There may be certainly other ways to generate metastasis without IL6/*PIK3CA* involvement. However, we provide functional mechanistic data demonstrating interdependence of *PIK3CA* mutations and responsiveness to HIL6. We further note that early DCCs differ from late M1-stage DCCs with regard to *PIK3CA* mutations. We agree with the reviewer that the term “microenvironmental dependence” is too broad. What we wanted to express is dependence on IL6 signaling. As IL6 signaling is important in bone marrow, we generalized. We will correct our wording.

Reviewer #4 (Remarks to the Author):

This Reviewer appreciates the significant efforts the authors made to answer critical questions raised by all Reviewers. The manuscript has certainly improved. Minor points should be considered before acceptance:

(1) New figure 1d: This data is very reassuring, yet single cell CNV is notoriously prone to interpretation errors, mostly given by uneven coverage of the sequenced cell, etc.. While I do not doubt the result obtained with a representative DCC, it would be good to see a CNV analysis of one (or more) non-neoplastic single cells as control.

Reply: We already published 50 profiles of control cells that were identically processed as the cells in Figure 1d. Please, see 10.1038/s41467-017-02674-y. However, we added more profiles to Fig. 1d and Supplementary Fig. 1b.

(2) Experiment with MDA-MB-231 treated with IL6. The authors are certainly aware that bigger tumor size does not necessarily mean “conversion of progenitor into stem-like cells” like stated in their paragraph title. Authors should exclude differences in proliferation and apoptosis rate during tumor growth, at minimum.

Reply: The claim that we convert progenitor cells into mammosphere forming cells is derived from the normal cells. In case of breast cancer, it is probably more correct to state that metastatic colony-formation is increased. This is how we interpret the data.

As requested by the reviewer we analyzed the tumors of the three experimental groups for Ki-67 and caspase 3 expression (Fig. 5i). Using TissueFAX-based cytometric quantification we could exclude that differences in proliferation and apoptosis rates during tumor growth accounted for the observed difference in tumor volume.

(3) Linked to the point above: it would be important to confirm the results (cancer cells treated with IL6, then injected in mice) with an independent tumor model, possibly a HR-positive model for consistency with the rest of the manuscript.

Reply: Obviously, we should point out more explicitly **that we focus on luminal progenitor-derived cancers**^{17,18}. **To this group belong HR-positive cancers and TN-cancers that display chromosomal instability (CIN) and copy number alterations (CIN-positive cancers)**^{18,20}. We show that HIL6 is acting on the population that is unable to generate myoepithelial cells, i.e. nLRC. Mammary stem cell-derived (MaSC) cancers like the non-CIN claudin-low cancers may react differently upon IL6 stimulation. However, since we did not have any patient falling in this group (**EpCAM-positive DCCs had to display CNAs to be included**), we are not addressing EpCAM-negative-MaSC-derived cancers. We show *in vitro* that MCF10A and MDA-231 react similarly. By having demonstrated that the effect is on the shared luminal progenitor cell, the additional animal experiment is obsolete because the CIN-positive MDA-MB-231 is representative for a luminal-progenitor derived cancer. For this reason, we would not even know how we could justify this experiment to obtain permission for additional animal experiments from the government. **More importantly, the strong expression of KIT (see ref¹⁹ and Fig. 3b) by DCCs identifies them as luminal progenitors.** We changed the text accordingly and added a short discussion on the application of our data to TN and HR breast cancers and the limitations related to claudin-low breast cancers.

References:

- 1 Eyles, J. *et al.* Tumor cells disseminate early, but immunosurveillance limits metastatic outgrowth, in a mouse model of melanoma. *The Journal of clinical investigation* **120**, 2030-2039 (2010).
- 2 Husemann, Y. *et al.* Systemic spread is an early step in breast cancer. *Cancer cell* **13**, 58-68, doi:10.1016/j.ccr.2007.12.003 (2008).

- 3 McCreery, M. Q. *et al.* Evolution of metastasis revealed by mutational landscapes of chemically induced skin
cancers. *Nature medicine* **21**, 1514-1520, doi:10.1038/nm.3979 (2015).
- 4 Rhim, A. D. *et al.* EMT and dissemination precede pancreatic tumor formation. *Cell* **148**, 349-361,
doi:10.1016/j.cell.2011.11.025 (2012).
- 5 Rhim, A. D. *et al.* Detection of circulating pancreas epithelial cells in patients with pancreatic cystic lesions.
Gastroenterology **146**, 647-651, doi:10.1053/j.gastro.2013.12.007 (2014).
- 6 Gundem, G. *et al.* The evolutionary history of lethal metastatic prostate cancer. *Nature* **520**, 353-357,
doi:10.1038/nature14347 (2015).
- 7 Hong, M. K. *et al.* Tracking the origins and drivers of subclonal metastatic expansion in prostate cancer. *Nat*
Commun **6**, 6605, doi:10.1038/ncomms7605 (2015).
- 8 Hu, Z. *et al.* Quantitative evidence for early metastatic seeding in colorectal cancer. *Nat Genet* **51**, 1113-1122,
doi:10.1038/s41588-019-0423-x (2019).
- 9 Ishaque, N. *et al.* Whole genome sequencing puts forward hypotheses on metastasis evolution and therapy in
colorectal cancer. *Nat Commun* **9**, 4782, doi:10.1038/s41467-018-07041-z (2018).
- 10 Kim, T. M. *et al.* Subclonal Genomic Architectures of Primary and Metastatic Colorectal Cancer Based on
Intratumoral Genetic Heterogeneity. *Clinical cancer research : an official journal of the American Association for*
Cancer Research **21**, 4461-4472, doi:10.1158/1078-0432.CCR-14-2413 (2015).
- 11 Kroigard, A. B. *et al.* Genomic Analyses of Breast Cancer Progression Reveal Distinct Routes of Metastasis
Emergence. *Sci Rep* **7**, 43813, doi:10.1038/srep43813 (2017).
- 12 Leung, M. L. *et al.* Single-cell DNA sequencing reveals a late-dissemination model in metastatic colorectal cancer.
Genome Res **27**, 1287-1299, doi:10.1101/gr.209973.116 (2017).
- 13 Naxerova, K. & Jain, R. K. Using tumour phylogenetics to identify the roots of metastasis in humans. *Nat Rev Clin*
Oncol **12**, 258-272, doi:10.1038/nrclinonc.2014.238 (2015).
- 14 Schrijver, W. *et al.* Mutation Profiling of Key Cancer Genes in Primary Breast Cancers and Their Distant
Metastases. *Cancer research* **78**, 3112-3121, doi:10.1158/0008-5472.CAN-17-2310 (2018).
- 15 Ullah, I. *et al.* Evolutionary history of metastatic breast cancer reveals minimal seeding from axillary lymph nodes.
The Journal of clinical investigation **128**, 1355-1370, doi:10.1172/JCI96149 (2018).
- 16 Yates, L. R. *et al.* Genomic Evolution of Breast Cancer Metastasis and Relapse. *Cancer cell* **32**, 169-184 e167,
doi:10.1016/j.ccell.2017.07.005 (2017).
- 17 Keller, P. J. *et al.* Defining the cellular precursors to human breast cancer. *Proceedings of the National Academy of*
Sciences of the United States of America **109**, 2772-2777, doi:10.1073/pnas.1017626108 (2012).
- 18 Morel, A. P. *et al.* A stemness-related ZEB1-MSRB3 axis governs cellular pliancy and breast cancer genome
stability. *Nature medicine* **23**, 568-578, doi:10.1038/nm.4323 (2017).
- 19 Lim, E. *et al.* Aberrant luminal progenitors as the candidate target population for basal tumor development in
BRCA1 mutation carriers. *Nature medicine* **15**, 907-913, doi:10.1038/nm.2000 (2009).
- 20 Curtis, C. *et al.* The genomic and transcriptomic architecture of 2,000 breast tumours reveals novel subgroups.
Nature **486**, 346-352, doi:10.1038/nature10983 (2012).

REVIEWERS' COMMENTS:

Reviewer #2 (Remarks to the Author):

The authors have further revised their manuscript in response to the reviewer comments. I specifically appreciate that the authors have clarified their reasoning on the identity of the presumed DCC's they isolate from patients' bone marrow. However, in the light of the large amount of somatic variation emerging in normal tissues, and the available genomics data on primary/metastasis pairs which consistently shows a clear common origin in the form of shared mutations and copy number changes, I remain unconvinced that all rare cells picked by expressing EpCAM and showing (any) genetic alterations are necessarily related to a primary breast cancer. This is an important caveat of the study.

The authors very state (line 151-153): "We followed the reasoning that epithelial cell identity in bone marrow plus the presence of genetic alterations is sufficient to claim malignant epithelial origin or malignant potential of a cell.". The authors should come back to that in a short paragraph in the discussion clearly stating that this is a hypothesis that has been contested in the literature.

Given the above, I would be very much in favour of the reviewer comments (and the authors' replies) being published alongside the manuscript (an option I understand the authors will be given later on), such that both sides of the argument are available to the interested reader.

A second main point is regarding data availability. The authors should make the RNA sequencing data for all sequenced cells, as well as the raw copy number profiling data for all cells available to the community. At present, the authors' data availability statement simply says "All genomic results of this study are available within the article and its Supplementary Information. This is not sufficient (and not entirely true) – the raw data should be deposited in public databases such as GEO or EGA.

Particularly in the light of the point above, these data should be made available to the scientific community.

Reviewer #3 (Remarks to the Author):

The authors have addressed the overwhelming majority of the comments raised during the review rounds, and this reviewer is generally satisfied.

Reviewer #4 (Remarks to the Author):

The authors provided answers to some of the concerns raised by the reviewers. These answers are only partially satisfactory and weaknesses still remain. I provide some comments below with the intent to give the authors some constructive criticism and to help the editors with their final decision.

(1) Statements on KIT expression are hard to “decipher”. First of all, the scalebar in Fig.3b is unbalanced (only 0 is blue, while 3, 6, 9 are fully red) and given the way it is done, prone to visually highlight possibly minor differences. It is also unclear what exactly are the values 0, 3, 6, 9 referring to: if these are TPM, all should be considered very low expression levels, particularly in light of obvious biases due to single cell gene expression profiles (cell cycle etc etc). All in all, certainly statements such as “strong expression of the luminal progenitor marker KIT...” do not seem to be supported by data at this stage.

(2) My previous questions in regard to the in vivo findings remain unanswered. The authors keep insisting that only one cell line is sufficient: the argument of luminal progenitor-derived is totally irrelevant in this context, and sentences like “we would not even know how we could justify this experiment to obtain permission for additional animal experiments from the government” are hardly understandable. I remain of the opinion that they should exclude cell-line dependent effects for their in vivo experiments, and add additional models to improve their manuscript. Further, in figure 5 (despite adventurous titles) I still see very little stem-like abilities: asymmetric cell division (self-renewal + differentiation), tumor seeding ability in limiting dilution assays, ability to recreate a heterogeneous lesion from a single cell are required assays for statements of this kind, and none of this is presented. Tumorsphere size and CD44/24 are surrogate assays with very questionable validity, and tumor volume has really nothing to do with stem-like traits.

Universität Regensburg

FAKULTÄT FÜR MEDIZIN

Lehrstuhl für Experimentelle
Medizin und Therapieverfahren

Prof. Dr. C. Klein Franz-Josef-Strauß-Allee 11 93053

Prof. Dr. Christoph Klein
Telefon +49 941 944-6720
Telefax +49 941 944-6719

Franz-Josef-Strauß-Allee 11
D-93053 Regensburg

christoph.klein@klinik.uni-regensburg.de
www.uni-regensburg.de

Reviewer Reports:

Reviewer#1:

The authors have further revised their manuscript in response to the reviewer comments. I specifically appreciate that the authors have clarified their reasoning on the identity of the presumed DCC's they isolate from patients' bone marrow. However, in the light of the large amount of somatic variation emerging in normal tissues, and the available genomics data on primary/metastasis pairs which consistently shows a clear common origin in the form of shared mutations and copy number changes, I remain unconvinced that all rare cells picked by expressing EpCAM and showing (any) genetic alterations are necessarily related to a primary breast cancer. This is an important caveat of the study.

Reply: As before, we agree with the reviewer's opinion that the primary/metastasis pair data often revealed a common origin. However, also as argued before, there is no evidence of a hardwired-sequence of events or a minimum of shared alterations that would allow to easily determine the malignant identity or origin genetically. Cancer cell heterogeneity has proven to be extensive with some metastasis being very similar and others almost completely different.

The authors very state (line 151-153): "We followed the reasoning that epithelial cell identity in bone marrow plus the presence of genetic alterations is sufficient to claim malignant epithelial origin or malignant potential of a cell.". The authors should come back to that in a short paragraph in the discussion clearly stating that this is a hypothesis that has been contested in the literature.

Reply: We respect and share the skepticism of the reviewer and will add our reasoning and a short paragraph in the discussion as suggested. However, although working since more than two decades in the field, I am not aware of any article that contested the hypothesis that cells with an epithelial phenotype and concomitant genetic alterations that are isolated from a patient with known carcinoma are bona fide cancer cells. To quote such a paper, I would like to ask the reviewer for help and providing the reference.

We added the following sentence to the discussion: "We focused our analysis on EpCAM-positive cells from bone marrow. We followed the rationale that cells with an epithelial phenotype and concomitant genetic alterations that are isolated from a patient with known carcinoma are *bona fide* cancer cells. We are aware that this does not provide a formal proof of their origin from the individual cancer. However, given the enormous intra-tumoral heterogeneity not only of the cancer at diagnosis but also of its complete evolutionary history, such a formal proof may be difficult to establish."

Given the above, I would be very much in favour of the reviewer comments (and the authors' replies) being published alongside the manuscript (an option I understand the authors will be given later on), such that both sides of the argument are available to the interested reader.

Reply: We have no objections and appreciate this suggestion.

A second main point is regarding data availability. The authors should make the RNA sequencing data for all sequenced cells, as well as the raw copy number profiling data for all cells available to the community. At present, the authors' data

availability statement simply says “All genomic results of this study are available within the article and its Supplementary Information. This is not sufficient (and not entirely true) – the raw data should be deposited in public databases such as GEO or EGA. Particularly in the light of the point above, these data should be made available to the scientific community.

Reply: We will make all data publicly available as permitted by law, meaning we are not allowed to place patient sequencing reads into databases as sequencing reads are considered personal data – at least in Germany. As it has been shown that individuals can also be identified from gene expression profiles (Schadt, Nature Genetics 2012; doi: 10.1038/ng.2248), our data protection officer does not allow us to put patient-derived RNA-seq data into GEO. However, we can make the data accessible using individual data transfer agreements (DTA) upon request. Unless you inform us about another legal option to make the data publicly available, we will state in the manuscript that all RNA-seq data can be obtained upon request via DTA. For the low pass and CGH data, we suggest to place CNV information for each individual cell as full karyotype information according to the ISCN annotation.

Reviewer#2:

The authors have addressed the overwhelming majority of the comments raised during the review rounds, and this reviewer is generally satisfied.

Reviewer#3:

The authors provided answers to some of the concerns raised by the reviewers. These answers are only partially satisfactory and weaknesses still remain. I provide some comments below with the intent to give the authors some constructive criticism and to help the editors with their final decision.

(1) Statements on KIT expression are hard to “decipher”. First of all, the scalebar in Fig.3b is unbalanced (only 0 is blue, while 3, 6, 9 are fully red) and given the way it is done, prone to visually highlight possibly minor differences. It is also unclear what exactly are the values 0, 3, 6, 9 referring to: if these are TPM, all should be considered very low expression levels, particularly in light of obvious biases due to single cell gene expression profiles (cell cycle etc etc). All in all, certainly statements such as “strong expression of the luminal progenitor marker KIT...” do not seem to be supported by data at this stage.

Reply: We apologize for using a two-color scale bar which has now been replaced by a three-color scale. Furthermore, we added to the y-axis the information that gene expression levels are provided as log₂-normalized counts as it is standard routine in single cell analyses.

To enable a comparative judgement, we also provide the results of our detection antigen, EpCAM, which served to isolate the cells and is also expressed on luminal progenitor cells (as opposed to MaSCs). The results demonstrate that KIT expression is significantly higher in M0-DCCs as compared to EpCAM-positive cells from non-cancer patients. Furthermore, M0-DCCs express KIT stronger than M1-DCCs. The EpCAM-data are consistent with what we saw in the microscope: EpCAM expression is higher in cancer cells as compared to bone marrow-derived control cells and highest in M1-DCCs. Together, the EpCAM and KIT data are consistent with observations that expression of EpCAM is lower in luminal progenitor cells than in more mature luminal cells and vice versa for KIT.

We suggest to place the PCA analysis in the figure 3b as before and the box plots showing the log₂-normalized counts in the supplement.

Modified Figure 3b and novel supplementary Fig S1.

Color code has been changed to a three-color scale reflecting the log₂-normalized counts (upper panels)

Lower panels: Log₂-normalized counts of all cells from upper panel for *EpCAM* and *KIT*. Differences between groups are indicated by the stars (Wilcoxon-Mann-Whitney Test). Expression of *EpCAM* in M1 cells compared to M0-stage DCCs is of borderline significance. *KIT* expression is highest in M0-stage DCCs.

(2) My previous questions in regard to the *in vivo* findings remain unanswered. The authors keep insisting that only one cell line is sufficient: the argument of luminal progenitor-derived is totally irrelevant in this context, and sentences like “we would not even know how we could justify this experiment to obtain permission for additional animal experiments from the government” are hardly understandable. I remain of the opinion that they should exclude cell-line dependent effects for their *in vivo* experiments, and add additional models to improve their manuscript.

Reply: We are a bit surprised about the comments of the reviewer. In the previous review, the reviewer classified this aspect of “minor” importance. We therefore may not have paid sufficient attention to provide a full answer, but provided only the requested data on proliferation and apoptosis by performing cytometric quantification (Fig. 5i).

We do not state that one cell line is sufficient; on the contrary, our experiments demonstrate the opposite. We provide data (i) from more than 16 different patients with reduction mammoplasty; (ii) data from two cell lines with close-to-normal (i.e. non-tumorigenic cells) genomes (MCF-10A, HME) and (iii) two cancer cell lines (MCF-7 and MDA-MB-231). Furthermore, we control the MCF-10A experiments by using a syngenic knock-in of PIK3CA mutant. Our *in vivo* experiments address the effect of IL6-transignaling on engraftment and therefore colony formation. We use MDA-231 and normal mammary epithelial cells to cover the spectrum of normal to de-differentiated phenotypes (as the reviewer would probably agree luminal cancers are more differentiated than TN-BC.) We show that IL-6 trans-signaling acts on the level of progenitor cells *in vitro* and our *in vivo* and bioinformatics results (Figure 5) are fully consistent with this. In our opinion additional transplantation experiments would add little, as there is no reason to believe that IL6-transignaling responsive cell lines would not be responsive *in vivo*. The transplantation of normal mammary epithelial cells derived from various donors minor the risk of “cell-line dependent” effects (which are likely to always exist to some degree as each cancer is individual) and we are struggling with the

statement that “the argument of luminal progenitor-derived is totally irrelevant” as this is the biology we are addressing.

The statement referring to the animal permission is easy to explain. Due to corona we were forced to completely shut down all animal experiments and associated breedings at the time of the last review. Now, selected experiments are possible again, however only with extremely well-grounded justification that is carefully reviewed by a committee. Given the reasons above, we cannot justify experiments that the reviewer called of minor importance and that is in our opinion unlikely to reveal additional information be classified as extremely important.

Further, in figure 5 (despite adventurous titles) I still see very little stem-like abilities: asymmetric cell division (self-renewal + differentiation), tumor seeding ability in limiting dilution assays, ability to recreate an heterogeneous lesion from a single cell are required assays for statements of this kind, and none of this is presented. Tumorsphere size and CD44/24 are surrogate assays with very questionable validity, and tumor volume has really nothing to do with stem-like traits.

Reply: We have difficulties to understand the point of the reviewer. Gabriela Dontu has shown in her landmark paper (Genes and Development 2003) that sphere-forming cells divide asymmetrically, have differentiation potential and display self-renewal ability. Our studies build on this paper that has been quoted 2500 times in the field. We use the same nomenclature and are equally careful about it and we show in many experiments (label-retention assay for asymmetric division, mammosphere assay for colonization/self-renewal and proliferation ability; in vivo engraftment and differentiation ability) that the use of this nomenclature is justified. We therefore kindly disagree that our respective title would be adventurous. The bioinformatics analysis in Fig. 5c further supports our reasoning with respect to IL6-signaling effects and the in vivo data in Fig. 5e, f and h demonstrate in vivo effects of IL6 signaling for engraftment, proliferation and differentiation ability.

However, as note of caution, we added the following sentence to the text describing the in vivo experiments:

While this is consistent with an elevated stemness of MDA-MB-231 cells, limiting dilution experiments, ideally performed with additional cell lines, would be needed to fully establish the stemness-conferring activity of IL6-trans-signaling to human cell line models in xenotransplantation.